# Structural mechanisms for centrosomal recruitment and organization of the microtubule nucleator γ-TuRC

Qi Gao[1,5], Florian W. Hofer [1,5], Sebastian Filbeck [1,5], Bram J. A. Vermeulen [1,5], Martin Würtz [1,2], Annett Neuner[1], Charlotte Kaplan[3], Maja Zezlina[1], Cornelia Sala [1], Hyesu Shin[1], Oliver J. Gruss [4], Elmar Schiebel [1]✉ & Stefan Pfeffer [1]✉

The γ-tubulin ring complex (γ-TuRC) acts as a structural template for microtubule formation at centrosomes, associating with two main compartments: the pericentriolar material and the centriole lumen. In the pericentriolar material, the γ-TuRC is involved in microtubule organization, while the function of the centriole lumenal pool remains unclear. The conformational landscape of the γ-TuRC, which is crucial for its activity, and its centrosomal anchoring mechanisms, which determine γ-TuRC activity and turnover, are not understood. Using cryo-electron tomography, we analyze γ-TuRCs in human cells and purified centrosomes. Pericentriolar γ-TuRCs simultaneously associate with the essential adapter NEDD1 and the microcephaly protein CDK5RAP2. NEDD1 forms a tetrameric structure at the γ-TuRC base through interactions with four GCP3/MZT1 modules and GCP5/6-specific extensions, while multiple copies of CDK5RAP2 engage the γ-TuRC in two distinct binding patterns to promote γ-TuRC closure and activation. In the centriole lumen, the microtubule branching factor Augmin tethers a condensed cluster of γ-TuRCs to the centriole wall with defined directional orientation. Centriole-lumenal γ-TuRC-Augmin is protected from degradation during interphase and released in mitosis to aid chromosome alignment. This study provides a unique view on γ-TuRC structure and molecular organization at centrosomes and identifies an important cellular function of centriole-lumenal γ-TuRCs.

The de novo assembly of microtubules (MTs) from α/β-tubulin dimers is regulated by the γ-tubulin ring complex (γ-TuRC), which serves as a structural template for MT nucleation by providing binding sites for α/β-tubulin[1]. The molecular architecture of the biochemically isolated vertebrate γ-TuRC core has been resolved using cryo-electron microscopy (cryo-EM) single particle analysis (SPA). The γ-TuRC has a helical shape consisting of 14 spokes, each containing a copy of γ-tubulin presented by the C-terminal gamma ring protein 2 (GRIP2) domain of one of five paralogous γ-tubulin complex proteins (GCPs). The GCP proteins interact via their N-terminal GRIP1 domains in a specific order: the γ-TuRC core harbors four pairs of GCP2/GCP3 (spoke 1–8), followed by a sequence of vertebrate-specific paralogs GCP4/GCP5/GCP4/GCP6 (spoke 9-12) and finally another pair of GCP2/GCP3 (spoke 13-14)[2–5]. The inside of the γ-TuRC is lined by a lumenal bridge

[1]Zentrum für Molekulare Biologie der Universität Heidelberg (ZMBH), Heidelberg, Germany. [2]European Molecular Biology Laboratory (EMBL), Heidelberg, Germany. [3]BioQuant, Universität Heidelberg, Heidelberg, Germany. [4]Institut für Genetik, Universität Bonn, Bonn, Germany. [5]These authors contributed equally: Qi Gao, Florian W. Hofer, Sebastian Filbeck, Bram J. A. Vermeulen. ✉e-mail: e.schiebel@zmbh.uni-heidelberg.de; s.pfeffer@zmbh.uni-heidelberg.de

composed of one actin molecule[2–5], as well as two α-helical modules formed by the microprotein MZT1 (mitotic-spindle organizing protein 1) and the N-terminal extensions of GCP3 and GCP6 (N-GCP3 and N-GCP6), respectively[5–7]. The remaining GCP proteins, except for GCP4, also contain binding sites for MZT1 (GCP3, 5, 6) or its paralog MZT2 (GCP2) in their N-terminal extensions, which are separated from the GRIP1 domains by long, largely disordered linkers and hence resolved by cryo-EM only under specific circumstances[4,7–9]. In addition, the N-terminal extensions of GCP5 and GCP6 contain various secondary structure elements docked directly at the bottom of their GRIP1 domains[5,7] and spanning the surface of virtually all other spokes, rendering even the surfaces of recurring GCP2-, GCP3- and GCP4-containing spokes structurally unique.

In γ-tubulin complexes, the exposed γ-tubulin molecules act as templates for the de novo assembly of MTs[10]. Remarkably, the isolated vertebrate γ-TuRC assumes an asymmetric open conformation in which the arrangement of γ-tubulin molecules deviates from MT geometry, especially in the last few spokes[2–5]. In the absence of structural data on the γ-TuRC in cells, it is unclear if this open conformation faithfully reflects the situation in vivo.

The localization of γ-TuRCs to the centrosome, the predominant MT-organizing center (MTOC) in vertebrate cells, requires different γ-TuRC-interacting proteins. The adapter protein NEDD1 (neural precursor cell expressed, developmentally down-regulated 1), CDK5RAP2 (Cyclin-dependent kinase 5 regulatory subunit-associated protein 2), and CEP192 (centrosomal protein of 192 kDa) are required for γ-TuRC modulation and targeting to the pericentriolar material (PCM) of centrosomes[11–19]. Recently, light microscopy data indicated that a second pool of γ-TuRCs resides inside the centriolar lumen[20,21], recruited by NEDD1, the inner scaffold protein POC5 (proteome of centriole 5) and the MT branching factor Augmin[21]. In different structural studies, the isolated CM1 motif (centrosomin motif 1) of CDK5RAP2 was shown to bind either only to GCP2 at spoke 13 (hereafter referred to as GCP2(13))[4] or to all five GCP2 subunits of the complex[8,9]. Which binding sites are physiologically relevant in vivo, however, has remained elusive. Biochemical studies have revealed that the key adapter protein NEDD1 co-purifies with the γ-TuRC[2–4,22]. Available data suggest that N-GCP3/MZT1 modules of the γ-TuRC[23] interact with the tetrameric NEDD1 C-terminus (NEDD1^C)[15,24]. However, the structural underpinnings and stoichiometry of NEDD1 binding to the γ-TuRC remain unclear.

Uncovering the precise conformational landscape and distribution of γ-TuRC in cells and the structural mechanisms underlying its localization and activation at the centrosome is critical for our understanding of how cells assemble and organize MTs. Here, we study the structure and organization of the γ-TuRC using cryo-electron tomography and super-resolution fluorescence microscopy techniques in situ, complemented with high-resolution cryo-EM SPA of isolated complexes. We characterize both the structure and binding mode of the γ-TuRC adapter protein NEDD1 and identify two interaction modes of CDK5RAP2's CM1 motif with the γ-TuRC. Lastly, we uncover the spatial organization of γ-TuRC, NEDD1, Augmin, and POC5 inside centrioles and identify a protective function for γ-TuRC recruitment to the centriole lumen. Our data provide an unprecedented understanding of the organization of the γ-TuRC in its natural environment.

## Results
### Cryo-ET imaging of PCM-embedded γ-TuRCs identifies a novel binding site for γ-TuRC interactors
To image and analyze γ-TuRCs embedded in the native PCM, we applied cryo-electron tomography on centrosomes purified from human KE37 cells (Fig. 1a and Supplementary Fig. 1a, b). After pre-processing, tilt-series alignment, and tomogram reconstruction (Supplementary Fig. 2a, "Methods"), we assessed the quality of tomograms and retained only those depicting intact centrioles surrounded by a dense PCM. We frequently observed γ-TuRCs as ring-shaped multi-spoked particles within the PCM, but only rarely within the centriole lumen, where ice thickness was highest (Supplementary Fig. 1b).

To analyze the structure of the γ-TuRC embedded in the native PCM, we manually located γ-TuRC-shaped particles in five tomograms and subjected the subvolumes to reference-free alignment, in which the average of particles iteratively evolved from a featureless sphere to a low-resolution density of the γ-TuRC (Supplementary Fig. 2b). Using the density obtained from reference-free alignment as a template, γ-TuRCs were automatically and comprehensively localized within all tomograms by template matching and computational particle sorting (Supplementary Fig. 2b, c). The retained subvolumes were refined to a cryo-EM density of the γ-TuRC at 22.4 Å resolution in RELION 3.1 (Supplementary Fig. 2c, d)[25,26]. Using ribosomes distributed around the PCM in our tomograms as fiducial markers for local tilt-series refinement in M[27] (Supplementary Fig. 2b,c), the density of the γ-TuRC could be improved to 16.4 Å (Supplementary Fig. 2d). Docking an atomic model of the human γ-TuRC confirmed that all expected structural elements of the γ-TuRC, including the lumenal bridge with an integrated molecule of actin, were resolved in situ (Fig. 1b).

Notably, the cryo-EM density of the native γ-TuRC featured a large stoichiometric density segment at the narrow end of the γ-TuRC cone that was not resolved in available cryo-EM structures of the purified human γ-TuRC, identifying a previously unknown binding site for γ-TuRC interactors (Fig. 1b).

### The NEDD1 adapter protein binds the native γ-TuRC as a MZT1 module-associated tetramer
We next aimed to identify the γ-TuRC interactor resolved in the cryo-EM reconstruction of the native PCM-embedded γ-TuRC. We considered the essential γ-TuRC adapter protein NEDD1 as a prime candidate because it was identified as a crucial factor for centrosomal recruitment of the γ-TuRC[15,28], but its binding site was unknown.

Firstly, we wanted to test whether the structure of NEDD1 predicted by AlphaFold2[29,30] was compatible with the shape of the unidentified density segment. As the C-terminus of NEDD1 was previously suggested to tetramerize in isolation[24], and NEDD1 was reported to interact with N-GCP3/MZT1 modules[23], we used AlphaFold2 to predict the structure of tetrameric NEDD1 C-terminus in complex with four N-GCP3/MZT1 modules. AlphaFold2 confidently predicted a grapnel-like structure, in which the N-GCP3/MZT1 modules dock onto a 4-fold coiled-coil formed by the C-terminal segments of NEDD1 (Fig. 1c, Supplementary Fig. 3a–c). The AlphaFold2 model could be seamlessly docked as a rigid body into the unoccupied density segment capping the γ-TuRC cone (Fig. 1d), providing a first indication that NEDD1 may bind as an N-GCP3/MZT1-associated tetramer to the cone of the γ-TuRC.

Aiming to confirm the model predicted by AlphaFold2, we next created a FLAG-tagged *Xenopus laevis* NEDD1-N-GCP3 fusion construct and co-expressed it in insect cells together with MZT1 (Supplementary Fig. 4a, b), which was readily co-purified, suggesting successful complex formation between MZT1 and N-GCP3 (Supplementary Fig. 4c). Consistent with previous data for the isolated NEDD1 C-terminus[15,24], mass photometry indicated that NEDD1-N-GCP3/MZT1 was able to form stable tetramers at the expected size of ~460 kDa, clearly supporting the oligomeric state of NEDD1 predicted by AlphaFold2 (Supplementary Fig. 4d). Negative stain electron microscopy of purified NEDD1-N-GCP3/MZT1 complexes in conjunction with fully data-driven and reference-free particle averaging produced a 3D density highly similar to the AlphaFold2 model in terms of dimensions and shape (Supplementary Fig. 4e–g). Overall, these data indicate that AlphaFold2 correctly predicted the structure of NEDD1 in complex with MZT1 modules.

To further support our assignment of NEDD1 and to gain high-resolution insights into the interaction of NEDD1 with the vertebrate γ-TuRC, we used *X. laevis* egg extract to purify γ-TuRCs for cryo-EM SPA

analysis[3]. Mass spectrometry analysis indicated that γ-TuRC components and NEDD1 were among the most abundant proteins in the sample (Supplementary Table 1 and Supplementary Data 1), suggesting that NEDD1 was co-purified with γ-TuRC at high stoichiometry. NME7, another known γ-TuRC interactor[31] abundant in the sample, could be excluded to contribute to the grapnel density based on its structure, which does not contain any coiled-coil segments (PDB 7UNG[32], PDB 8J07[33], PDB 7RRO[34]). We subjected the purified *X. laevis* γ-TuRCs to cryo-EM SPA and identified a subset of particles with stoichiometric NEDD1/N-GCP3/MZT1 grapnel density (Fig. 2a, Supplementary Fig. 5a–c and Supplementary Table 2). Local resolution ranged from 3.9 Å in the core of the complex to 5-8 Å for the NEDD1 segment (Supplementary Fig. 5d–f), where the resolution was initially limited by the plasticity of the NEDD1 segment with respect to the γ-TuRC core (Supplementary Fig. 5g). We, therefore, subjected the NEDD1/N-GCP3/MZT1 density segment to focused refinement (see "Methods"), improving local resolution for NEDD1 to 4-5 Å (Supplementary Fig. 5a, f, h, i). Next, we performed molecular dynamics flexible fitting (MDFF)[31] of the AlphaFold2 model into the consensus cryo-EM density, resulting only in minor changes. The very C-terminal NEDD1 helix segments were an exception, as they were predicted straight, but are kinked in the cryo-EM reconstruction (Supplementary Fig. 6a). After optimizing the model by another round of MDFF into the density obtained by focused refinement (Supplementary Fig. 3d–f), resolved bulky hydrophobic side chains and the length of resolved alpha-helices confirmed the density segment as NEDD1 (Fig. 2b).

Notably, the N-terminal beta-propeller domain of NEDD1 was not resolved, most likely because they are separated by long and flexible linkers from the stably docked NEDD1[C] helices and, therefore, averaged out. Moreover, the conformation of the γ-TuRC was not affected by the presence or absence of NEDD1 (Supplementary Fig. 6b), suggesting that NEDD1 binding does not activate the γ-TuRC for MT nucleation[35].

## A complex multivalent interaction network embeds NEDD1 into the γ-TuRC structure

Our atomic model allowed detailed analysis of the interaction between NEDD1, the N-GCP/MZT1 modules, and the γ-TuRC core. We first aimed to identify the N-GCP/MZT1 modules docked on NEDD1. The single N-GCP6/MZT1 module, as well as one of five N-GCP3/MZT1 modules, are embedded within the γ-TuRC core structure as part of the lumenal bridge and are thus not available to interact with NEDD1 in the grapnel[7]. In addition to the two N-GCP/MZT1 modules in the lumenal bridge, a third module of unclear identity is stably docked and resolved on spoke 14 of the γ-TuRC. Cross-correlation analysis against atomic models for N-GCP5/MZT1 and N-GCP3/MZT1 modules

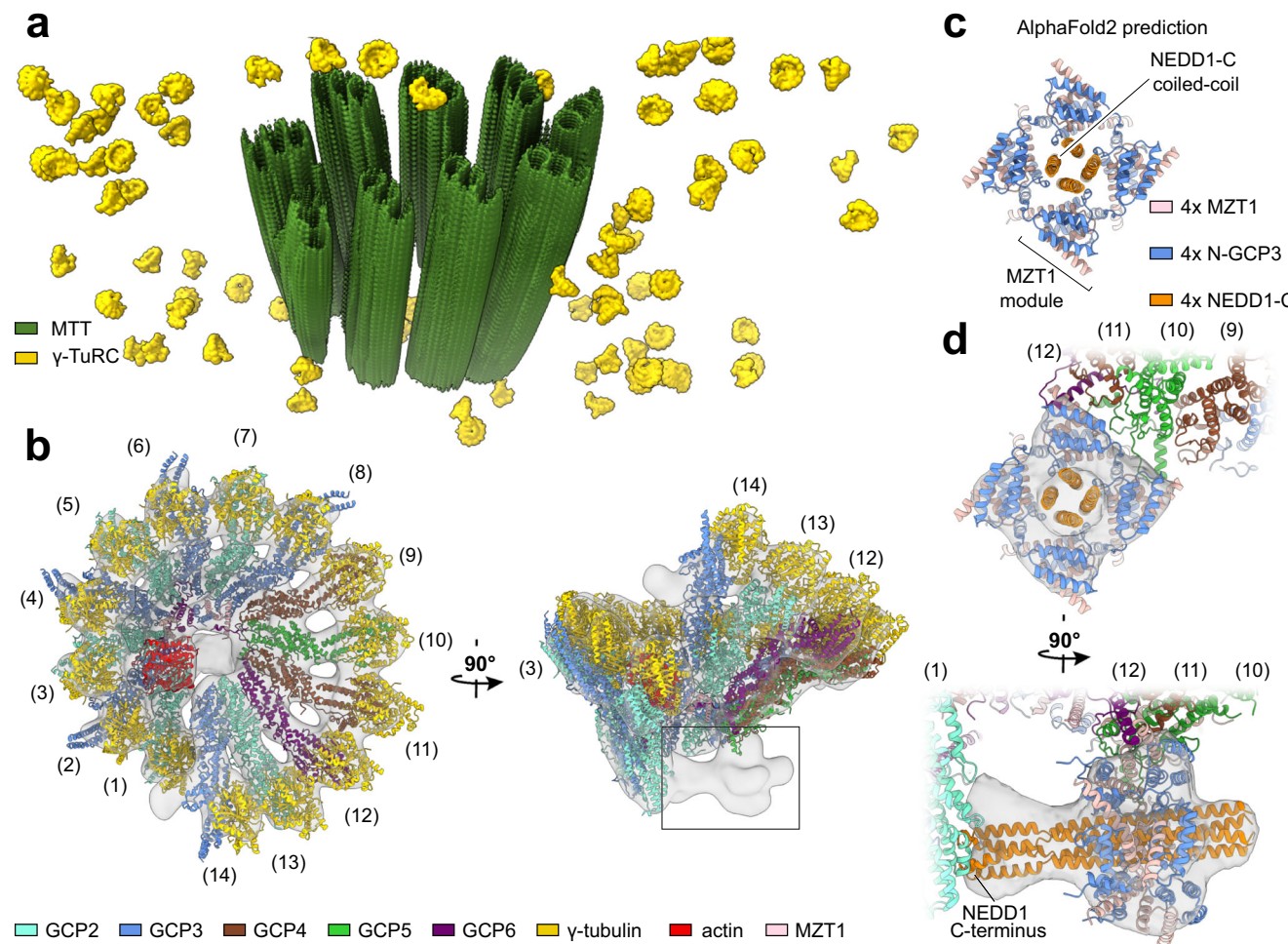

**Fig. 1 | Cryo-ET imaging of pericentriolar γ-TuRCs identifies a novel binding site for γ-TuRC interactors. a** Three-dimensional rendering of a cryo-electron tomogram depicting a centrosome purified from human KE37 cells. Microtubule triplets (MTTs) and identified γ-TuRCs are annotated in green and yellow, respectively. **b** Cryo-EM density of the human γ-TuRC from purified centrosomes at 16.4 Å resolution. The atomic model of the recombinant human γ-TuRC was superposed (PDB: 7QJ5[7]; N-GCP3/MZT1 and N-GCP5/MZT1 not shown). Coloring as indicated; box indicates density segment not previously observed on the γ-TuRC. **c** AlphaFold2 prediction for the tetrameric NEDD1[C]/N-GCP3/MZT1 complex (disordered regions not shown). The tetrameric NEDD1[C] coiled coil is surrounded by four symmetrically arranged N-GCP3/MZT1 modules. Coloring as indicated. **d** AlphaFold2 model from (**c**), rigid body-docked into the density segment from (**b**).

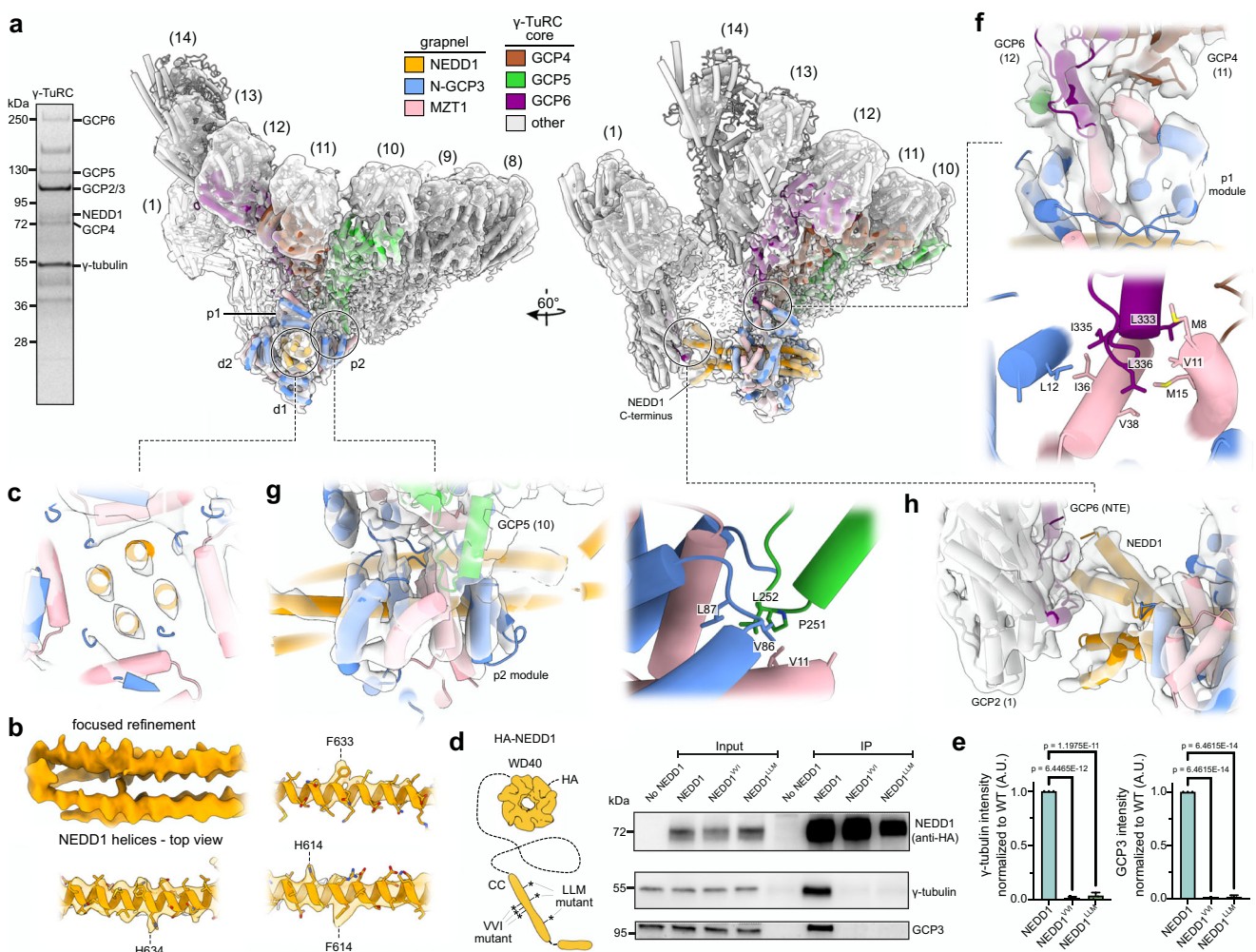

**Fig. 2 | A complex and multivalent interaction network embeds NEDD1 into the γ-TuRC structure. a** Left: Coomassie-stained SDS-PAGE gel of γ-TuRC affinity-purified from *X. laevis* egg extract. $N = 2$ independent experimental repeats. Right: Consensus cryo-EM SPA reconstruction of the γ-TuRC with stoichiometric NEDD1 grapnel with fit atomic model after MDFF. Coloring as indicated. The numbering of spokes and N-GCP3/MZT1 modules (p = proximal, d = distal to the γ-TuRC) is indicated. Density was weighted using OccuPy[93]. Spokes 13 and 14 are fully covered at a lower density threshold (Supplementary Fig. 5j). **b** Top left: Top view of the 4 long NEDD1[C] helices after focused refinement. Other panels: exemplary aromatic side chain densities in the coiled-coil helices of the focused cryo-EM SPA reconstruction, supporting their assignment to NEDD1[C]. **c** Slice through the NEDD1 grapnel in the consensus cryo-EM SPA reconstruction and associated model, highlighting the tetrameric coiled-coil formed by NEDD1 at the center of the grapnel. **d** Left: schematic overview of HA-NEDD1 protomer architecture, indicating the location of mutated residues in NEDD1[VVI] (V612A, V616A, I619A; human numbering) and NEDD1[LLM] (V623, V627, I630; human numbering). Right: representative

immunoprecipitation results from HEK293 cell lysates without NEDD1 or containing overexpressed HA-tagged wild-type NEDD1, NEDD1[VVI], or NEDD1[LLM] mutants. Anti-HA NEDD1 immunoprecipitations were analyzed for co-immunoprecipitation of γ-tubulin and GCP3 by immunoblotting. **e** Quantification (mean ± SD) of relative band intensities from the NEDD1-γ-TuRC immunoprecipitation experiment in (**d**). Co-immunoprecipitated γ-tubulin was normalized to NEDD1, with the γ-tubulin:NEDD1[WT] ratio set to 1. Quantification based on $N = 3$ independent experiments. Statistical analysis was performed using ordinary one-way ANOVA. **f–h** Zoom-in of the consensus cryo-EM SPA reconstruction and associated model, highlighting the interaction of the helix-turn-loop element in GCP5 with the p2 N-GCP3/MZT1 module (**f**, left) and the hydrophobic interactions at this interface (**f**, right), the interaction of GCP6 at spoke 12 with the p1 N-GCP3/MZT1 module (**g**, top) and the hydrophobic interactions at this interface (**g**, bottom), and the interaction of the upper NEDD1 dimer with the N-terminal extension (NTE) of GCP6 located at spoke 1 (**h**). Coloring in panels b,c,f,g and h as in panel a. Source data are provided as a Source Data file.

(Supplementary Table 3 and Supplementary Fig. 7a) indicated that the density segment on spoke 14 corresponds to the single N-GCP5/MZT1 module in the complex. Vice versa, three out of four N-GCP/MZT1 modules bound to NEDD1 showed a higher correlation towards N-GCP3/MZT1, while the remaining N-GCP/MZT1 module showed no unambiguous preference for either N-GCP3/MZT1 or N-GCP5/MZT1, consistent with the lower local resolution. Notably, cross-correlation analysis suggested that none of the resolved MZT modules corresponds to an N-GCP2/MZT2 module (Supplementary Table 4 and Supplementary Fig. 8a–c). Overall, this indicates that the NEDD1 tetramer interacts with the remaining four available N-GCP3/MZT1 modules.

Our cryo-EM reconstruction resolves a tetramer of the NEDD1 C-terminus, comprising two consecutive helices in each protomer (NEDD1[598-648] and NEDD1[651-670]; all residue numbers refer to *X. laevis* proteins). At its center, NEDD1[598-648] forms a tetrameric coiled-coil that provides the binding site for four N-GCP3/MZT1 modules (Fig. 2c). The interaction between NEDD1 and the N-GCP3/MZT1 modules is characterized by complementary charged and hydrophobic patches (Supplementary Fig. 7b). In addition to direct interaction with NEDD1, the N-GCP3/MZT1 modules also share an extensive interaction surface with each other (Supplementary Fig. 7c), suggesting cooperative binding of N-GCP3/MZT1 modules to NEDD1. To test the model for molecular architecture of the NEDD1 grapnel, we designed point

mutations in NEDD1 expected to disrupt either NEDD1 tetramerization (*NEDD1^LLM*; human residues: L600D, L614M, M629D; *X. laevis* equivalent residues*: L611D, L625D, I640D) or binding of N-GCP3/MZT1 modules to NEDD1^C via the hydrophobic interface shown in Supplementary Fig. 7b (*NEDD1^VVI*; human residues: V612A, V616A, I619A; *X. laevis* equivalent residues: V623A, V627A, I630A) (Fig. 2d). HA-tagged wild-type NEDD1 (NEDD1^WT) and mutant variants were overexpressed in HEK293 cells and the interaction with the γ-TuRC was probed by HA-pulldown. The γ-TuRC co-eluted with HA-NEDD1^WT, but the interaction was completely lost for both NEDD1 mutant variants (Fig. 2d, e). This further supports our model for the NEDD1 grapnel architecture and indicates that both NEDD1 tetramerization and the interaction between NEDD1^C and the N-GCP3/MZT1 modules are essential for γ-TuRC binding.

The NEDD1 grapnel interacts with the asymmetric side of the γ-TuRC through docking of the two proximal N-GCP3/MZT1 modules (p1, p2) to GCP5- and GCP6-specific extensions just N-terminal to the GRIP1 domains at spokes 10-12 (Fig. 2a, f, g), respectively. Consistent with AlphaFold2 predictions of the interface, a helix-turn-loop structure specific to GCP5 inserts the conserved GCP5 residue Leu252 (Supplementary Fig. 7d) into a hydrophobic pocket formed by the N-GCP3/MZT1 module p2 (Fig. 2g), which is supported by basic residues interacting with aromatic and acidic residues on GCP5 (Supplementary Fig. 7e). Similarly, a GCP6-specific helix extends onto the surface of the N-GCP3/MZT1 module p1, where the hydrophobic triad Leu333, Ile335 and Leu336 of GCP6 binds to hydrophobic patches on the N-GCP3/MZT1 module (Fig. 2f), as also supported by AlphaFold2 predictions. Notably, the same interface on the N-GCP3/MZT1 module is involved in both of the observed interactions as well as the integration of the N-GCP3/MZT1 module in the lumenal bridge (Supplementary Fig. 7f)[7].

In contrast to NEDD1^598-648, the NEDD1^651-670 helix breaks the fourfold symmetry and forms two small dimeric coiled-coils that splay out from each other (Fig. 2h). One protomer in the upper dimer of NEDD1^651-670 contacts GCP6^223-240, which is part of a larger region of N-GCP6 that lines the inside of GRIP1 domains of spoke 1 and 2 (Fig. 2h and Supplementary Fig. 3m−o)[7,36]. This puts the NEDD1 grapnel in a unique position, bridging both ends of the γ-TuRC helix. The lower dimer of NEDD1^651-670 appears to make little to no contact to spoke 1 and 2 (Fig. 2h), but its conformation is likely stabilized by an unidentified wedge density between the upper and lower NEDD1^651-670 dimers (Supplementary Fig. 7g).

Cumulatively, we identified the binding site and structure of tetrameric NEDD1 on the γ-TuRC, forming an adapter for γ-TuRC-interacting proteins underlying the centrosomal organization of γ-TuRCs.

## Conformation-dependent pattern of CDK5RAP2 CM1 module recruitment to pericentriolar γ-TuRCs

After having structurally characterized the universal γ-TuRC adapter protein NEDD1, we next focused on the conformational plasticity of pericentriolar γ-TuRCs imaged in purified human centrosomes. Computational particle sorting revealed two distinct γ-TuRC conformations (Fig. 3a, Supplementary Fig. 2c and Supplementary Fig. 9a, b). In the first class, the overall conformation is similar to the isolated vertebrate γ-TuRC (Supplementary Fig. 9c)[4], and spokes 13 and 14 are positioned outwards from the helical axis (outwards conformation of spokes 13 and 14). In the second class, a hinge motion of GRIP2-γ-tubulin segments relative to GRIP1 domains coordinated around the entire circumference of the complex (Fig. 3b) brings the γ-tubulin molecules on spokes 1−8 into an almost perfectly MT-compatible conformation (Fig. 3c) and spokes 13 and 14 move slightly inward towards the helical axis of the γ-TuRC (inwards conformation of spokes 13 and 14).

In both classes, prominent GCP2-associated densities decorate the outside of the γ-TuRC. To resolve these densities at higher resolution, we treated each GCP2/3 unit (γ-tubulin small complex, γ-TuSC)

as an individual particle and subjected them to refinement and 3D classification (Supplementary Fig. 2e). We observed that the GCP2-associated extra densities precisely matched the position and shape of CM1-interacting N-GCP2/MZT2 modules (Fig. 3d and Supplementary Fig. 10a; hereafter referred to as CM1 modules), as previously observed on GCP2(13)[4] and more recently on all GCP2 subunits of the γ-TuRC[8,9], as well as predicted by AlphaFold3[37] (Supplementary Fig. 8d−f). To confirm that the extra densities represent CM1 modules, we tested whether they were dependent on CDK5RAP2, the most likely CM1 motif-containing candidate protein colocalizing with γ-tubulin in the PCM (Supplementary Fig. 10b, c). Using an RPE1 *CDK5RAP2* knockout (*CDK5RAP2^-/-*) cell line[14], we observed through immunofluorescence microscopy that the γ-tubulin signal at centrosomes was unaffected by the absence of *CDK5RAP2* (Supplementary Fig. 10c, d), as reported before[14]. We thus compared γ-TuRC structures in centrosomes purified from wild-type and RPE1 *CDK5RAP2^-/-* cells using cryo-electron tomography (Supplementary Fig. 10e and Supplementary Fig. 11a, b). While well-resolved for wild-type RPE1 cells, the extra densities were no longer present at any of the GCP2 subunits for centrosomes purified from *CDK5RAP2^-/-* cells (Supplementary Fig. 10e), indicating that they indeed represent CDK5RAP2-derived CM1 modules.

Examining the distribution of CM1 modules on our γ-TuRC structures, we unexpectedly observed CM1 modules in two distinct patterns, highly correlating with the overall γ-TuRC conformation (Fig. 3e and Supplementary Fig. 12a). In the outwards conformation, density for CM1 modules could be detected exclusively on spoke 13, as described before[4]. In contrast, in the inwards conformation, the CM1 module binding site on GCP2 at spoke 13 was vacant and the γ-TuRC harbored CM1 modules on the remaining GCP2 subunits at spokes 1, 3, 5, and 7, suggesting the CM1 module binding to these spokes is associated with partial ring closure.

Next, we investigated whether there is simultaneous binding of NEDD1 and CDK5RAP2 to the same γ-TuRC particle, possibly allowing crosstalk between these two γ-TuRC-interacting factors. We selected the set of γ-TuSC units observed to contain CM1 modules (Supplementary Fig. 2e), identified the corresponding γ-TuRC particles, and averaged them to obtain a cryo-EM density of γ-TuRCs associated with at least one CM1 module. In the resulting cryo-EM reconstruction, NEDD1 density was stoichiometrically present (Supplementary Fig. 12b), indicating that CDK5RAP2 and NEDD1 can indeed simultaneously bind γ-TuRCs at centrosomes. At the same time, the stoichiometry of NEDD1 at the base of the γ-TuRC remained unaffected by the absence of CDK5RAP2 (Supplementary Fig. 10e), suggesting that NEDD1 and CDK5RAP2 binding to γ-TuRCs is independent.

Cumulatively, we show how multiple copies of the CDK5RAP2 CM1 motif bind to the γ-TuRC in different patterns that are highly correlated with γ-TuRC conformation. This indicates that the concentration of available CM1 modules, together with other factors may regulate and control γ-TuRC conformation and activity in the PCM.

## Mapping the centrosomal distribution of γ-TuRCs at single molecule precision reveals two distinct pools of γ-TuRCs in cells

Biochemical isolation and preparation of cryo-EM grids could potentially influence PCM organization in purified centrosomes. Aiming to faithfully dissect the native centrosomal distribution of the γ-TuRC, we thus used cellular cryo-electron tomography to image centrosomes in vitreous sections of human HCT116 cells produced by focused ion beam milling. For statistical analyses, we extended the dataset with publicly available tilt series of centrosomes in vitreous sections of HeLa cells (EMPIARC-200003)[38]. For tilt-series processing and quality assessment, we followed a similar approach as for the purified centrosomes (Supplementary Fig. 13a−c). Subtomogram analysis enabled us to map the distribution of γ-TuRCs within the native centrosome at single molecule precision (Fig. 4a) and to obtain a structure of the γ-TuRC at 30.6 Å resolution (Supplementary Fig. 13d). This confirmed

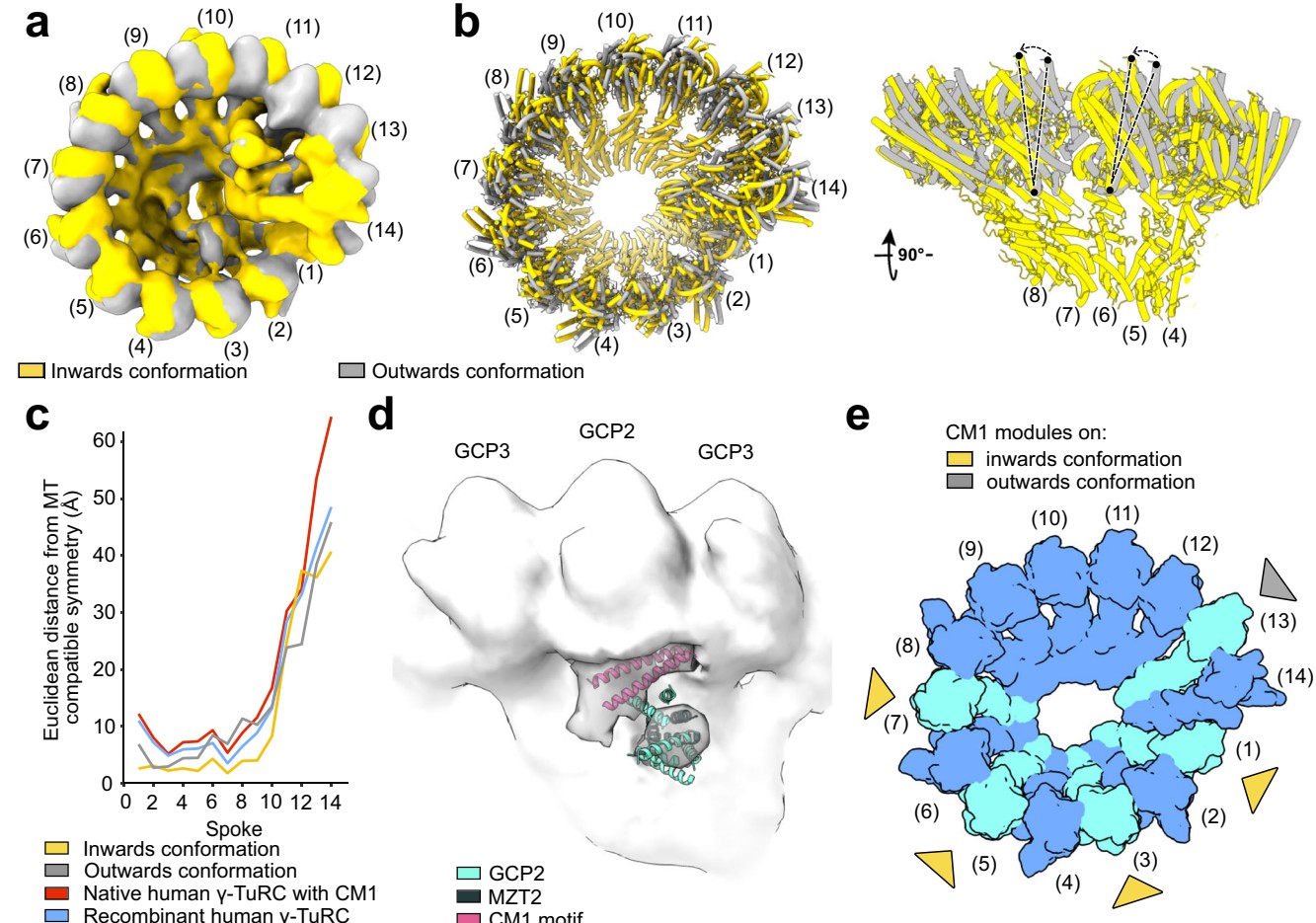

**Fig. 3 | Conformation-dependent pattern of CDK5RAP2 CM1 module recruitment. a** Cryo-EM densities of the inwards (yellow) and outwards (gray) conformations superposed according to GCP GRIP1 domains. **b** Superposed rigid body-fitted models of the inwards and outwards conformations seen in the top (left) and side view (right). See Supplementary Fig. 9 and Methods for details. Arrows indicate the rigid body motion of GRIP2 domains and γ-tubulins. **c** Deviation of γ-TuRC geometry from the fully closed MT-capping human γ-TuRC (PDB: 8VRK[53]). The inwards and outwards conformations of PCM-located human γ-TuRCs, as well as purified recombinant (PDB: 7AS4[5]) and native human γ-TuRCs (PDB: 6V6S[4]) were analyzed. All models were superposed according to spokes 2–8, and deviations were calculated based on the γ-tubulin centers determined in ChimeraX using the "measure center" command. **d** Cryo-EM density of a GCP2/3 unit (white) with a stoichiometric additional density segment on the outside of GCP2 (gray). The experimental model of the CDK5RAP2 CM1 module was superposed (PDB: 6V6S[4]). **e** Schematic representation of conformation-dependent CDK5RAP2 CM1 module binding for the inwards (yellow) and outwards (gray) conformations (see also Supplementary Fig. 12a). GCP2-containing spokes are indicated in light blue. Source data are provided as a Source Data file.

the asymmetric conformation and overall architecture of the γ-TuRC as observed within purified centrosomes (Supplementary Fig. 13e).

We next aimed to characterize the centrosomal organization of γ-TuRCs in cellular tomograms in more detail. Pericentriolar γ-TuRCs followed a broad radial distribution around the centrioles at approx. 85 nm center-to-center distance from the nearest microtubule triplet (MTT) B-tubule (Fig. 4b; 90% between 48.20 and 132.50 nm). Consistent with our own (Fig. 4c) and previously published high-resolution light microscopy data[20,21,39], as well as immunocytochemical stainings of centrosome sections[40], we also observed a highly condensed cluster of γ-TuRCs within the centriolar lumen, positioned at approx. 75 nm center-to-center distance from the B-tubule of the MTT (Fig. 4a, b; 90% between 62.80 and 82.70 nm). Thus, at least two separate pools of γ-TuRCs with defined spatial arrangements exist in the human centrosome during interphase.

**Defined supramolecular organization of centrosomal γ-TuRCs**

Aiming to characterize the distribution of centriole-lumenal γ-TuRCs along the length of centrioles, we subjected MTT segments to subtomogram analysis (Supplementary Fig. 14a) and obtained several distinct MTT structures with features that enabled us to assign them to

topological domains of the centriole (Supplementary Fig. 14b). Based on this assignment, we observed that lumenal γ-TuRCs preferentially localize to the central region of centrioles (Supplementary Fig. 14c), as also observed by ultrastructure expansion microscopy (U-ExM, Fig. 4c). γ-TuRCs in the centriole lumen were densely packed with a center-to-center distance of approx. 25 nm between each γ-TuRC and its closest neighbor (90% between 17.40 and 38.30 nm), while pericentriolar γ-TuRCs were spaced further apart and were more broadly distributed (approx. 50 nm distance, 90% between 24.80 and 90.60 nm) (Supplementary Fig. 14d).

We next analyzed the spatial organization of γ-TuRC particles relative to their centriolar environment, i.e., the MTTs forming the centriole wall. For each γ-TuRC particle, we identified the closest MTT segment contained in the tomograms and retrieved their exact positions and orientations relative to γ-TuRC particles by subtomogram analysis. When clustering γ-TuRC-MTT pair configurations according to their distance, relative position, and orientation (Supplementary Fig. 15a–c), we observed one main cluster encompassing 42% of all analyzed centrosomal γ-TuRC particles (63% of lumenal γ-TuRC particles), reflecting a preferred relative arrangement of γ-TuRC particles with respect to their closest MTT. In this arrangement, γ-TuRCs reside

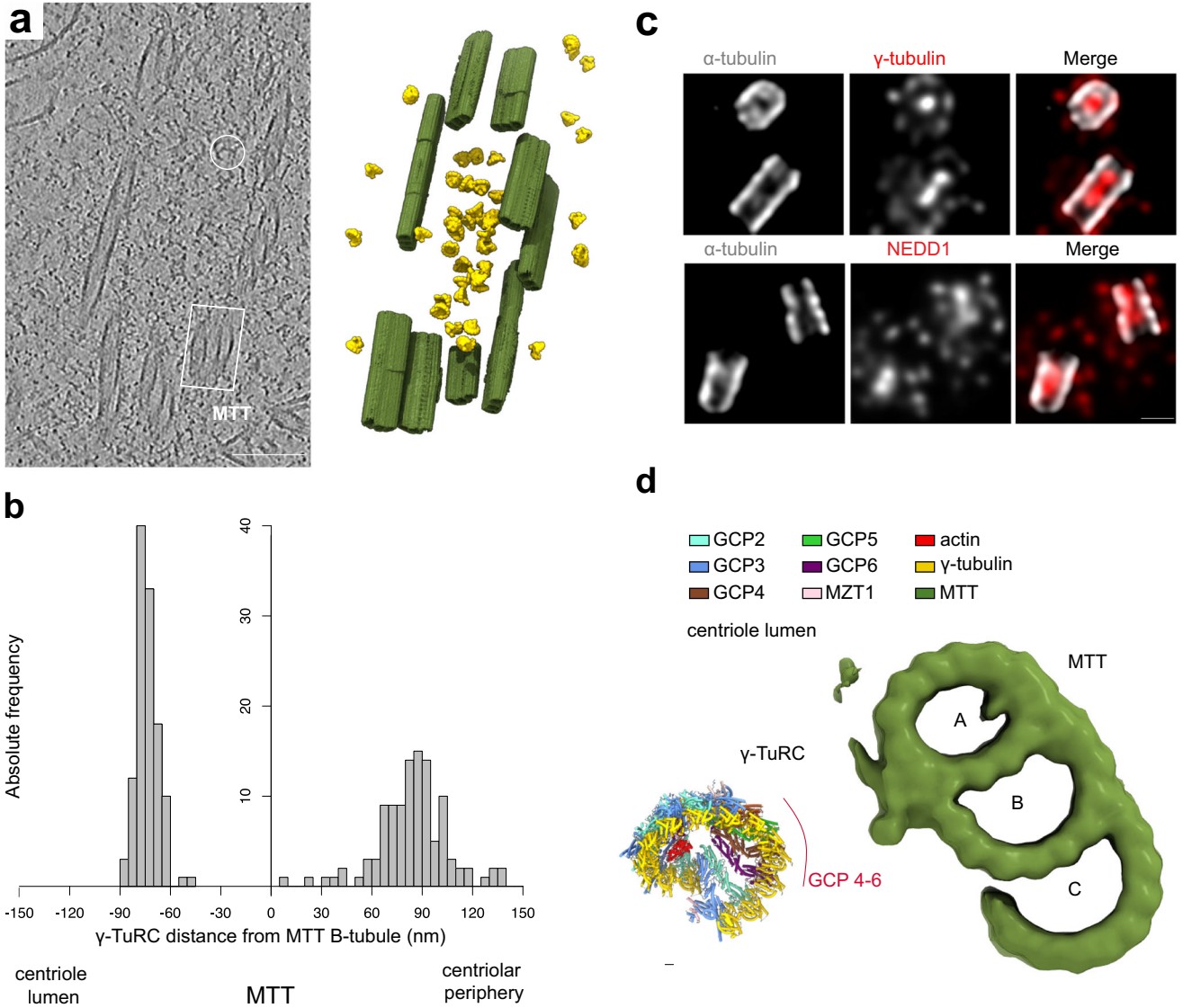

**Fig. 4 | Defined centrosomal organization of γ-TuRCs. a** Left: Tomographic slice depicting a centrosome in vitrified sections of cells (*n* = 14). Example γ-TuRCs are highlighted. Scale bar: 100 nm. Right: 3D rendering of the tomogram with MTTs depicted in green and localized γ-TuRCs in yellow. *N* = 3 biologically independent experiments. **b** Distribution of center-to-center distances between MTT B-tubules and luminal (negative distance) and pericentriolar (positive distance) γ-TuRCs, respectively, in *n* = 15 centrioles. **c** U-ExM of centrioles stained against α-tubulin (gray) and γ-tubulin or NEDD1 (red) illustrating γ-TuRC distribution in centrosomes. Scale bars: 200 nm. *N* = 3 biologically independent experiments. **d** Preferred spatial arrangement of lumenal γ-TuRCs with respect to the nearest MTT. A-, B-, and C-tubules of the MTT are indicated. Source data are provided as a Source Data file.

in the centriole lumen with spokes 9–12 oriented towards the centriolar wall (Fig. 4d), suggesting a defined but flexible physical interaction between MTTs and lumenal γ-TuRCs.

## POC5-Augmin interactions are required for centriole-lumenal γ-TuRC localization

We next analyzed how the defined spatial arrangement of γ-TuRCs in the centriolar lumen could be established. Recent studies have demonstrated that the localization of γ-TuRCs within the centriole lumen depends on Augmin, which in turn was reported to require the centriole inner scaffold protein POC5 for localization[21]. This makes Augmin the prime candidate for positioning γ-TuRCs inside centrioles via γ-TuRC-Augmin-POC5 interactions[41]. Augmin is an elongated complex with pronounced flexibility (Fig. 5a), which prevents its detection in cellular tomograms[42,43]. We therefore pursued alternative strategies to locate Augmin's approx. 45 nm-separated opposing ends within the centriolar lumen.

To explore whether Augmin could be recruited to the centriole lumen by directly interacting with POC5, we used AlphaFold2 and predicted complexes between POC5 and Augmin's structural elements (TII subcomplex: HAUS2,6,7,8 and TIII subcomplex: HAUS1,3,4,5)[43] (Fig. 5b and Supplementary Fig. 16a–d). AlphaFold2 predicted an interaction of POC5 with Augmin's TII subunits HAUS2 and HAUS7 in the N-terminal half of the clamp-shaped Augmin TII (N-clamp), but not Augmin TIII. To experimentally validate this prediction, we immunoprecipitated recombinant Augmin TII N-clamp constructs and POC5-HA expressed in HEK293 cells. Consistent with AlphaFold2 predictions, Augmin TII N-clamp co-immunoprecipitated with wild-type POC5, while a POC5 mutant (POC5^mut) designed to disrupt the predicted POC5-Augmin interface (Supplementary Fig. 16a) showed a 3-fold reduced interaction with Augmin (Fig. 5c, d). In addition, pulldown assays using recombinant Augmin TII N-clamp and a previously described recombinant POC5/centrin 2 complex[44] support the direct binding

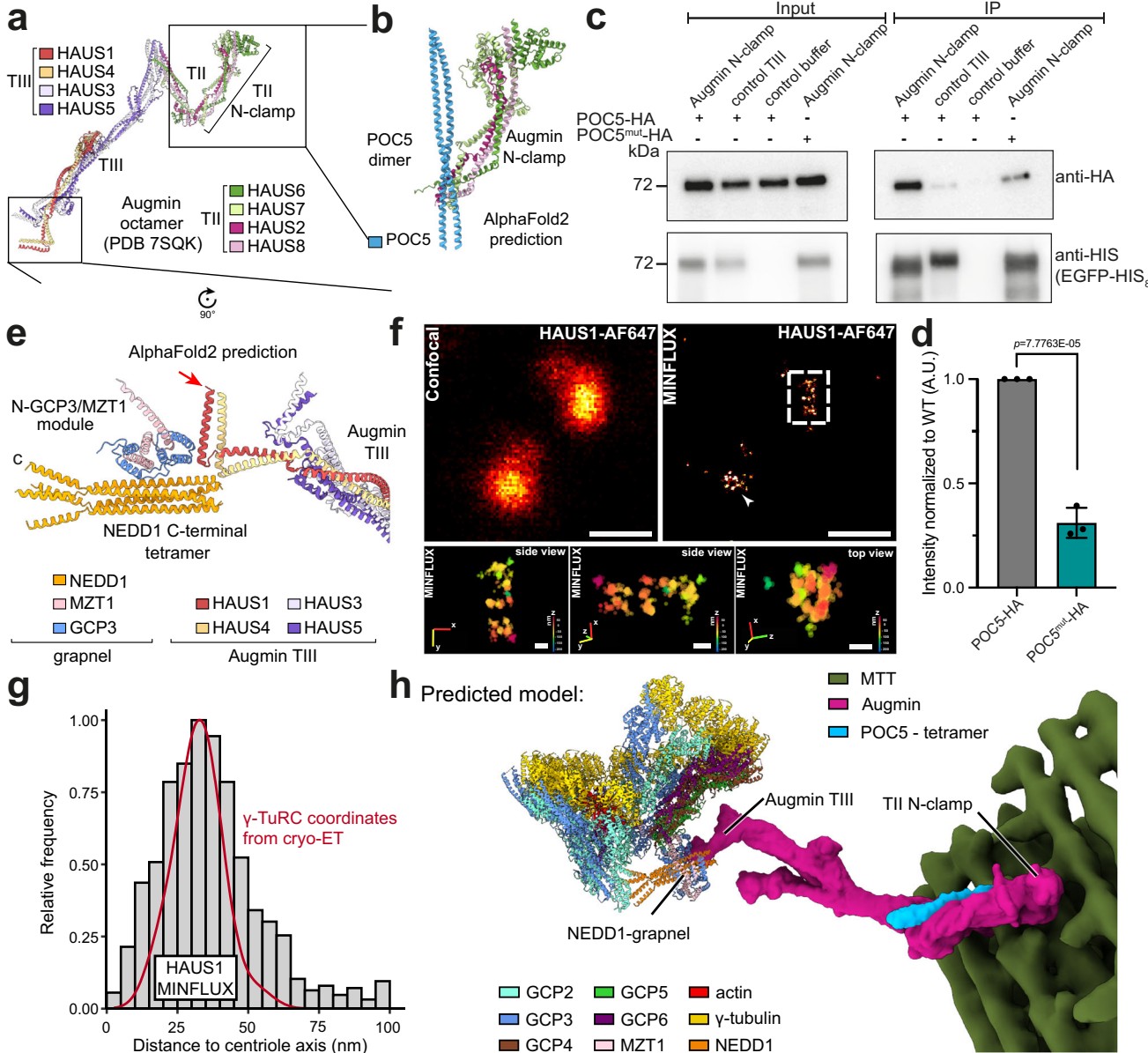

**Fig. 5 | POC5-Augmin-NEDD1 interactions tether γ-TuRCs to the inner centriolar wall. a** Structure of the Augmin complex with subunits colored as indicated (PDB: 7SQK). **b** AlphaFold2 prediction of the interaction between POC5 and the Augmin TII N-clamp. **c** Representative immunoprecipitation results using HEK293 cell lysate with overexpressed POC5-HA or POC5$^{mut}$-HA. EGFP pull down of HAUS7-EGFP-8His (TII) and HAUS1-EGFP-8His (TIII) proteins was performed to enrich Augmin subcomplexes. The 8His tag was used for the detection of HAUS proteins. Buffer only and Augmin TIII was used as the negative controls. HA detection was used to compare the bound POC5$^{WT}$ and POC5$^{mut}$. **d** Quantification (mean ± SD) of band intensities from the immunoprecipitation experiment in (**c**) normalized to POC5$^{WT}$, based on $N = 3$ independent experiments. Statistical analysis was performed using an unpaired two-tailed $t$ test. **e** AlphaFold2 prediction of the interaction between the NEDD1 grapnel and Augmin TIII, Coloring as indicated. The

location of the C-terminal SNAP-S tag on HAUS1 is indicated by a red arrow. The C-terminus of NEDD1 is indicated by the letter c. **f** Representative confocal (upper left) and MINFLUX images (upper right) after endogenous tagging of HAUS1 with a C-terminal SNAP-S tag labeled via BG-AF647 (see red arrow in (**e**) and Supplementary Fig. 20). Scale bars: 5 μm. The lower image panels show zoomed 3D renderings of the HAUS1-SNAP-S BGAF647 organization highlighted by the dashed white square. The structure is rotated in each image around 90° to show side and top views. All scale bars: 50 nm. $N = 3$ biologically independent experiments. **g** Centriole radial distribution of HAUS1-SNAP-S signal from MINFLUX ($n = 17$ centrosomes) (gray) relative to centriole-lumenal γ-TuRC coordinates from cellular cryo-ET (red) ($n = 5$ centrioles). **h** Integrated structural model (predicted) for γ-TuRC recruitment to the centriole lumen by POC5 and Augmin. Coloring as indicated. Source data are provided as a Source Data file.

of Augmin to the identified interaction site in POC5 (Supplementary Fig. 17a–d).

In order to confirm the role of the POC5-Augmin interaction for centriole-lumenal γ-TuRC localization in cells, we constructed stable cell lines by expressing either HA-tagged *POC5^mut* or *POC5* in *POC5^-/-* RPE1 cells. In all experiments, *POC5* cells behaved very similarly to RPE1 control cells (Supplementary Fig. 18a–f), indicating that the complementation was successful. We confirmed that *POC5* and *POC5^mut*

were expressed at comparable levels (Supplementary Fig. 18a). In addition, indirect anti-HA immunofluorescence analysis showed similar signal intensity of POC5 and POC5$^{mut}$ at centrioles (Supplementary Fig. 18b, c). In *POC5* and *POC5^mut* cells, the inner centriole marker POC5[44] (Supplementary Fig. 18g) co-localized with γ-tubulin in the vast majority of cells (Supplementary Fig. 18d), confirming that γ-tubulin is a bona fide marker for centrosomes in this analysis. Using γ-tubulin, we observed two dot-like signals in most control, *POC5* and *POC5^mut* cells

(Supplementary Fig. 18b, e), indicating that centrosomes were not amplified or fragmented in POC5$^{mut}$ cells and behaved as in control and POC5 cells. This was different for POC5$^{-/-}$ cells, which showed an increase in cells with < 2 and > 2 γ-tubulin signals, consistent with the known function of POC5 in centriole stabilization[44]. U-ExM analysis further indicated that POC5$^{mut}$ centrioles did not exhibit the length defect seen in POC5$^{-/-}$ cells, nor were they as frequently broken as in POC5$^{-/-}$ cells (Supplementary Fig. 18h, i). Together, these findings suggest that the stabilizing inner scaffold remains functional in POC5$^{mut}$ centrioles. However, consistent with the direct POC5-Augmin TII N-clamp interaction observed in vitro (Supplementary Fig. 17), U-ExM and indirect immunofluorescence indicated that γ-tubulin and Augmin HAUS4 signal was strongly reduced in POC5$^{mut}$ centrioles, very similar to POC5$^{-/-}$ centrioles (Supplementary Fig. 18f, j, k). This clearly indicates that Augmin is anchored within the centriole lumen via POC5-TII interaction, positioning the TII N-clamp in close proximity to the centriolar wall.

### Augmin tethers γ-TuRCs in the centriole lumen

It has been shown that Augmin TIII interacts with the C-terminus of NEDD1 to bind to the γ-TuRC[23]. Thus, to investigate the structural basis of the Augmin TIII-γ-TuRC interactions in more detail, we used AlphaFold2 and predicted structures of Augmin TIII together with tetrameric NEDD1$^C$ and one N-GCP3/MZT1 module. AlphaFold2 consistently predicted an interaction in which the C-terminal helices of HAUS1 and HAUS4 bind to the C-terminal NEDD1 helix and an associated N-GCP3/MZT1 module (Fig. 5e and Supplementary Fig. 16e–g). To support the AlphaFold2 prediction, we recombinantly expressed and purified Augmin TIII from insect cells[43] and tested binding to the purified NEDD1-N-GCP3/MZT1 complex in vitro. We observed a pull-down of NEDD1-N-GCP3/MZT1 dependent on Augmin TIII (Supplementary Fig. 19a–d). Notably, the interaction between TIII and NEDD1-N-GCP3/MZT1 was strongly increased by pre-incubation with cyclin-dependent kinase 1 (CDK1) and polo-like kinase 1 (PLK1) kinases (Supplementary Fig. 19e), consistent with previous findings[23]. This indicates that Augmin TIII directly interacts with NEDD1-N-GCP3/MZT1 to recruit the γ-TuRC.

Next, to determine the localization of the predicted Augmin-γ-TuRC interface within the centriolar lumen of RPE1 cells, we employed MINFLUX nanoscopy, which has a resolution in the low nm range[45]. For this, we genetically integrated a SNAP-Strep double tag (referred to as SNAP-S) at the C-terminus of HAUS1, which is located on Augmin TIII[43], in close proximity to the predicted interaction site with NEDD1 (Fig. 5e). This design minimizes the distance between the fluorophore and the target protein, a critical requirement for achieving super-resolution microscopy. The cell line was validated by chromosomal PCR (Supplementary Fig. 20b) and sequencing of the PCR products (Supplementary Fig. 20c). HAUS1-SNAP-S was expressed at a similar level as endogenous HAUS1 and, as expected, was detected at higher molecular weight as compared to wild-type HAUS1 by immunoblotting (Supplementary Fig. 20d).

To validate the unaltered function of HAUS1-SNAP-S, we confirmed by confocal microscopy and STED microscopy that HAUS1-SNAP-S is associated with spindle MTs (Supplementary Fig. 20e). In addition, since the loss of Augmin function impairs mitotic progression[46], we tested the cell cycle distribution of these cells. The SNAP-S-tag on endogenous HAUS1 did not cause accumulation of cells in mitosis, as indicated by the similar mitotic index compared to parental cells (Supplementary Fig. 20f), the similar behavior of the cell cycle marker proteins CDK1, Aurora A and phospho-histone H3 (pH3) (Supplementary Fig. 20g–j) and the same percentage of KI67-positive cells in the HAUS1 and HAUS1-SNAP-S populations (Supplementary Fig. 20k). Cumulatively, this indicates that HAUS1-SNAP-S is fully functional.

MINFLUX nanoscopy of HAUS1-SNAP-S labeled with SNAP-specific BG-AF647 resolved its localization at centrioles (Fig. 5f), which were detected using POC5 antibodies in confocal microscopy (Supplementary Fig. 21). Rendering HAUS1-SNAP-BGAF647 in 3D revealed a cylindrical hollow organization (Fig. 5f, insets). Normalizing the spatial coordinates from 17 HAUS1-SNAP-S structures (averaged median localization precision $\sigma_{x,y}$ ~ 6 nm and $\sigma_z$ ~ 5 nm) to the central axis, we observed a radial distribution at approx. 35 nm around the central axis (95 percentile of 11.50 and 68.80 nm, respectively) (Fig. 5g and Supplementary Table 5). This accurately matches the distribution of lumenal γ-TuRCs around the centriolar axis we observed by cryo-electron tomography (Fig. 5g), in line with a direct interaction between Augmin TIII and γ-TuRC/NEDD1 within the centriole lumen, as predicted by AlphaFold2 and confirmed by in vitro binding assays.

When mapping the predicted Augmin-NEDD1 interaction into the preferred γ-TuRC-MTT configuration observed for lumenal γ-TuRCs (Fig. 4d), the Augmin TII N-clamp, including the identified POC5 binding site, is positioned in close proximity to the MTT wall, providing a structural model for how γ-TuRCs could be tethered within the centriole lumen in a defined spatial pattern, mediated through the centriole-radial arrangement of Augmin (Fig. 5h).

### Augmin and γ-TuRC are released from the centriole lumen upon mitotic onset

To characterize the properties of centriole-lumenal Augmin recruitment (Supplementary Fig. 20l), we examined the dynamics of Augmin during different phases of the cell cycle. Prolonged FRAP experiments over 30 min with BG-AF647-labeled HAUS1-SNAP-S indicated that HAUS1 was stably associated with centrioles in interphase cells (Supplementary Fig. 20m).

Next, we investigated how mitosis affected the intensity and distribution of Augmin, γ-TuRC components, and POC5 inside centrioles. For this analysis, we arrested cells in the prometaphase of mitosis with the Eg5 inhibitor STLC[47] (Supplementary Fig. 22a) and performed U-ExM for HAUS4, NEDD1, GCP4, and POC5 (Supplementary Fig. 22b). Quantification of the signal intensities inside centrioles indicated a 50% decrease compared to interphase for HAUS4, NEDD1, and GCP4 (Supplementary Fig. 22c). Consistent with the reduction in signal intensity, we observed that in mitosis, the spatial coverage of HAUS4, GCP4, and NEDD1 inside centrioles was reduced by 50, 85 and 80% compared to interphase centrioles (Supplementary Fig. 22d). In contrast, the centriolar inner scaffold protein POC5 showed a constant centriolar signal in interphase and mitosis (Supplementary Fig. 22b–d), indicating that its centriole localization is independent of the cell cycle.

To exclude the possibility that STLC-induced mitotic arrest caused the release of Augmin from centrioles, we confirmed the cell-cycle-dependent decrease of centriole-lumenal Augmin in mitotic cells using unsynchronized cells (Fig. 6a–c and Supplementary Fig. 23) with conventional fluorescence microscopy. While the signal intensity of γ-TuRC (GCP4 and NEDD1 as markers) inside the centriole lumen decreased during mitosis, as indicated by U-ExM (Supplementary Fig. 22b, c), the overall centrosomal γ-tubulin signal increased in prophase in this experiment due to mitotic recruitment of γ-TuRCs to the PCM[48].

To test whether the kinase PLK1, which has multiple cell-cycle dependent regulatory functions at centrosomes[49], is involved in regulating Augmin-γ-TuRC release from the centriolar lumen, PLK1 activity was inhibited with BI2536 in the presence of STLC[50]. Consistent with published data[51], mitotic γ-tubulin recruitment to the PCM was blocked after PLK1 inhibition by BI2536 (Fig. 6a–c; "STLC + BI2536"). Interestingly, under the same conditions, BI2536 partially blocked the release of Augmin from centrioles in mitosis[50] (Fig. 6a–c), suggesting a role for this kinase in the regulation of Augmin inside centrioles.

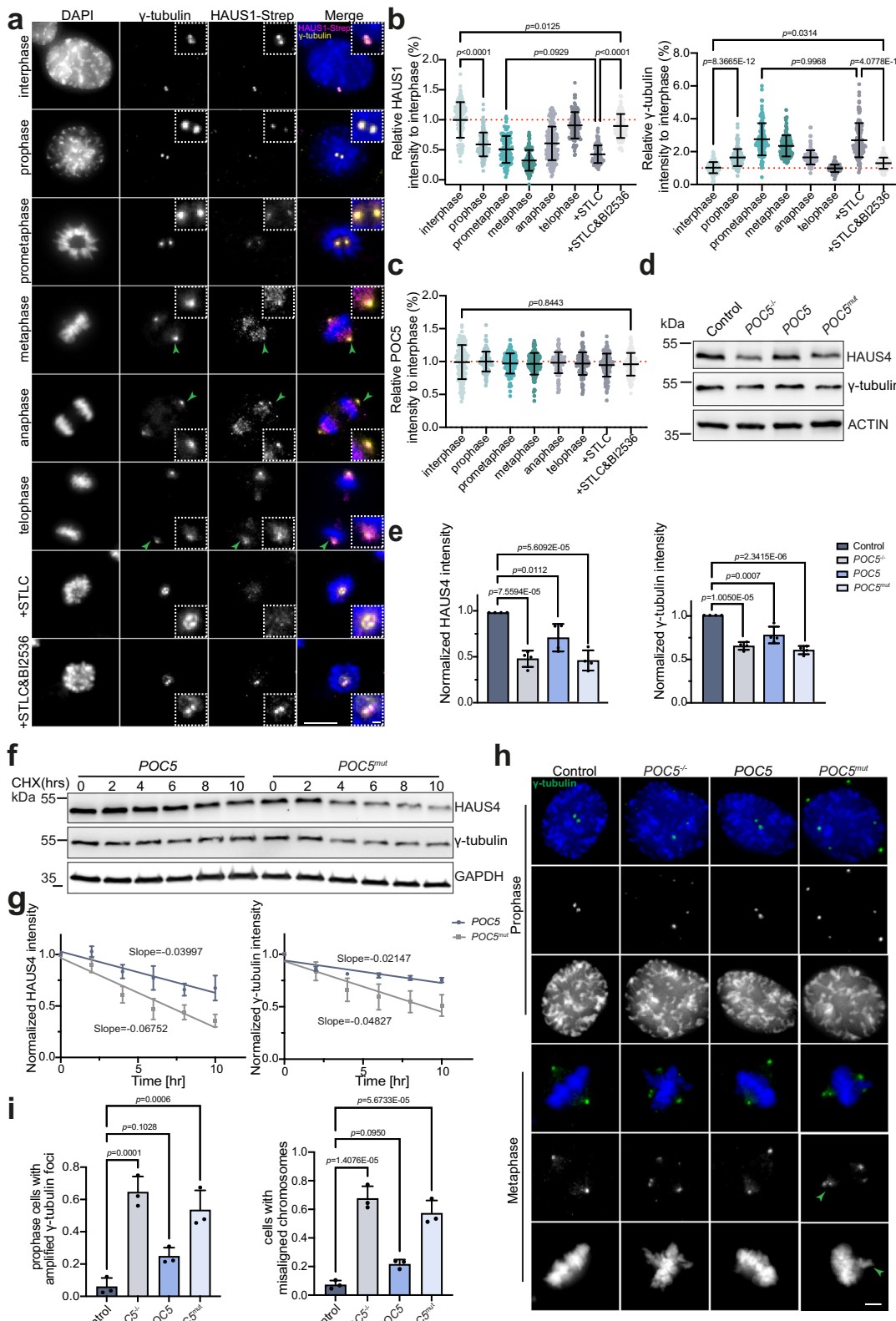

## Augmin and γ-TuRC are stabilized in interphase centrioles by POC5-Augmin interaction

Next, we aimed to understand the function of centriole-lumenal Augmin and γ-TuRC localization. In immunoblots of total cell lysate, which contains centrosomes and the majority of cytosolic γ-tubulin, we observed a reduction in the cellular levels of HAUS4 and γ-tubulin in both *POC5⁻/⁻* and *POC5^mut* cells compared to wild-type control and

*POC5^WT* cells (Fig. 6d, e). This indicates that Augmin and γ-TuRC might be stabilized by localization inside centrioles, which is disturbed in *POC5⁻/⁻* and *POC5^mut* cells. To test this notion further, we performed cycloheximide (CHX) chase experiments. In these experiments, CHX inhibits de novo protein synthesis on ribosomes, allowing us to monitor the degradation of pre-existing proteins. These experiments revealed 3- and 2.6-fold higher stability of γ-tubulin and HAUS4,

**Fig. 6 | Augmin and γ-TuRC are stabilized in the interphase centriole by POC5-Augmin interaction and released upon mitotic onset. a** Representative immunofluorescence images of RPE1 cells in different cell cycle stages stained against γ-tubulin (yellow) and HAUS1-SNAP-S with anti-Strep antibodies (HAUS1-Strep, magenta). DNA was visualized with DAPI (blue). Green arrows indicate the centrioles shown in the close-up view. **b, c** Quantification of fluorescence intensity at centrosomes for the indicated proteins shown in a) and Supplementary Fig. 23. *n* = 125 (interphase), 132 (prophase), 121 (prometaphase), 125 (metaphase), 162 (anaphase), 128 (telophase), 126 (+ STLC), and 117 (+ STLC&BI2536) centrioles were analyzed for HAUS4 intensity. *n* = 108 (interphase), 127 (prophase), 121 (prometaphase), 126 (metaphase), 121 (anaphase), 122 (telophase), 120 (+ STLC), and 120 (+ STLC&BI2536) centrioles were analyzed for γ-tubulin intensity. *n* = 127 (interphase), 118 (prophase), 130 (prometaphase), 123 (metaphase), 121 (anaphase), 127 (telophase), 131 (+ STLC), and 119 (+ STLC&BI2536) centrioles were analyzed for POC5 intensity. **d** Immunoblot showing protein levels of HAUS4 and γ-tubulin in wild-type control, *POC5[-/-]*, *POC5* and *POC5[mut]* cells. **e** Quantification of relative HAUS4 and γ-tubulin levels in (**d**) (normalized to actin). **f** Representative immunoblots of *POC5[-/-]* cells expressing *POC5* or *POC5[mut]*, treated with 50 μg/ml CHX for the indicated durations, using anti-HAUS4, anti-γ-tubulin and anti-GAPDH (loading control) antibodies. **g**) Quantification of panel **f**). Relative protein levels were normalized to protein levels at time 0. Linear regression was performed; R$^2$ values are provided in the Source Data. **h** Representative images of wild-type control, *POC5[-/-]*, *POC5*, and *POC5[mut]* cells in prophase and metaphase stained against the centrosomal marker γ-tubulin (green). DNA was visualized with DAPI. Green arrows indicate misaligned chromosomes. **i** Quantification of cells with amplified γ-tubulin foci and misaligned chromosomes for h). *N* = 139 (control), 136 (*POC5[-/-]*), 129 (*POC5*), and 137 (*POC5[mut]*) cells were analyzed for amplified γ-tubulin foci. *n* = 135 (control), 133 (*POC5[-/-]*), 125 (*POC5*) and 131 (*POC5[mut]*) cells were analyzed for misaligned chromosomes. (**a**), (**h**), scale bars: 10 μm, magnification scale bars: 1 μm; in (**b, c, e, i**), data are presented as mean ± SD, and all statistics were derived from ordinary one-way ANOVA analysis of *N* = 3 biologically independent experiments. Source data are provided as a Source Data file.

respectively, in *POC5[WT]* and control cells, where both proteins were localized within centrioles, compared to *POC5[-/-]* and *POC5[mut]* cells (Fig. 6f, g).

Finally, we investigated how impaired Augmin and γ-TuRC localization inside centrioles in interphase subsequently affects mitosis, as the amounts of Augmin and γ-TuRC to be released from the centriole lumen at the beginning of mitosis are strongly reduced. We observed multiple γ-tubulin signals in prophase and an increase in misaligned chromosomes at the metaphase plate in *POC5[-/-]* and *POC5[mut]* cells, while such defects were not observed in control and *POC5* cells (Fig. 6h, i).

Based on these findings, we propose that Augmin and γ-TuRC are stored in the centriole lumen during interphase for stabilization. Both complexes are then released from the POC5 scaffold at the onset of mitosis in response to increased PLK1 activity to promote the formation of a functional mitotic spindle, most likely by boosting MT nucleation activity at centrosomes.

## Discussion

While data on γ-TuRCs after biochemical isolation have become available over the past years, our work provides structural insights into the native γ-TuRC structure, conformational landscape, and supramolecular organization in centrosomes. By imaging γ-TuRCs in a physiological context, we were able to capture the structure of the established but previously not resolved essential γ-TuRC interactor NEDD1. NEDD1 is a general adapter and targeting factor of the vertebrate γ-TuRC and promotes interaction with important binding partners, including Augmin and CEP192[23,52]. The NEDD1-γ-TuRC interaction is bimodal: First, the tetramerized C-terminus of NEDD1 binds four N-GCP3/MZT1 modules to form a grapnel-like structure, a step that, in theory, would highly flexibly tether NEDD1 to the γ-TuRC. In the second step, the NEDD1 grapnel rigidly docks onto the narrow end of the γ-TuRC cone by binding to GCP5- and GCP6-specific elements, stabilizing the interaction in a unique orientation. While rigid docking of NEDD1 may have a central functional role during MT branching where the γ-TuRC has to be pre-oriented by NEDD1-Augmin interactions for establishing correct MT growth direction, flexible tethering of NEDD1 to γ-TuRC may be relevant for alternative functions or interaction partners. Notably, the release of N-GCP3/MZT1 modules from the outer surface of GCP3 GRIP2 domains, where they have been resolved in native and recombinant assembly intermediates of the γ-TuRC[7,8], may serve as a mechanism to prevent premature NEDD1 binding or recruitment of other binding partners via NEDD1 before full γ-TuRC assembly. The length of flexible segments linking the N-GCP/MZT1 modules to the γ-TuRC core structure would allow N-GCP/MZT1 modules to reach adjacent γ-TuRCs in the centriole lumen, possibly generating a flexible network of interconnected γ-TuRCs that may assist retention of γ-TuRCs in the centriole lumen or the PCM. The structure of the NEDD1 grapnel with pseudo-fourfold symmetry may serve as a scaffold to allow specific recruitment of binding partners in different stoichiometries and potentially enabling them to simultaneously interact. As the NEDD1 grapnel bridges the two ends of the γ-TuRC that move relative to each other during conformational changes, such as partial and full closure as observed after MT nucleation[53–55], the hinge between the short and long NEDD1[C] helices may allow NEDD1 to bind to different conformations of the γ-TuRC or potentially vice versa, to participate in conformational regulation of the γ-TuRC under specific conditions. Notably, while the lower dimer of NEDD1[651-670] appears to only weakly interact with spoke 1/2, stable contacts might be established upon closure of the γ-TuRC before or during MT nucleation[53–55].

In the context of purified centrosomes, we could distinguish two different γ-TuRC conformations that were highly correlated with mutually exclusive patterns of CDK5RAP2 CM1 module binding on the outside of the γ-TuRC. While binding of one CM1 module to GCP2(13) was associated with an asymmetric open conformation, concomitant binding of four CM1 modules to all GCP2 subunits in the first half of the ring (position 1,3,5,7) was linked to an allosteric conformational change that transitions the first half of the γ-TuRC into a closed and potentially more active conformation, providing a possible mechanism for how CDK5RAP2 CM1 modules may promote and regulate MT nucleation activity of the γ-TuRC. A similar conformational change could be observed for purified γ-TuRCs in vitro when subjected to high concentrations of the isolated recombinant CDK5RAP2 CM1 motif[8,9]. However, in contrast to the striking mutually exclusive patterns of endogenous CDK5RAP2 CM1 module binding observed in our in situ study, the purified CM1 peptide indiscriminately bound to all five GCP2 subunits in the in vitro setting[8,9]. This suggests that other interaction partners or post-translational modifications present only in the context of the centrosome participate in and finetune conformational regulation of the γ-TuRC by CDK5RAP2. This is also consistent with the absence of γ-TuRC-associated MTs in the PCM of cellular centrosomes, which suggests that additional regulatory layers beyond CDK5RAP2 CM1 motif binding may impact the MT nucleation activity of γ-TuRCs in the cellular context.

Our structural data on γ-TuRCs embedded into the native PCM indicate that CDK5RAP2 and NEDD1 can simultaneously bind to γ-TuRCs at centrosomes and do not define distinct pools of γ-TuRCs, as was previously suggested in studies using purified γ-TuRCs[35,56]. This discrepancy may stem from the limitations of biochemical purifications, in which CDK5RAP2 may have dissociated during γ-TuRC purification via NEDD1, and vice versa.

Analyzing the structure and centrosomal distribution of γ-TuRCs in human cells, we observed two spatially separated pools of γ-TuRCs, as observed previously[20,21]. While the ordered distribution of pericentriolar γ-TuRCs is most likely established by CDK5RAP2 and CEP192, the latter of which binds to NEDD1, cooperation between the inner

scaffold protein POC5 and Augmin coordinates γ-TuRCs in the centriole lumen[19–21,40,52,57]. Importantly, previous studies using high-resolution light microscopy have shown that γ-TuRCs are detectable in the centriole lumen only during the later stages of centriole assembly[20,21]. This suggests that most of the centrioles analyzed in our study were fully mature.

Combining our structural data on how NEDD1 docks to the γ-TuRC with AlphaFold2-based structure predictions of the NEDD1-Augmin interaction provides a structural model of how γ-TuRCs are organized within the centriole lumen (Fig. 5h). However, it may also provide hints towards how Augmin recruits and orients γ-TuRCs on the surface of pre-existing MTs during the formation of branched MT networks during mitotic spindle formation. Notably, Augmin binding to all four units of the tetrameric NEDD1/N-GCP3/MZT1 grapnel would be sterically possible without clashes between Augmin and other Augmin copies or the γ-TuRC. However, whether all of the four binding sites are sampled in practice and how this could impact the coordination of the γ-TuRC during MT branching, remains to be investigated.

Strikingly, none of the γ-TuRCs identified in the PCM of cellular centrosomes was associated with an MT or had a partially or fully closed conformation as observed in MT-bound γ-TuRC structures[53–55]. This may reflect the overall reduced MT nucleation activity during interphase and is consistent with studies reporting that MTs formed at centrosomes during interphase are readily released from γ-TuRC after nucleation[58]. It also underlines that the open conformation of γ-TuRCs observed after biochemical purification is indeed a physiological state and most likely linked to the regulation of the MT nucleation activity.

Augmin and the γ-TuRC are likely shielded from degradation in the centriole lumen during interphase, as indicated by CHX chase experiments. However, we cannot entirely exclude the possibility that POC5 may also contribute to stabilizing γ-TuRC and Augmin outside centrioles. Activation of the centriolar pool of Augmin at the onset of mitosis coincides with the release of Augmin from importin upon nuclear envelope breakdown in proximity to chromosomes[59,60], where the Ran activator RCC1 resides[61]. In contrast, the centriolar pool of Augmin could function in the vicinity of the spindle poles. These two mechanisms of Augmin activation likely work synergistically during mitotic spindle formation to generate a large pool of active Augmin/γ-TuRC within a short time frame and promote a strong increase in MT nucleation capacity. This notion is consistent with the metaphase chromosome misalignment in POC5[mut] cells. As centriole fragmentation was not observed in POC5[mut] cells with an intact inner POC5 scaffold, this defect is likely attributable to reduced levels of γ-tubulin and Augmin and the strongly reduced amounts of both proteins to be released from the centriole lumen at the beginning of mitosis. However, we cannot entirely rule out the possibility that subtle centriole aberrations also contribute to this phenotype.

In summary, our findings demonstrate that the γ-TuRC at centrosomes engages in defined modular interaction patterns with different centrosomal proteins. This interaction network contributes to γ-TuRC localization and likely facilitates structural transitions of the γ-TuRC into various states with specific MT nucleation properties. Deciphering the underlying code will be essential to understand the regulation of MT nucleation in cells.

## Methods

### Plasmid construction

pRetroX-Tet3G and pVSVG plasmids (Retro-X Tet-On 3 G Inducible Expression System, Clontech) were used to generate HEK293 cells and RPE1 cells with Tet-On 3 G System. HA-tagged POC5 was amplified via PCR and cloned into pRetroX-TRE3G. Similarly, HA-tagged human NEDD1 was cloned into pRetroX-Tet3G. All PCR reactions were carried out using Q5 DNA polymerase (NEB) with primers listed in Supplementary Table 7. Amplified plasmids and DNA inserts were combined using the NEBuilder Hifi assembly master mix (NEB). To generate the

POC5[mut]-HA (S224E, R227A, K228A, V231A, S234K, L235A) variant, the wild-type plasmid was amplified and used for NEBuilder reaction together with a short DNA fragment (5′-GGAGCTGCAAAAAACCTTT GAAATCGAGATTGGGGCAGCAGATGAG GCGATTTCTAAGGCGTCTC ATGCCATAGGCAAGCAAAAGGAAAAG-3′) including the mutated sequence. For the HA-NEDD1[VVI] and HA- NEDD1[LLM] mutants, wild-type plasmids were amplified with mutation primers listed in Supplementary Table 7, and afterwards incubated with Dpn1 (1 hr, 37 °C, NEB) and used for transformation.

DNA fragments of human TII N-clamp HAUS2 (codons 1 to 118), HAUS6 (1 to 269), HAUS7 (1 to 299), and HAUS8 (1 to 248) were cloned into two separate E. coli expression plasmids based on the pETDUET-1 and pET26b expression vectors. For each cloning step, the vectors were linearized and amplified via PCR. In two subsequent cloning steps, HAUS8 and HAUS6 were inserted into pETDUET-1 adding a C-terminal 8xHis tag to the HAUS8 sequence. Similarly, HAUS2 and HAUS7 were first cloned into pETDUET-1, and afterward, the expression cassette was cloned into pET26b with a TEV-EGFP-8xHis cleavable tag in frame with the C-terminus of HAUS7 based on a pET26 backbone described in ref. 43. The N-clamp HAUS8 truncation, HAUS8Δ [1-138], referred to as Augmin N-clamp was generated via PCR amplification of the pETDUET-1 construct. After the PCR reaction, Dpn1 restriction enzyme (NEB) was added to the mixture. After incubation for 1 hr at 37 °C, the resulting DNA was used for transformation in DH5α E. coli as for the other constructs.

For the expression of X. laevis NEDD1 constructs in insect cells, the MultiBac system (Geneva Biotech) was utilized. X. laevis NEDD1 cDNA (NM_001094388.1) was amplified via PCR and cloned into a pACEBac1 vector containing an N-terminal 2 × FLAG-TEV tag. For the NEDD1-N-GCP3/MZT1 construct, MZT1 (Q5U4M5 · MZT1_XENLA) with an N-terminal ALFA-tag and GCP3 (1–240, O73787 · GCP3_XENLA) with an N-terminal linker and 3C protease site were ordered as gene fragments (Twist Biosciences). These fragments were synthesized with overhangs compatible with the MultiBac constructs. In the case of the NEDD1-GCP3 fusion construct, the 2 × FLAG-TEV-NEDD1 was used as the template. Subsequently, the expression cassettes of the NEDD1-N-GCP3 fusion and MZT1 were combined using primers as described previously[7] and employed for bacmid production.

To endogenously tag HAUS1 with SNAP-S, a gene fragment was designed containing a SNAP-tag sequence, a Strep-tag sequence, and 380 bp homology arms overlaid with the C-terminus of HAUS1 gene (Integrated DNA Technologies), amplified via PCR and inserted into a backbone that only contains Ori and AmpR. Guide RNA against the C-terminus of HAUS1 (5′-5′-TACAAGAAGAGTAGACATGA-3′) was annealed and cloned into pSpCas9-2 A-GFP (px458, Addgene, #48138) vector[62], which was linearized with BbsI.

### Antibodies

Primary antibodies used in this study were: Mouse α-tubulin (Proteintech 660311-1-Ig; ExM: 1:500), rabbit α-tubulin (Proteintech 11224-1-AP; ExM: 1:500), mouse γ-tubulin (abcam Ab27074; IF: 1:500; ExM: 1:500; WB: 1:1000), rat HA (Merck 11867423001; IF: 1:1000; ExM: 1:1000; WB: 1:1000), rabbit POC5 (Bethyl A303-341A-T; IF 1:2000; ExM: 1:2000), rabbit HAUS4 (Proteintech 20104-1-AP; IF: 1:500; ExM: 1:500; WB: 1:1000), mouse ACTIN (WB 1:1000), rabbit CDK5RAP2 (Merck 2952319; IF: 1:500; ExM: 1:500), mouse NEDD1(Santa Cruz sc-100961; IF: 1:500; ExM: 1:500), mouse GCP4 (Santa Cruz sc-271876; ExM: 1:500), mouse HIS (QIAGEN 169033106; WB: 1:1000), rabbit GAPDH (Cell Signaling Technology 14C10; WB: 1:1000), rabbit GCP3 (Proteintech 15719-1-AP; WB: 1:1000), rabbit HAUS1 (ATLAS ANTIBODIES HPA040601; WB: 1:1000), mouse GCP5 (Invitrogen PA5-83495; WB: 1:1000), rabbit Aurora A (Cell Signaling Technology 1G4; WB: 1:1000), rabbit CDK1 (Proteintech 10762-1-AP; WB: 1:1000), rabbit pH3 (Cell Signaling Technology D2C8 WB: 1:1000), mouse KI67 (Santa Cruz Biotechnology sc-23900; WB: 1:1000), rabbit STREP (Abcam EPR28119-43; WB: 1:1000;

IF: 1:500; ExM 1:200). Secondary antibodies used in this study were: mouse Alexa Fluor Plus 488 (Thermo Fisher Scientific A-11001; 1:500), rabbit Alexa Flour Plus 488 (Thermo Fisher Scientific A-11008; 1:500), mouse Alexa Flour Plus 555 (Thermo Fisher Scientific A32727; 1:500), rabbit Alexa Flour Plus 555 (Thermo Fisher Scientific A32732; 1:500), mouse Alexa Flour Plus 647 (Thermo Fisher Scientific A32728; 1:500), rabbit Alexa Flour Plus 647 (Thermo Fisher Scientific A32733; 1:500), mouse Abberior STAR 635 P (Abberior ST635P-1001-500UG; 1:500), rabbit Abberior STAR 635P (Abberior ST635P-1002-500UG; 1:500). Anti-ALFA HRP-conjugated anti-alpaca (Nanotag N1505HRP, 1:5000). Secondary HRP-conjugated antibodies used in this study for immunoblotting are HRP anti-mouse (Jackson 715-035-151; 1:5000), HRP anti-rabbit (Jackson 711-035-152; 1:5000) and HRP anti-rat (Jackson 712-035-153; 1:5000), mouse True blot (Rockland 18-8817-33; WB: 1:1000). HRP anti-alpaca (Nanotag N1505HRP, 1:5000).

## Cell culture and treatment

KE37 cells were cultured in RPMI 1640 medium (Gibco) supplemented with 10% fetal bovine serum (FBS). Human telomerase-immortalized RPE1 cells (hTERT-RPE1 *TP53*[-/-])[7,63,64] were used as wild-type control for *POC5*[-/-], *HAUS1-SNAP-S* and in Supplementary Fig. 23. *POC5*[-/-] and *HAUS1-SNAP-S* were constructed in RPE1 *TP53*[-/-] cells. *CDK5RAP2*[-/-] cells[14] were *TP53*[+/+], and RPE1 *TP53*[+/+] cells were used as control. Human embryonic kidney 293 (HEK 293) cells, HEK GP2-293 (Clonetech) cells, and HCT116 cells were cultured in Gibco Dulbecco's Modified Eagle's medium (DMEM)/F-12 (Gibco) medium supplemented with 10% FBS, 1% penicillin–streptomycin and 1% L-glutamine. All cells were cultured at 37 °C in a 5% $CO_2$-humidified incubator.

To arrest RPE1 cells in prometaphase, cells were treated with 5 µM STLC (Sigma-Aldrich) for 4 hr. To test the function of PLK1 in mitosis, cells were treated with 5 µM of STLC 4 hr, followed by 2 h of 50 nM of BI2536 (Cell signaling) treatment. Protein expression in RPE1 TetON cell lines and HEK293 TetON cell lines was induced by adding the concentration of 1 µg/ml doxycyclin (Sigma-Aldrich).

CHX chase experiment: RPE1 *POC5*[-/-] cells expressing *POC5* or *POC5*[mut] were grown in 10 cm dishes to 80% confluency in the presence of doxycycline for 48 hrs, followed by 0, 2, 4, 6, 8, and 10 hr treatment of 50 nM of CHX (Sigma-Aldrich). Cells were detached and resuspended in 1 ml of PBS. Then, cell number was counted, and cells were lysed with lysis buffer. The volume of the lysis buffer was adjusted according to the cell number.

## Plasmid transfection

For RPE1 cells, transfection was performed using electroporation with the Neon™ Transfection system 100 µl Kit (Thermofischer Cat. #MPK10096) following the manufacturer's instructions. Transfection of HEK293 and HEK GP2-293 was accomplished with polyethylenimine reagent (25 kD, #9002-98-6; Sigma-Aldrich).

## Protein expression in *E. coli*

For recombinant protein expression in *E. coli*, 100–150 ng of plasmid DNA was transformed into 100 µl of competent *E. coli* BL21 CodonPlus-RIL cells (Stratagene). The following day, transformed cells were cultured in 2 × YT medium (Roth) at 37 °C in volumes of up to 500 ml until reaching an optical density ($OD_{600}$) of approximately 0.8. Subsequently, the expression cultures were shifted to 18 °C and diluted with an equivalent volume of chilled 2 × YT medium. Protein expression was induced with 0.5 mM IPTG at an $OD_{600}$ of 0.6–0.8 and carried out at 18 °C for 21 h with shaking. After expression, cells were harvested by centrifugation (4000 × *g*, 4 °C, 20 min), washed with PBS (Gibco), and either directly used for protein purification or flash-frozen in liquid nitrogen and stored at − 80 °C for later use. The constructs expressed in *E. coli* include human Augmin TII N-clamp, human His-MBP-TEV-PLK1 (residues 37–338) in pETM-41[65], and pET-51b-10 × HIS-hyperTEV60[66].

## Protein expression in insect cells

For protein expression in insect cells, bacmid DNA was prepared following established protocols and the manufacturer's instructions. Insect cells were maintained as shaking cultures in SF900 III medium (Thermo Fisher Scientific) at 27 °C. v0 virus generation was performed in Sf9 cells using Cellfectin® II (Invitrogen, Thermo Fisher Scientific) according to the manufacturer's instructions. After 72 h, the v0 baculovirus was harvested and used to infect 30 mL of Sf9 cells at a density of $1 × 10^6$ cells/ml for v1 virus generation. 48 hrs after complete infection was detected, cells were harvested by centrifugation at 800 × *g* for 5 min and stored at 4 °C, protected from light, until further use for protein expression.

For protein expression, the v1 virus was diluted 1:50 - 1:100 in expression cultures ranging from 100 ml to 400 ml of Sf21 cells at a density of $1.5 × 10^6$ cells/ml. After 60–72 hrs, cells were harvested by centrifugation at 800 × *g* for 5 min, flash-frozen in liquid nitrogen, and stored at − 80 °C for subsequent use.

The constructs expressed in insect cells included pACEBac1 *X. laevis* 2 × FLAG-TEV-NEDD1, pACEBac1 2 × FLAG-TEV-NEDD1-3C-GCP3(1–240)/ALFA-MZT1, pACEBac1 human POC5-HA-TEV-2×FLAG/Centrin2[44], *X. laevis* Cre-pACEBac1-augmin-TIII with pACEBac1-pIDK-pIDS encoding *X. laevis* Augmin-TIII (HAUS1-TEV-EGFP, 2 × FLAG-HAUS3, HAUS4, HAUS5), and pF1 human CDK1/cyclin B1/CKS1[65], Addgene #177329).The constructs expressed in insect cells included pACEBac1 *X. laevis* 2 × FLAG-TEV-NEDD1, pACEBac1 2 × FLAG-TEV-NEDD1-3C-GCP3(1–240)/ALFA-MZT1, pACEBac1 human POC5-HA-TEV-2×FLAG/Centrin2[44], *X. laevis* Cre- pACEBac1-augmin-TIII with pACEBac1-pIDK-pIDS encoding *X. laevis* augmin-TIII (HAUS1-TEV-EGFP, 2 × FLAG-HAUS3, HAUS4, HAUS5), and pF1 human Cdk1/cyclin B1/Cks1[65], Addgene #177329).

## Purification of augmin TII N-clamp

For protein purification of recombinant Augmin TII N-clamp complexes, cell pellets from a 1 l culture were resuspended in 30–35 ml of cold lysis buffer (20 mM HEPES, pH 7.4, 300 mM KCl, 1 mM $MgCl_2$, 1 mM EGTA, 2 mM DTT, 0.1% Tween-20) supplemented with one Complete EDTA-free protease inhibitor tablet (Roche) and 10 µl Benzonase (Sigma Aldrich) and kept on ice. Resuspended cells were sonicated (6 × 1 min at 100% amplitude; Hielscher UP50H), and the lysate was cleared by centrifugation at 20,000 × *g* for 30 min at 4 °C. Ni-NTA beads (Qiagen) were equilibrated with lysis buffer and incubated with the total soluble fraction of the lysate for 90 min with rotation at 4 °C, using 300 µl of Ni-NTA beads per 1 l culture. After incubation, the beads were separated from the flow-through by centrifugation (800 × *g*, 3 min, 4 °C) and washed once with lysis buffer, followed by two washes with wash buffer (20 mM HEPES, pH 7.4, 200 mM KCl, 15 mM imidazole, 1 mM $MgCl_2$, 1 mM EGTA, 1 mM DTT). Beads were sedimented by centrifugation (800 × *g*, 3 min, 4 °C) between each washing step. Recombinant proteins were eluted three times from Ni-NTA beads by incubating with 1–1.25 bead volumes of elution buffer (20 mM HEPES, pH 7.4, 200 mM KCl, 400 mM imidazole, 1 mM $MgCl_2$, 1 mM EGTA, 1 mM DTT) for 10 min at 4 °C. Elution fractions were collected via centrifugation (800 × *g*, 3 min, 4 °C) and subjected to size exclusion chromatography (SEC) using a Superose 6 Increase 10/300 GL column equilibrated in SEC buffer (20 mM HEPES, pH 7.4, 150 mM KCl, 1 mM $MgCl_2$, 1 mM EGTA, 1 mM DTT, 2.5% glycerol) at a flow rate of 0.5 ml/min. Peak fractions were analyzed by Coomassie-stained SDS-PAGE (4–20% gradient SDS precast gel; BIO-RAD), pooled, and subjected to anion exchange chromatography (AEC). AEC was performed on a Capto HiRes Q 5/50 GL column (Cytiva) equilibrated with buffer A (20 mM HEPES, pH 7.4, 75 mM KCl, 1 mM $MgCl_2$, 1 mM EGTA, 1 mM DTT, 2.5% glycerol). Proteins were eluted using a constant gradient from buffer A to buffer B (20 mM HEPES, pH 7.4, 1000 mM KCl, 1 mM $MgCl_2$, 1 mM EGTA, 1 mM DTT, 2.5% glycerol) at

a flow rate of 0.5 ml/min over 20 column volumes. Peak fractions were verified via Coomassie-stained SDS-PAGE (4–20% gradient SDS precast gel; BIO-RAD), pooled, and concentrated using Amicon centrifugal filters (30 kDa MWCO). Aliquots were flash-frozen in liquid nitrogen and stored at −80 °C for future use. All chromatography runs for protein purifications were performed using an ÄktaPure instrument (Cytiva) controlled by Unicorn software (version 7.9).

### FLAG purification of constructs expressed in insect cells

FLAG batch purification, previously described for POC5/centrin 2 and Augmin TIII, was also applied to NEDD1-N-GCP3/MZT1. Insect cell pellets from 100–400 ml of SF21 cultures were resuspended in 25–50 ml of cold lysis buffer, specific to each construct (NEDD1 constructs: 20 mM HEPES, pH 7.4, 300 mM KCl, 1 mM MgCl$_2$,1 mM EGTA, 2 mM DTT, 0.1% Tween-20; Augmin TIII: 20 mM Tris-HCl, pH 7.5, 150 mM NaCl, 1 mM MgCl$_2$, 1 mM EGTA, 1 mM DTT, 0.05% (v/v) Tween-20; POC5/centrin2: 50 mM Tris, pH 7.5, 200 mM NaCl, 1 mM MgCl$_2$). All buffers were supplemented with one Complete EDTA-free protease inhibitor tablet (Roche cOmplete Mini EDTA-free tablets Roche) and kept on ice. Resuspended cells were sonicated (3 × 1 min at 60% amplitude; Hielscher UP50H), and the lysate was clarified by centrifugation at 20,000 × $g$ for 30 min at 4 °C. Anti-FLAG M2 affinity resin (Sigma-Aldrich), equilibrated in lysis buffer (80 µl resin per 100 ml culture), was incubated with the clarified lysate for 90 min with rotation at 4 °C. At every washing or elution step FLAG resin was separated by centrifugation at 800 × $g$ for 3 min. The beads were washed once with lysis buffer and twice with wash buffer, specific to each construct. Wash buffer for NEDD1 constructs contained 20 mM HEPES, pH 7.4, 200 mM KCl, 1 mM MgCl$_2$, 1 mM EGTA, and 1 mM DTT. For Augmin TIII, it contained 20 mM Tris-HCl, pH 7.5, 150 mM NaCl, 1 mM MgCl$_2$, 1 mM EGTA, 0.5 mM DTT. For POC5/centrin2, the wash buffer consisted of 50 mM Tris, pH 7.5, 150 mM NaCl, 1 mM MgCl$_2$, 1 mM EGTA, and 0.5 mM DTT. Proteins were eluted three times from the FLAG beads by incubating the beads with wash buffer supplemented with 0.3–0.5 mg/ml 3 × FLAG peptide (Gentaur) for 20 min with rotation at 4 °C in Eppendorf tubes. Elutions were either concentrated using an Amicon 30 kDa MWCO concentrator (Merck), flash-frozen in liquid nitrogen, and stored at −80 °C or subjected to a second purification step.

Augmin TIII FLAG eluates were further purified via anion exchange chromatography (AEC) as described for N-clamp constructs, using buffers containing either 150 mM or 1000 mM NaCl in 20 mM Tris-HCl, pH 7.5, 1 mM MgCl$_2$, 1 mM EGTA, 0.5 mM DTT. For negative-stain electron microscopy (EM) experiments, FLAG eluates of NEDD1-N-GCP3/MZT1 were further purified via size exclusion chromatography (SEC), as described for N-clamp constructs.

For 100 ml insect cell pellets of Cdk1/cyclin B1/Cks1, the cell pellets were resuspended in 15 ml of lysis buffer (50 mM Tris, pH 7.5, 300 mM NaCl, 0.5 mM EDTA, 2 mM DTT, 5% glycerol, 0.1% Tween), supplemented with 1 mM PMSF. The resuspended solution was sonicated (3 × 1 min at 60% amplitude; Hielscher UP50H) and centrifuged at 20,000 × $g$ for 60 min at 4 °C. The supernatant was incubated with 100 µl of equilibrated Ni-NTA beads (Qiagen) at 4 °C for 90 min. The beads were then washed twice with lysis buffer and twice with wash buffer (50 mM Tris, pH 7.5, 600 mM NaCl, 0.5 mM EDTA, 2 mM DTT, 5% glycerol, 20 mM imidazole). Proteins were eluted 4–5 times with 100 µl of elution buffer (50 mM Tris, pH 7.5, 300 mM NaCl, 2 mM DTT, 5% glycerol, 400 mM imidazole), with each elution performed by incubating the beads for 10 min at 4 °C. The eluates were loaded onto a Superdex 75 10/300 GL column equilibrated with SEC buffer (50 mM Tris, pH 8.0, 300 mM NaCl, 1 mM DTT, 5% glycerol) at a flow rate of 0.5 ml/min. Peak fractions were verified via SDS-PAGE, combined, concentrated using Amicon 30 kDa MWCO filters, and stored at −80 °C until further use.

### Purification of proteins expressed in *E. coli*

For hTEV immunoprecipitation experiments, His-tagged hyper TEV60 protease (hTEV) was used[66]. Briefly, cell pellets from a 1 l bacterial culture were resuspended in lysis buffer (20 mM HEPES, pH 7.4, 300 mM KCl, 1 mM MgCl$_2$, 1 mM EGTA, 1 mM DTT, 0.1% Tween-20, 50 mM imidazole) supplemented with 1 mM PMSF. Cells were lysed by sonication (Hielscher UP50H, 6 × 1 min cycles, 100% amplitude), and the lysate was clarified by centrifugation at 20,000 × $g$ for 30 min at 4 °C. The soluble fraction was incubated with 500 µl Ni-NTA beads (Qiagen), pre-equilibrated in lysis buffer, for 40 min at 4 °C. The beads were washed sequentially with lysis buffer, wash buffer (20 mM HEPES, pH 7.4, 400 mM KCl, 1 mM MgCl$_2$, 1 mM DTT, 50 mM imidazole; 10% glycerol), and basic buffer (20 mM HEPES, pH 7.4, 150 mM KCl, 1 mM MgCl$_2$, 1 mM EGTA, 1 mM DTT). Bound proteins were eluted four times using 1.25 bead volumes of elution buffer (basic buffer supplemented with 400 mM imidazole). Elution fractions were analyzed via SDS-PAGE, pooled, and buffer exchanged into storage buffer (40 mM HEPES, pH 7.4, 150 mM KCl, 1 mM MgCl$_2$, 1 mM DTT, 2.5% glycerol) using a HiTrap column (Cytiva). The protein was concentrated to 400 µM, and glycerol was added to a final concentration of 50%. The protein, at a concentration of 200 µM, was stored at −20 °C until further use. An *E. coli* expression pellet of PLK1 (residues 37–338) kinase domain[65] from a 1 l culture was resuspended in 30 ml lysis buffer (50 mM Tris, pH 7.5, 300 mM NaCl, 2 mM DTT, 5% glycerol, 0.05% Tween) supplemented with 1 mM PMSF. The mixture was sonicated (6 × 1 min at 100% amplitude; Hielscher UP50H) and centrifuged at 20,000 × $g$ for 60 min at 4 °C. The supernatant was incubated with 500 µl equilibrated Ni-NTA beads (Qiagen) at 4 °C for 90 min. The beads were washed twice with wash buffer (50 mM Tris, pH 7.5, 600 mM NaCl, 2 mM DTT, 5% glycerol, 20 mM imidazole). Bound proteins were eluted 4–5 times with 1.5 bead volumes of elution buffer (50 mM Tris, pH 7.5, 300 mM NaCl, 2 mM DTT, 5% glycerol, 500 mM imidazole), incubating the beads for 10 min at 4 °C for each elution. The eluates were combined and incubated overnight at 4 °C with hTEV protease (25 µl of 200 µM). The following day, the solution was incubated with 400 µl of pre-washed Ni-NTA beads (Qiagen) to remove His-MBP and His-hTEV. The mixture was centrifuged at 800 × $g$ for 3 minutes at 4 °C, and the supernatant was loaded onto a Superdex 75 10/300 GL column equilibrated with SEC buffer (50 mM Tris, pH 7.5, 300 mM NaCl, 2 mM DTT, 5% glycerol) at a flow rate of 0.5 ml/min. Peak fractions were verified via SDS-PAGE, combined, concentrated using an Amicon 30 kDa MWCO filter, and stored at −80 °C until further use.

### γ-TuRC purification

γ-TuRC purification from *X. laevis* egg extract followed an established protocol[67]. Briefly, 380 µl of CSF-arrested *X. laevis* egg extract was incubated with an anti-γ-tubulin antibody coupled to Dynabeads Protein A (Life Technologies) for 30 min at room temperature (RT) and then placed on ice for 5 min. Afterwards, the beads were washed three times with 1 ml of CSF-XB buffer (10 mM HEPES pH7.7, 5 mM EGTA, 2 mM MgCl$_2$, 50 mM sucrose, 100 mM KCl, 0.1 mM CaCl$_2$), three times with 1 ml CSF-XB buffer containing 250 mM KCl and 0.3% Triton X-100 and twice with 0.5 ml HB100 buffer (50 mM Na-HEPES, pH 8.0, 1 mM EGTA, 1 mM MgCl$_2$, 1 mM GTP and 100 mM NaCl). After washing, the beads were incubated with 25 µl of Elution buffer (HB100 buffer containing 0.1 mg ml 1 γ-tubulin antigenic C-terminal peptide, 1 mM GTP, and 0.02% Tween20) overnight at 4 °C.

### Mass photometry

Mass photometry measurements were performed using a Refeyn TwoMP mass photometer (Refeyn Ltd, Oxford, UK). Videos of 1 min in duration were recorded with the default image size using Refeyn AcquireMP 2024 R1 software (Refeyn Ltd, Oxford, UK). High-precision microscope coverslips (24 × 50 mm) were freshly washed, dried, and

prepared for measurements. A silicone gasket with six cavities was placed in the center of the coverslip to create measurement wells. For each measurement, 19 µl of the corresponding buffer (elution buffer without 3xFLAG peptide) for the protein samples was applied to each gasket hole. Autofocus was performed before every measurement. A 1 µl aliquot of protein at 400 nM concentration was diluted in SEC buffer to a final concentration below 40 nM for measurement. Data analysis and plotting were performed using Refeyn DiscoverMP 2024 R1 software (Refeyn Ltd, Oxford, UK). Standard contrast-to-mass calibration curves were generated using bovine serum albumin (BSA, 66 kDa) and Immunoglobulin G (IgG, 150 kDa and 300 kDa).

### Generation of knock-out, knock-in and stable cell lines

Recently constructed RPE1 $TP53^{-/-}$ $POC5^{-/-}$ cells (named $POC5^{-/-}$) were used in this manuscript[44]. Endogenously SNAP-S tagged $HAUS1$ RPE1 ($p53^{-/-}$) cells were generated by electroporation with one plasmid containing the guide RNA and one donor plasmid containing the homology arm to the C-terminus of HAUS1 as described previously[68]. After electroporation, GFP-positive cells were collected and expanded for single-cell sorting. DNA from single clones was extracted with QuickExtract DNA extraction solution (Epicenter #QE09050) and verified via PCR. Clones that showed a homozygous knock-in were then sequenced and further verified by immunoblotting and immunofluorescence. RPE1 $POC5^{-/-}$ cells with the Retro-X Tet-On 3 G Inducible Expression System (Clontech) stably expressing $POC5$-HA and $POC5^{mut}$-HA were generated by integrating the constructs under TRE3G promoter via Retrovirus transfection (Clonetech). When induced, doxycycline was added to the culture at a concentration of 20 ng/ml for 72 hr.

### Centrosome isolation

The protocol for centrosome purification from KE37 and RPE1 cells was adapted from Moudjou and Bornens with some modifications[69]. Briefly, around $10^8$ cells were treated with 200 nM nocodazole and 2 µM Cytochalasin B at 37 °C for 1 hr. Following treatment, cells were pelleted by centrifugation at $280 \times g$ for 8 min and washed with 50 ml of Tris-buffered saline (TBS: 20 mM Tris-Cl, pH7.6, 150 mM NaCl, 0.1% (w/v) Tween® 20. After another centrifugation step at $280 \times g$ for 8 min, cells were resuspended in 25 ml of 0.1x TBS with 8% sucrose. Then samples were resuspended with 2 ml of 0.1x TBS with 8% sucrose and lysed by adding 8 ml of lysis buffer (1 mM HEPES, pH7.2, 0.5% NP40, 1 mM PMSF, 0.1% β-mercaptoethanol and protease inhibitor, (Roche cOmplete Mini EDTA-free tablets Roche), 0.5 mM MgCl$_2$ for RPE1 cells and 0.75 mM MgCl$_2$ for KE37 cells). The lysate was incubated on ice for 5 min. After incubation, the lysate was centrifuged at $2500 \times g$ for 10 min. 1 µg/ml DNaseI and 10 mM HEPES pH7.2 (final concentrations) was added to the supernatant, followed by incubation on ice for 30 min. To perform the sucrose cushion, the supernatant was transferred onto 1.25 ml of 60% sucrose solution (w/w) prepared in 10 mM Pipes, pH7.2, 0.1% Triton X-100, 0.1% β-mercaptoethanol and centrifuged at $10400 \times g$ for 30 min. After the centrifugation, approximately 8 ml of supernatant from the top was removed, and the remaining samples were mixed well and prepared for plunge freezing.

### Electron microscopy of resin-embedded centrosomes

Purified centrosomes from KE37 cells were processed for electron microscopy as follows: purified centrosomes on coverslips were rinsed with phosphate-buffered saline (PBS) 3 times and then pre-fixed with a mixture of 2.5% glutaraldehyde (GA)/1.6% paraformaldehyde (PFA)/2% sucrose in 50 mM cacodylate buffer for 30 min at RT. Coverslips with centrosomes were washed 5 times with cacodylate buffer, post-fixed with 2% OsO$_4$ for approximal 45 min on ice, in darkness. After postfixation coverslips with centrosomes were washed 4 times with dH$_2$O and incubated overnight at 4 °C in 0.5% uranyl acetate. On the following day coverslips were rinsed again 4 times with dH$_2$O and subsequently stepwise dehydrated with ethanol. Coverslips were

immediately placed on capsules filled with Spurr-resin (Sigma-Aldrich) and polymerized at 60 °C for approximately two days. Resin-embedded centrosomes were sectioned using a Reichert Ultracut S Microtome (Leica Instruments, Vienna, Austria) to a thickness of approximately 80 nm. Post-staining with 3% uranyl acetate and lead citrate was performed. Sections were imaged at a Jeol JE-1400 (Jeol Ltd., Tokyo, Japan), operating at 80 kV, equipped with a 4k × 4k digital camera (F416, TVIPS, Gauting, Germany). Electron micrographs were adjusted in their brightness and contrast by applying Fiji software.

### Negative stain EM of NEDD1-N-GCP3/MZT1

For the preparation of NEDD1-N-GCP3/MZT1 samples, 4 µl of SEC elution were applied onto glow-discharged copper-palladium 400 mesh EM grids, coated with an approximately 10 nm thick continuous carbon layer (G2400D, Plano GmbH). The grids were incubated for 30 seconds at RT before being blotted with Whatman filter paper 50 (CAT N.1450-070) and subsequently washed with three drops of distilled water. The samples were then stained using 3% uranyl acetate in water. Imaging was conducted with a Talos L120C TEM equipped with a 4k × 4k Ceta CMOS camera (Thermo Fisher Scientific). Data acquisition was performed using EPU software (v2.9, Thermo Fisher Scientific), with a nominal defocus of 2 µm and an object pixel size of 0.1992 nm.

Image processing of negative stain EM data was performed in RELION 3.1[25] at 1.992 Å/px. Micrographs were subjected to initial CTF estimation, using the RELION implementation of Gctf[70]. Initially, a subset of 512 manually picked particles, recentered after one round of 2D classification into 6 classes, was used as a template for Autopicking. The resulting 37873 candidate particles were subsequently extracted in a 128 px box and subjected to 2 consecutive rounds of 2D classification into 200 and 10 classes respectively, using a mask radius of 150 Å. Ultimately, 464 particles were retained from 2D classifications and were used for the generation of a 3D ab initio model, yielding a NEDD1 grapnel-like 3D density.

### Immunofluorescence

Cells were grown on 12 mm coverslips and fixed with cold methanol at −20 °C for 5 min. Subsequently, cells were blocked in PBS supplemented with 10% FBS and 0.1% Triton X-100 for 1 h at RT. After 1 h of incubation with primary antibody, which is diluted in 3% BSA solution, cells were incubated with secondary antibody and DAPI in 3% BSA solution for 45 min and mounted with Moviol. Image acquisition was performed using the DeltaVision RT system (Applied Precision) with an Olympus IX71 microscope equipped with 60 ×/1.42 and 100 ×/1.40 oil objective lenses.

### Expansion microscopy

Cells seeded on 12 mm coverslips were extracted with CSK buffer (10 mM K-PIPES, pH7.0, 100 mM NaCl, 3 mM MgCl$_2$, 300 mM sucrose, and 0.5% Triton X-100) and fixed with FA/AA solution (1% acrylamide, 0.7% formaldehyde in PBS) at 37 °C for 5 h. Afterward, 35 µl of monomer solution (containing Na-acrylate [38%], acrylamide [40%], N,N′-methylenebisacrylamide [2%], 10 × PBS) supplemented with 0.5% APS and 0.5% TEMED were added to cells on coverslips, incubated for 5 min, followed by incubation at 37 °C for 1 hr. After that, gels were denatured by shaking at 400 rpm at RT and then incubated at 95 °C for 1.5 hr. After denaturation, gels were expanded in distilled water twice for 30 min and put back in PBS before incubating with primary antibody diluted in 1% BSA solution overnight at 30 °C. After three washes in PBS, the gels were incubated with a secondary antibody diluted in 1% BSA solution for 3 hr at 30 °C. After three more washes, gels were expanded again with distilled water and then transferred to Ibidi µ-Dish 35 mm coated with poly-L-lysine for imaging. Samples were imaged using an inverted Leica TCS SP8 STED 3x microscope equipped with FALCON FLIM with a HC PL APO 100 ×/1.40 STED White Oil objective. Raw images were deconvoluted by Huygens' Deconvolution

software (SVI Inc.). The z-stack spanning the centrioles was z-projected using Fiji[71].

## Fluorescence recovery after photobleaching (FRAP) analysis

RPE1 cells endogenously expressing SNAP-tagged HAUS1 were grown to 80% confluency on μ-Slide 8-well chambered coverslips (Ibidi). Cells were incubated in SNAP-tag labeling medium (1 μM SNAP-Cell TMR-Star dye diluted in complete medium supplemented with 10% FBS, 1% penicillin-streptomycin) for 30 min at 37 °C for, 5% CO$_2$. After three washes in culture medium, cells were incubated in fresh medium for another 30 min, followed by one more wash and re-cultivation. For FRAP, the cells were analyzed with an LSM 780 confocal microscope using a 63x/1.40 NA Plan-Apochromat oil objective lens (Carl Zeiss) and a 561 nm diode-pumped solid-state laser. A small circular region of interest was photobleached, and florescence recovery was monitored over time, as described previously[72]. For short-term FRAP observing 2 min of recovery time, single plane images were recorded at 1 s intervals. For long-term FRAP monitoring 30 min of recovery time, 3 μm axially-ranging volumes (z-stacks) were recorded at 10 s intervals. Z-stacks and open pinhole settings were used to increase the detection volume in order to keep the signal in focus at possible cell movements over the extended time course. Image analysis to determine intensity recovery curves was done with Fiji[71]. Average intensity projections of the z-stacks were used for the quantification of long-term FRAP.

## Mass spectrometry

Proteins were digested using 5 mM TCEP (Aldrich), 20 mM CAA (Aldrich), and trypsin (Promega) in a 1:50 (enzyme:protein) ratio for overnight digestion at 37 °C. The next day, peptides were cleaned up with an OASIS® HLB μElution Plate (Waters) according to manufacturer instructions. Desalted peptides were reconstituted in 30 μL of 4% LC/MS grade acetonitrile (TH Geyer), 1% LC/MS grade formic acid (Fisher Scientific) in LC/MS grade water (TH Geyer) prior subjection to LC-MS/MS.

In an UltiMate 3000 RSLC nano-LC system (Dionex) fitted with a trapping cartridge (μ-Precolumn C18 PepMap 100, 5 μm, 300 μm i.d. x 5 mm, 100 Å) and an analytical column (nanoEase™ M/Z HSS T3 column 75 μm x 250 mm C18, 1.8 μm, 100 Å, Waters), trapping was performed for 6 minutes with a constant flow of trapping solvent (0.05% trifluoroacetic acid in water) at 30 μL/min onto the trapping column. This was followed by peptide elution and separation on the analytical column using a gradient composed of Solvent A ((3% DMSO, 0.1% formic acid in water) and solvent B (3% DMSO, 0.1% formic acid in acetonitrile) with a constant flow of 0.3 μL/min. After the analytical column, the sample was directly fed into a QExactive plus (Thermo Fisher) mass spectrometer using the nanoFlex source (positive ion mode).

Using a spray voltage of 2.4 kV and a capillary temperature of 275 °C, peptides were injected into the Orbitrap Fusion Lumos via a Pico-Tip Emitter 360 μm OD x 20 μm ID; 10 μm tip (CoAnn Technologies). For mass range 300–1500 m/z, full mass scans were obtained in profile mode (resolution of 120000), using a filling time of a maximum of 250 ms and a limitation of $2 \times 10^5$ ions. The instrument was operated in data-dependent acquisition (DDA) mode, and MSMS scans were acquired in the Orbitrap with a resolution of 15000, with a fill time of up to 32 ms, and a limitation of $\times 10^5$ ions (AGC target). The normalized collision energy was set to 30. MS$^2$ data was acquired in centroid mode.

Using MSConvert (ProteoWizard)[73], raw files were converted to mzML format, using peak picking, 64-bit encoding, and zlib compression, and filtering for the 1000 most intense peaks. Subsequently, we used MSFragger (v4.0)[74] in FragPipe (21.1) to search converted files against FASTA database UP000186698_XenopusLaevis_J2019_I-D8355_34806entries_28102022_dl18012023 which also includes common contaminants and reversed sequences. The workflow used Philosopher (v5.1.0) and IonQuant (1.10.12) for label-free

quantification. For searches, the following modifications were considered: carbamidomethylation (C, 57.0215) as fixed modification and oxidation (M, 15.9949) and acetylation (protein N-terminus, 42.0106) as variable modifications. A mass error tolerance of 20 ppm was used for the full scan (MS1). For MS/MS (MS2), spectra of 20 PPM were applied. For protein digestion, the protease setting was 'stricttrypsin' (allowing maximum 2 missed cleavages) requiring a minimum peptide length of 7 amino acids. Further settings were a false discovery rate on peptide and protein level of 0.01. The standard settings of the FragPipe workflow 'Default' were used. The following modifications were made: ionquant.maxlfq: 0, ionquant.mbr: 0, ionquant.normalization: 0, ionquant.uniqueness: 1, quantitation.run-label-free-quant: true.

## Protein pull-down from human cell lysate

For the interaction between POC5 and Augmin TII-N-clamp, HEK293 cells from three 10 cm dishes with overexpressed POC5, POC5$^{mut}$ were resuspended with 1 ml Lysis buffer (50 mM Tris, pH7.5, 1 mM EGTA, 0.5 mM DTT, 150 mM NaCl, 1 mM MgCl$_2$ (0.1% Tween-20- Benzonase 5 μl and Protease inhibitor (1:50)) and incubated for 10 min. The lysate was pipetted up and down every 3 min 20 times up and down and afterward centrifuged for 20 min at 20,000 × g at 4 °C. Afterward, the supernatant was incubated either with 25 μg of Augmin TII N-clampΔH8N, or negative controls, X. laevis Augmin TIII tetramer or buffer. Samples were mixed with 30 μl of GFP binder beads and incubated for 1 hr at 4 °C[75]. Afterward, the beads were washed once with lysis buffer and 3 times with 600 μl of basic buffer (50 mM Tris, pH7.5; 1 mM EGTA, 0.5 mM DTT, 150 mM NaCl, 1 mM MgCl$_2$) and eluted with 2xSDS-Laemmli buffer.

For the interaction between NEDD1 and γ-TuRC, HEK293 cells from one 10 cm dish with overexpressed NEDD1, NEDD1$^{VVI}$, and NEDD1$^{LLM}$ were resuspended with 400 μl of Lysis buffer (50 mM Tris, pH7.5, 150 mM NaCl, 1 mM DTT, 10% glycerol, 0.1% Triton X-100; Protease inhibitor (Roche cOmplete Mini EDTA-free tablets Roche) and PhosSTOP (Roche) and incubated on ice for 30 min. Afterward, the lysate was cleared by centrifugation at 14000 × g for 30 min at 4 °C. The supernatant was incubated with 60 μl of HA magnetic beads for 1 hr at 4 °C with rotating. After that, beads were washed three times with lysis buffer and eluted with 2x SDS-Laemmli buffer.

## hTEV pull-down of purified protein complexes

Pull-down experiments of purified proteins were conducted using GFP-binder beads (NHS-Activated Sepharose Fast Flow, Cytiva)[75] and hTEV protease for specific elution. For PD1, POC5/Augmin - N-clamp, reaction mixtures (200 μl total) were prepared with 25 μl SEC-purified POC5/centrin2 complex (0.3 μg/μl) and 25 μl AEC-purified Augmin N-clamp (7 μM) in a buffer containing 50 mM HEPES, pH 7.4, 150 mM NaCl, 1 mM EGTA, 1 mM MgCl$_2$, 1 mM DTT, 0.1% Tween-20, and 5% glycerol. For PD2, NEDD1-GCP3-MZT1 - Augmin TIII, reaction mixtures (120 μl total) were prepared with 25 μl FLAG-purified NEDD1-N-GCP3/MZT1 complex (0.3 μg/μl) and 25 μl AEC-purified Augmin TIII tetramer (2 μM) in a reaction buffer containing 50 mM HEPES, pH 7.4, 150 mM NaCl, 1 mM EGTA, 10 mM MgCl$_2$, 1 mM DTT, 0.1% Tween-20, 5% glycerol, and 1 mM ATP. In one condition, recombinant 15 μl PLK1 (8 μM) and 15 μl Cdk1/cyclin B1/Cks1 complex (2 μM) were added. Negative control for both pull-down experiments were prepared by replacing EGFP-tagged Augmin complexes with buffer. Input samples (20 μl) were collected and mixed with 40 μl of 2 × Laemmli buffer. For IP2, input samples were collected after 30 min incubation of the reaction mix at RT. Reactions were incubated with 25 μl equilibrated GFP-binder beads, pre-blocked with 3 mg/ml BSA, for 1 hr on ice. Afterward, the beads were washed twice with 400 μl of Reaction Buffer and twice with 400 μl of Wash Buffer (Reaction Buffer supplemented with 200 mM NaCl). Elution was performed by adding 2.5 μl hTEV protease (200 μM) in 100 μl Reaction Buffer directly to the beads, followed by incubation at 4 °C for 1 hr. Eluted proteins were separated from the beads by

centrifugation ($800 \times g$, 4 min, 4 °C), mixed with 4 × Laemmli buffer, and analyzed by immunoblot using the corresponding antibodies. Note that in PD2 experiments, HAUS1 migrates at a lower molecular weight on SDS-PAGE after hTEV digestion compared to the purified sample. This shift is due to the cleavage of the EGFP tag during the elution process.

## Immunoblotting
RPE1 cells were lysed with lysis buffer (50 mM Tris, pH7.5, 150 mM NaCl, 5 mM EDTA, 5 mM EGTA, 0.5% NP-40, 1 mM DTT, 1 mM PMSF, and 1 tablet of cOmplete Protease Inhibitor Cocktail tablet (Roche)). After incubating on ice for 10 min, cell lysates were centrifuged at $10800 \times g$ at 4 °C for 10 min. The supernatant was mixed with 4x SDS-Laemmli buffer (3:1 sample:buffer) and boiled at 95 °C for 8 min. After boiling and centrifugation, samples were subjected to SDS-PAGE. The proteins were transferred onto a 0.2 μm PVDF membrane in 7 min using the Trans-Blot Turbo Transfer System (Bio-Rad) following the manufacturer's protocol.

## Sample preparation for MINFLUX
Cells were grown on 18 mm coverslips and fixed with 4% (w/v) formaldehyde in PBS for 20 min at RT. Formaldehyde was quenched for 5 min in 100 mM $NH_4Cl$ in PBS followed by three washes with PBS. Cells were then incubated with 0.3% Triton X-100 in PBS for permeabilization and blocked in PBS supplemented with 10% FBS 0.1% Triton X-100 for 1 hr at RT. After incubation with primary antibody diluted in 0.1% BSA solution overnight at 4 °C, cells were incubated with secondary antibody for 30 min and subsequently incubated with 2.5 μM SNAP-Surface® Alexa Fluor® 647(S9136S, New England Biolabs) for 1 hr at RT. Samples were washed once with PBS. Subsequently, the PBS solution was replaced with a gold bead dispersion (A11-200-CIT-DIH-1-100, Nanopartz) for 5 min. After gold bead incubation samples were gently washed twice by pipetting flow to remove unbound gold beads. To stick gold beads very tightly to the cells, samples were submerged for 30 sec in poly-L-lysin (P8920, Sigma Aldrich). Subsequently, samples were stored in PBS until mounting them for MINFLUX measurements. Samples were mounted on microscopy slides with a concave well (1320002, Marienfeld) in STORM buffer: 50 ml Tris pH 8.0, 10% glucose, 10 mM NaCl, 10 mM cysteamine (30070, Sigma Aldrich), 0.4 mg/ml glucose oxidase (G2133, Sigma Aldrich), 64 μg/ml catalase (C1345, Sigma Aldrich) and sealed with eco-sil extrahart (Picodent).

## MINFLUX data acquisition
MINFLUX measurements were performed on an Abberior MINFLUX microscope (Abberior Instruments[45] and controlled by the software Imspector (v16.3.15636-m2205-win64-MINFLUX). The MINFLUX set up was equipped with a 1.45x UPLanXAPO Olympus objective on an inverted Olympus body (IX83) and four laser lines: 405, 488 nm, 561, and 640 nm.

Region of interest (ROI) were identified by POC5-AF 488 and respective colocalization of HAUS1-SNAP-AF 647 staining, using 488 nm and 640 nm lasers in confocal mode. Once an ROI was selected, the active sample stabilization system was locked (x,y, and $z > 1$ nm during measurements). During all measurements, the MINFLUX system's beamline monitoring was used to correct for drift with a minimum of four selected gold beads. Three-dimensional MINFLUX measurements were performed with the MINFLUX sequence (Supplementary Table 5) provided by the manufacturer. Before MINFLUX measurements were started, a short-lasting preexcitation step was performed with ~ 4 μW of 640 nm laser power to transfer enough AF 647 fluorophores into a non-fluorescent state. MINFLUX acquisitions were performed with ~ 20 μW laser power and a pinhole size of 0.83 AU. To keep a constant rate of incoming valid localization, the UV laser was ramped up manually, in a step-wise manner to ~300 nW until the end of the measurement. All laser powers were measured at the sample plane.

## MINFLUX data postprocessing
Localization data were obtained from 3 to 4 independent experiments (cells from different passages and respective newly prepared samples). In each experiment, a minimum of three centrosomes from three different cells were acquired. All data were exported from Imspector and further processed in Python 3.11.4. Valid localizations were extracted from the 9th iteration, and the z-position localizations were corrected with the scaling factor of 0.7[76]. All traces with a minimum of 4 localizations were kept. Only localizations with <100 kHz effective frequency at offset (EFO) were used for further analysis.

MINFLUX data visualization was performed by using a pipeline of pyMINFLUX version 0.3.0[77] and the visualization software Paraview 5.11.2[78]. The NumPy file of the MINFLUX raw data was uploaded in pyMINFLUXv0.3.0. This data set was filtered with an effective frequency at offset upper limit of 100 kHz likewise used in the data analysis pipeline. This data set was saved in a PMX file format and then loaded into Paraview. In ParaView, the data was rendered with a Gaussian of 6 nm and displayed in plain circles. The image files were saved in tiff file format.

For further analysis, the xyz coordinates of all localizations per trace were averaged and transformed into UCSF Chimera-compatible marker files[79]. Next, the axes were defined per centriole by calculating a collective centroid through which the axis was fitted by minimizing the radial distance to all data points.

## Grid preparation and cryo-ET data acquisition for purified centrosomes
Holey carbon grids (R2/1 Cu 200 mesh; Quantifoil) were glow discharged for 60 sec under an oxygen atmosphere in a Gatan Solarus plasma cleaner. The grids were prepared in a Vitrobot Mark IV (Thermo Fisher Scientific) at 4 °C and 100% humidity by applying 3 μl of sample and 3 μl Protein A-conjugated 10 nm gold-beads (AURION) 10 times concentrated and resuspended in sample buffer. Grids were subsequently blotted for 0.5–1 sec at blot force 10 using Whatman 1 circular filter papers of 55 mm diameter and plunge frozen in liquid ethane cooled by liquid nitrogen.

Tilt-series were acquired on a Titan Krios (Thermo Fisher/FEI) operated at 300 kV with the 100 μm objective aperture inserted and equipped with a K3 camera mounted after a Quantum GIF (Gatan), using a 20 eV slit. Data acquisition was carried out in EPU Tomo5 (Thermo Fisher Scientific), using a dose-symmetric acquisition scheme over a tilting range of − 60° to + 60° at incremental steps of 3° with a starting angle of 0°. Tilt images were acquired with an object pixel size of 2.54 Å/px, an average dose of 5.3 e⁻/A² distributed over 5 frames, and a defocus of -3 to -6 μm. Data acquisition parameters are summarized in Supplementary Table 8.

## Processing of cryo-ET data for purified centrosomes
Motion correction of movie frames was performed in MotionCor2 as implemented into RELION 3.1 using $5 \times 5$ patches[25,26,80]. Following CTF estimation in Warp, tilt-series stacks and rawtlt files were exported for subsequent gold fiducial-based tilt-series alignment in IMOD[81–83]. Tilt series that could not be aligned and reconstructed due to an unbalanced distribution of gold beads were excluded. Tilt-series alignment parameters determined in IMOD were imported into Warp for tomogram reconstruction at a scaled pixel size of 17.78 Å/px[83].

The processing workflow for cryo-ET data collected on purified centrosomes has been summarized in Supplementary Fig. 2 (KE37 cells) and Supplementary Fig. 11 (RPE-1 wild-type and *CDK5RAP2* KO cells). Manually selected γ-TuRC particles and ribosome particles were extracted from five tomograms (17.78 Å/px) and iteratively aligned in PyTom v0.971 to generate reference-free and completely data-driven averages, which were used for template matching in PyTom v0.971[84]. Per tomogram, 1600–2000 candidate positions were identified at peaks of the cross-correlation function and imported into Warp for

subtomogram reconstruction at a scaled pixel size of 5.08 Å/px[83,84]. The resulting subtomograms were transferred into RELION 3.1 for two consecutive rounds of 3D classification using the upscaled averages from template matching as initial ref. 25,26. The final sets of subtomograms were subjected to 3D auto-refinement in RELION 3.1[25,26], followed by joint auto-refinement of particle poses and stage angles, as well as image and volume warp grids in M[27].

The following steps were performed only for purified centrosomes from KE37 cells. Refined γ-TuRC-containing subtomograms were reconstructed in Warp and transferred back to RELION 3.1 for global 3D auto-refinement, followed by focused 3D classification on spokes 11–13 (without sampling)[25,26,83]. Subsets of particles representing the inwards and outwards conformations were selected, and each set of particles was individually subjected to 3D auto-refinement for subsequent structural analysis[26]. To improve the resolution of GCP2-associated CM1 modules, all four GCP2 subunits in the first half of the ring were refined together and computationally sorted for the presence of the CM1 module. For this purpose, subtomograms centered on 4-spoked γ-TuRC segments comprising GCP2 at the first spoke (1–4, 3–6, 5–8, 7–10) were reconstructed and subjected to 3D auto-refinement in RELION 3.1, followed by two rounds of 3D classification focused on the CM1 module of GCP2 at spoke 1 of the 4-spoke segments. 3D classes featuring CM1 module density were merged and subjected to another round of 3D auto-refinement and subsequent post-processing in RELION 3.1. Data processing statistics are summarized in Supplementary Table 8. Local resolution estimation and angular distributions of cryo-ET reconstructions are shown in Supplementary Fig. 24.

## Rigid body fitting of γ-TuRC models into the inwards and outwards conformations

An atomic model of native human γ-TuRC (PDB: 6V6S)[4], was split into rigid bodies and fitted into post-processed cryo-EM reconstructions of the inwards and outwards conformations in ChimeraX[85–87] as follows. GCP subunits of spokes 1–12 were split into GRIP1 and GRIP2 domains (GCP2: GRIP1: N-term-536, GRIP2:537-C-term; GCP3: GRIP1: N-term-553, GRIP2:554-C-term; GCP4: GRIP1: N-term-349, 350-C-term; GCP5: GRIP1: N-term-713, 714-C-term; GCP6: GRIP1: N-term-1473, 1474-C-term). The GRIP1 domains of spokes 1-12 were grouped together and fitted as one rigid body. For each spoke individually, the GRIP2-domain and associated γ-tubulin were grouped and fitted as one rigid body. The GCP and γ-tubulin subunits of spokes 13 and 14 were grouped and fitted as one rigid body.

## Density subtraction for analysis of CM1 module binding

Density subtraction was performed in ChimeraX[85–87]. For analysis of conformation-dependent CM1 module binding on spokes 1,3,5,7 of γ-TuRCs from KE37 cells, post-processed reconstructions of the inwards and outwards conformations were superposed based on spokes 1–8 and subtracted from each other. For analysis of CM1 module binding on spoke 13, the cryo-EM reconstructions were superposed based on spokes 13 & 14 for density subtraction. For identification of CM1 modules, cryo-EM reconstructions of γ-TuRCs from RPE1 wild-type and *CDK5RAP2*[−/−] cells were low-pass-filtered to 30 Å using relion_image_handler, histogram-scaled in ChimeraX[85–87] and subsequently superposed and subtracted from each other.

## Geometry analysis of γ-TuRC conformational states

Geometry analysis for the atomic models of the γ-TuRC inwards and outwards conformations, as well as the recombinant (PDB: 7AS4)[7] and native[4] isolated γ-TuRC (PDB: 6V6S) was done in accordance with Vermeulen et al.[55], with the exception that only spokes 3–8 were used for superposition of the models. The template for geometry analysis was an atomic model of the closed γ-TuRC capping an MT minus-end (PDB: 8VRK), by Aher et al.[53], which was aligned along the *z*-axis based

on residue 15 of every MT-protofilament's α- & β-tubulins and of γ-tubulins, of the γ-TuRC itself, on spokes 2–14.

## Grid preparation, cryo-FIB milling and cryo-ET data acquisition for HCT116 cells

Cryo-EM grids (R2/1, gold, 200 mesh, Quantifoil Micro Tools) were glow-discharged in a Gatan Solarus 950 plasma cleaner under oxygen atmosphere for 20 sec. HCT116 cells were cultured on glow-discharged cryo-EM grids and vitrified in liquid ethane cooled by liquid nitrogen using a Vitrobot Mark IV (ThermoFisher Scientific) operated at RT and ambient humidity. One-sided blotting was performed with a blot force of 10 for 15 sec using Whatman filter paper Nr. 1 from the back side of the grid and a Fluoropolymer sheet (Science Service) from the front.

Vitreous lamellae were prepared in an Aquilos 2 dual-beam FIB-SEM microscope (ThermoFisher Scientific). Before milling selected cells, the specimen was coated with organometallic platinum for 5 sec. During milling, biological material was ablated in 4 successive steps using a Gallium-ion beam (acceleration voltage 30 eV) with decreasing current. The target thickness for lamella preparation was 150 nm.

Tilt-series were acquired using the same setup and parameters as described for purified centrosomes but using SerialEM[88] with a bidirectional tilt-scheme starting from 35° and an angular increment of 3° (first branch: 35° to −60°; second branch: 38° to 60°). The exposure dose at minimal tilt was 3.3 e⁻/Å2 and was increased on the fly for higher tilt angles to match the mean count of the minimal tilt image. Data acquisition parameters are summarized in Supplementary Table 8.

## Processing of cellular tomography data

The processing workflow for cellular cryo-ET data collected has been summarized in Supplementary Fig. 13. For in-house acquired tilt-series, motion correction of movie frames was performed in MotionCorr2 as implemented in RELION 3.1 using 5 × 5 patches[25,26,80]. Tilt images were scaled to an object pixel size of 4.3 Å during import into Warp[83] to match 200003-EMPIARC, which contains a tilt-series of native HeLa cell centrosomes obtained by cellular cryo-ET[38]. Motion-corrected tilt-series from 200003-EMPIARC were directly imported into Warp and merged with in-house acquired tilt-series. Following CTF estimation in Warp, tilt-series stacks and rawtlt files were exported for subsequent tilt-series alignment via patch tracking in IMOD[81–83]. Tilt-series alignment parameters determined in IMOD were imported into Warp for tomogram reconstruction at a scaled pixel size of 17.2 Å/px[83]. At this stage, only tilt-series without obvious radiation damage and with fully intact centrosomes were retained.

Manual selection and iterative alignment of candidate particles generated reference-free data-driven templates for subsequent template matching in PyTom v0.971[84], as described above. The resulting cross-correlation volume was manually trimmed to the centrosome area in each tomogram. Candidate positions were identified at peaks of the cross-correlation function and clearly false positive positions on MTTs and membranes were excluded by visual inspection. The remaining positions were imported into Warp for subtomogram reconstruction at the full pixel size of 4.3 Å/px[83,84]. The resulting subtomograms were subjected to two consecutive rounds of 3D classification in RELION 3.1 to remove the remaining false-positive candidate positions[25,26]. Cryo-ET data for HCT116 and HeLa cells were processed separately up to this point and merged for a final 3D auto-refinement run in RELION 3.1[25,26]. Notably, the number and distribution of ribosomes in the vicinity of centrosomes was too low for local tilt-series alignment in M.

The processing workflow for MTTs from cellular cryo-ET data has been summarized in Supplementary Fig. 14. A cryo-EM reconstruction of the MTT from HeLa cells (EMD-33417)[38] was converted into a PyTom v0.971-compatible average and used for template matching[84], as described above for γ-TuRCs. The resulting candidate positions were imported into Warp for subtomogram reconstruction at a scaled pixel

size of 8.6 Å/px[83]. Subtomograms were subjected to 3D auto-refinement, followed by one round of 3D classification without sampling in RELION 3.1[25,26]. Features resolved in the output 3D classes allowed the assignment of particles to the distal, proximal, and central regions of the centriole. Data processing statistics are summarized in Supplementary Table 8. Local resolution estimation and angular distributions of cryo-ET reconstructions are shown in Supplementary Fig. 24.

### Distribution analysis of centrosomal γ-TuRCs in cellular cryo-ET data

γ-TuRC positions were visually split into centriole-lumenal and pericentriolar pools and the normal distance towards the trajectory of MTT B-tubules approximated based on the cross-correlation volumes from MTT template-matching was determined.

### Cluster analysis of γ-TuRC-MTT configurations

For each γ-TuRC particle, we determined the normal vector to the closest MTT axis. Subtomograms containing the respective MTT segments were reconstructed at 8.6 Å/px in WARP and subsequently subjected to 3D auto-refinement in RELION 3.1[25,26]. The orientation of MTT segments determined during 3D auto-refinement was visualized in the context of the tomograms in UCSF Chimera and MTT segments with orientations deviating from the interpolated MTT axis were rejected[79]. The remaining 232 γ-TuRC-MTT particle pairs of 15 centrosomes, which were distributed over 12 tomograms, were analyzed for preferred spatial configurations using MATLAB scripts from Brandt et al.[89,90] In brief, the pair configurations were filtered according to a distance cutoff between the center of masses from 60–120 nm. The remaining particle pair configurations were defined based on the relative position and relative orientation of particles. The relative positions were represented by a shift vector between the centers of mass in Cartesian space (3 coordinates). The relative orientations were represented in quaternion space (4 coordinates). Particle pair configurations in the resulting 7-dimensional space were grouped by k-means clustering, revealing one main cluster representing 42% of analyzed particle pairs. The configuration was visualized based on models and cryo-EM densities placed using the average relative position and orientation of particle pairs included in the cluster.

### AlphaFold predictions

AlphaFold2 and AlphaFold Multimer[29,30] predictions (Supplementary Fig. 3a–o, Supplementary Fig. 16b–g, and Supplementary Fig. 8a–c) were performed using AlphaFold 2.3.1 or 2.3.2 (see Supplementary Table 6) with default settings, yielding 25 models per prediction. Relaxation was performed for select predictions (see Supplementary Table 6). AlphaFold3 predictions (Supplementary Fig. 8d–f) were performed on the AlphaFold server (release 2024.08.19; https://alphafoldserver.com/)[37]. A full list of sequence fragments used for AlphaFold predictions and the corresponding UniProt IDs can be found in Supplementary Table 6.

### Grid preparation and data acquisition for cryo-EM SPA of *X. laevis* γ-TuRCs

Inside the chamber of a Vitrobot Mark IV (Thermo Fisher Scientific) equilibrated to 4 °C at 100% humidity, 4 μl of γ-TuRC purified from *X. laevis* egg extracts was applied to a holey carbon grid (R2/1 Cu 200 mesh Quantifoil) that was glow-discharged on a PELCO easiGlow. After 10 sec of waiting, an initial blot of 0.5 sec with blot force 0 was performed. Subsequently, another 4 μl of sample was applied, followed by 5 sec of waiting, 5 sec of blotting at blot force 5 and, after 1 sec of draining time, plunged into liquid ethane. Grids were stored in liquid nitrogen until use.

Data was acquired on a Titan Krios (Thermo Fisher/FEI) operated at 300 kV with the 100 μm objective aperture inserted and equipped with a K3 camera mounted after a Quantum GIF (Gatan), using a 20 eV slit. Using EPU (Thermo Fisher), 29,516 movie stacks were acquired at a defocus of -1 to -3 μm at a nominal magnification of 81kx (1.07 Å/px). Per the movie, a cumulative dose of 51 e⁻/Å² was applied and distributed over 50 fractions.

### Data processing and model building for cryo-EM SPA

The processing workflow for cryo-EM SPA has been summarized in Supplementary Fig. 5a. In RELION 3.1[25], movies were motion-corrected using MotionCor2 in 5 × 5 patches[80], and the contrast transfer function (CTF) of the resulting micrographs was estimated using Gctf[70]. Particle positions were identified using auto-picking using an available cryo-EM reconstruction of the *X. laevis* γ-TuRC (EMD-10491) as a 3D ref.[3]. This reference contains no density for the NEDD1/N-GCP/MZT1 grapnel and density for the GRIP2 domains and γ-tubulins at spokes 5 and 6 was removed. 5,288,186 particles were extracted at 4.28 Å/px, split into 9 subsets and classified in 3D into 6 classes in one to two iterations. Afterward, true-positive γ-TuRC classes, totaling 1,044,985 particles, were merged and subjected to 3D auto-refinement.

To improve the comprehensiveness of particle picking, Topaz was trained on sorted and refined particles from a random subset of 60 micrographs[91]. After picking on the entire dataset, 5,917,849 particles were extracted at 4.28 Å/px, split into 8 subsets, and classified in 3D into 6 classes. The retained particles were split into 2 subsets and subjected to 3D classification into 6 classes. Particle selection in both classification runs was based on the presence of density for the GRIP2 domains and γ-tubulins at spokes 5 and 6 in the respective classes. The retained 537,241 particles were subjected to 3D auto-refinement and combined with the retained 1,044,985 particles from auto-picking. Duplicate particles were removed, leaving 1,209,627 particles that were then recentered and re-extracted at 2.14 Å/px before a round of 3D auto-refinement. Subsequently, particles were recentered and re-extracted at 1.43 Å/px and subjected to three rounds of 3D auto-refinement interspersed with Bayesian polishing and CTF refinement[92]. From Bayesian polishing onwards, all processing was performed in RELION 3.0[92] unless mentioned otherwise. Using a solvent mask covering the region where the grapnel-shaped density could be observed at low-density thresholds, particles were 3D classified into 6 classes without performing image alignment. One class with stoichiometric density for the NEDD1/N-GCP/MZT1 grapnel, containing 299,022 particles, was retained and subjected to 3D auto-refinement. The resulting reconstruction was sharpened and filtered to local resolution in RELION 3.0 (Supplementary Table 2). For visualization in Fig. 2a, the reconstruction was weighted using a sigmoid filter in OccuPy[93] ($\gamma = 2.6$, $\mu = 0.07$).

To build an atomic model of the native *X. laevis* γ-TuRC with bound NEDD1/N-GCP3/MZT1 grapnel, individual models of GCP2-6 and γ-tubulin molecules from an available structure of the *X. laevis* γ-TuRC (PDB: 6TF9) were rigid-body docked into the reconstruction[3]. For spoke 13 and 14, GCP and γ-tubulin were docked together as a single rigid body. GCP5[587-601] was added based on a publicly available AlphaFold2 prediction. The lumenal bridge N-GCP/MZT1 modules were taken from PDB: 9EOJ[55], rigid-body docked, and trimmed appropriately. GCP6[206-254], bound at spoke 1 and 2, was taken from an AlphaFold2 prediction of GCP6 with GCP2 and GCP3 (Supplementary Fig. 3m–o). The NEDD1/N-GCP3/MZT1 grapnel was predicted using AlphaFold2, relaxed into a segmented density of the grapnel in 3 consecutive rounds of molecular dynamics flexible fitting (MDFF) implemented in Namdinator[94] (Start temperature 600, 1000, and 1000 K, respectively, 200000 simulation steps per round, no Phenix relaxation; otherwise default settings) and finally trimmed. GCP5[227-261] and GCP5[292-307] was taken from an AlphaFold2 prediction of GCP4, GCP5, and the N-GCP3/MZT1 module. GCP4[16-36], GCP5[207-226] and GCP6[311-347] were taken from an AlphaFold2 prediction of GCP4, GCP5 and the N-GCP3/MZT1 module. GCP6[155-184] was taken from an AlphaFold2 prediction of full-length GCP6. Details on used AlphaFold2

predictions can be found in Supplementary Table 6. Chains were fused in Coot[95]. Finally, the atomic model was subjected to MDFF in Namdinator with 20000 simulation steps without Phenix relaxation and otherwise default settings[94].

To resolve the N-GCP/MZT1 module on spoke 14 in the γ-TuRC with stoichiometric NEDD1/N-GCP3/MZT1 grapnel, the particles after CTF refinement (see above) were, in parallel, sorted in 3D into 5 classes without sampling and with a solvent mask covering spoke 13 and 14, as well as into 3 classes without sampling and a solvent mask covering the grapnel. The 82,813 particles present in the selected classes from both classification runs were combined and subjected to 3D auto-refinement, followed by 3D auto-refinement with a solvent mask covering spokes 12–14 and applying local angular sampling. Lastly, the reconstruction was subjected to post-processing in RELION 3.0.

To explore the flexibility of the NEDD1/N-GCP3/MZT1 grapnel, all particles with stoichiometric grapnel (see paragraph above) were subjected to 3D classification without sampling and using a solvent mask covering the grapnel. All 3 classes were subjected to 3D auto-refinement and filtered to local resolution in RELION 3.0[92].

To further improve the resolution of alpha-helical density in the core of the grapnel attributed to NEDD1, we re-extracted the 299,022 γ-TuRC particles with stoichiometric NEDD1/N-GCP3/MZT1 grapnel at 1.43 Å/px (256 px box), centered on the NEDD1/N-GCP3/MZT1 grapnel. Using a solvent mask covering γ-TuRC segments that remained within the box, we performed 3D auto-refinement, followed by Bayesian polishing. We then performed another round of 3D auto-refinement starting with the same solvent mask. When the estimated angular accuracy reached below 1.5 degrees, we continued the same 3D auto-refinement with a soft focus mask covering the NEDD1/N-GCP3/MZT1 grapnel as well as the GRIP1 domains of spokes 9–12. The resulting half maps were independently corrected using spIsoNet's anisotropy correction module[96] using the default parameters and subsequently sharpened and filtered to local resolution using RELION 3.0[92] (Supplementary Table 2). For visualization in Fig. 2b, the reconstruction was resampled to 0.5 Å/px and segmented in ChimeraX; dust was hidden.

To generate an atomic model for this reconstruction, the output atomic model for the full γ-TuRC with grapnel obtained as described above was trimmed to retain the grapnel, GRIP1 domains of spokes 1 to 12, and the lumenal bridge without actin. Firstly, the entire model was rigid-body docked into the reconstruction, followed by individually rigid-body docking of each chain. Lastly, the atomic model was subjected to MDFF in Namdinator with a total of 240000 simulation steps without Phenix relaxation[94].

All cryo-EM SPA data collection and model statistics are summarized in Supplementary Table 2.

## Statistics

Signal intensity measurements of immunofluorescence images were performed on maximum intensity-projected images using Fiji software[71]. Band intensity of immunoblot images was quantified using Fiji[71]. For statistical analysis, unpaired two-tailed student *t* tests or one-way ANOVA (GraphPad Prism) was used to test the statistical significance between the sample conditions. All error bars show SD. Percentiles have been calculated in R[97], using a median-unbiased approach[98].

## Research animals

Husbandry of *Xenopus laevis* female frogs (Xenopus1, Dexter, MI, USA) was approved by the City of Bonn, Germany, quoting §11 Abs. 1, Nr.1 of the German law for animal protection under file reference 76-5/2022/, Amt für Umwelt und Stadtgrün according to EU guideline 2010/63 for aquatic anura. Hormone applications for egg production were approved by the Landesamt für Natur, Umwelt und Verbraucherschutz Nordrhein-Westfalen, Germany, under the file reference 81-02.04.40.2022.VG027.

## Reporting summary

Further information on research design is available in the Nature Portfolio Reporting Summary linked to this article.

## Data availability

The atomic coordinates and the cryo-EM densities for the *X. laevis* γ-TuRC with stoichiometric NEDD1/N-GCP3/MZT1 grapnel generated in this study have been deposited in the Protein Data Bank and the Electron Microscopy Data Bank under accession codes PDB: 9I8N and EMD-52730 (consensus reconstruction), PDB: 9I8M and EMD-52729 (focused refinement on grapnel with spoke 9-12 GRIP1), and EMD-52728 (focused refinement on spoke 12–14). The cryo-EM densities and associated models (if applicable) derived from cryo-ET data generated in this study have been deposited in the Electron Microscopy Data Bank and Protein Data Bank under accession codes EMD-52722 (γ-TuRCs from cells), EMD-52717 (KE37 γ-TuRC consensus refinement), PDB: 9I8G and EMD-52718 (KE37 γ-TuRC inwards conformation), PDB: 9I8H and EMD-52719 (KE37 γ-TuRC outwards conformation), EMD-52721 (RPE1 wild-type γ-TuRC), EMD-52720 (RPE1 *CDK5RAP2*[-/-] γ-TuRC), and EMD-52723 (MTTs from cells; all particles as well as topological classes). The models predicted by AlphaFold and AlphaFold-Multimer generated in this study have been deposited in the ModelArchive database (https://modelarchive.org/) with the identifiers ma-odixd (human NEDD1[C]/N-GCP3/MZT1 grapnel), ma-imf31 (*X. laevis* NEDD1[C]/N-GCP3/MZT1 grapnel), ma-phn3m (p1 N-GCP3/MZT1 module with GCP4, GCP5 and GCP6 GRIP1), ma-rmctb (p2 N-GCP3/MZT1 module with GCP4, GCP5 and GCP6 GRIP1), ma-1vc4y (GCP2 and GCP3 GRIP1 with N-GCP6), ma-89x6h (NEDD1[C] tetramer with N-GCP3/MZT1 module and Augmin TIII), ma-rwmkh (POC5 dimer with centrin 2 and Augmin TII N-clamp), ma-2zl3w (γ-TuSC with MZT2 and CDK5RAP2 dimer), ma-8bwwq (N-GCP2/MZT2 module), ma-yzz9w (N-GCP5/MZT1 module), ma-14965 (N-GCP3/MZT1 module) and ma-z68wf (*X. laevis* GCP6). Mass spectrometry data of purified *X. laevis* γ-TuRC are available as Supplementary Data 1 and have been deposited to the ProteomeXchange Consortium via the PRIDE[99] partner repository with the dataset identifier PXD060119. Tilt series of centrosomes in vitreous sections of HeLa cells used in this study are available in the EMPIARC database under accession code 200003. Source data are provided in this paper.

## Code availability

Custom scripts used in this study are available in Supplementary Software 1.

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

## Acknowledgements

Human *HAUS2*, *HAUS6*, *HAUS7*, and *HAUS8* genes were a gift from Dr. Laurence Pelletier. RPE1 *CDK5RAP2*[-/-] cells were a kind gift from Dr. Rosa M. Rios, and the PLK1 construct a kind gift from Dr. Andreas Boland. We thank Dr. Holger Lorenz from the ZMBH imaging facility for the help with the HAUS1 FRAP experiment. We are grateful to Sophie Kopetschke (ZMBH, Heidelberg University) and Dr. Sebastian Eustermann (EMBL, Heidelberg) for support and advice during cryo-EM data processing. Furthermore, we acknowledge access to the infrastructure of the Cryo-EM Network at Heidelberg University (HDcryoNET) and thank Dr. Petr Chlanda and Moritz Wachsmuth-Melm (BioQUANT, Heidelberg University), Dr. Götz Hofhaus (BioQUANT, Heidelberg University), Dr. Dirk Flemming (BZH, Heidelberg University) and Dr. Jan

Rheinberger (BZH, Heidelberg University) for support. We thank Giulia Tonon for providing a purified recombinant Augmin TII N-clamp. We further thank Dr. Karine Lapouge from the Protein Expression and Purification Core Facility (PEPCF) at the European Molecular Biology Laboratory (EMBL Heidelberg, Germany) for assistance with mass photometry. We are grateful to Dr. Mandy Rettel (Proteomics Core Facility, EMBL, Heidelberg) for acquiring and processing mass spectrometry data and Frank Stein (Proteomics Core Facility, EMBL, Heidelberg) for assistance with data deposition. The authors acknowledge support by the state of Baden-Württemberg through bwHPC and the German Research Foundation (DFG) through grant INST 35/1597-1 FUGG. The authors gratefully acknowledge the data storage service SDS@hd supported by the Ministry of Science, Research and the Arts Baden-Württemberg (MWK) and the German Research Foundation (DFG) through grant INST 35/1503-1 FUGG. The MINFLUX nanoscope was financed by the European Union's Fund for Regional Development (EFRE) -innovation and energy change as part of the reaction to the COVID-19 pandemic (REACT-EU). The MINFLUX nanoscope is associated with the CellNetworks Core Technology Platform (CCTP) of Heidelberg University. The CCTP is funded in part by the Federal Ministry of Education and Research (BMBF) and the Ministry of Science Baden-Württemberg within the framework of the Excellence Strategy of the Federal and State Governments of Germany. This work was further supported by grants of the Deutsche Forschungsgemeinschaft (DFG) to E.S. (DFG Schi 295/4-4; DFG SCHI 295/11-1; DBT-DFG Schi 295/9-1) and to S.P. (DFG PF 963/1-4; DFG PF 963/4-1).

## Author contributions

Conceptualization: Q.G., F.W.H., S.F., B.J.A.V., M.W., E.S., and S.P. Data Curation: Q.G., F.W.H., B.J.A.V., S.F., C.K., E.S., and S.P. Formal Analysis: Q.G., F.W.H., B.J.A.V., M.W. Funding Acquisition: E.S., S.P. Investigation: Q.G., F.W.H., S.F., B.J.A.V., M.W., C.K., A.N., and M.Z. Methodology: Q.G., F.W.H., S.F., B.J.A.V., M.W., C.K., A.N., C.S., H.S., and M.Z. Project Administration: E.S. and S.P. Resources: C.K., C.S., O.G., E.S., and S.P. Software: S.F. and S.P. Supervision: E.S. and S.P. Validation: Q.G., F.W.H., B.J.A.V., S.F., M.W., C.K., and A.N. Visualization: Q.G., F.W.H., B.J.A.V., M.W., C.K., and A.N. Writing – original draft: Q.G., F.W.H., S.F., B.J.A.V., M.W., E.S., and S.P. Writing – review & editing: Q.G., F.W.H., B.J.A.V., S.F., M.W., C.K., E.S., and S.P.

## Funding

## Competing interests

The authors declare no competing interests.
