## [Transparent Peer Review file · Nature Communications]

Structural mechanisms for centrosomal recruitment and organization of the microtubule nucleator γ -TuRC

Corresponding Author: Dr Stefan Pfeffer

Version 0:

Reviewer comments:

Reviewer #1

(Remarks to the Author)

Centrosomal recruitment, distribution around the centrosome, and key interacting partners of the microtubule nucleator gamma-tubulin ring complex (γ -TuRC) that facilitate the former have been enigmatic and difficult to study. This impressive work by Qi Gao and colleagues from Schiebel and Pfeffer labs is a breakthrough, uncovering numerous structural determinants of the specific spatial arrangement of γ -TuRCs around centrosomes, in their native cellular context, in various stages of the cell cycle. It is an important, elaborate study of high quality and impact. The manuscript is well written, with a good, logical storyline, easy to follow despite the volume of work presented. The interpretation and discussion of the results are convincing and interesting.

Minor corrections and suggestions

Abstract:

- 1) Line 25: not clear what the "ring" is in this context, as the nature of the γ -TuRC is not explained. Perhaps start the abstract with: "The γ -tubulin ring complex (γ TuRC)..."
- 2) Lines 28 and 31: inconsistent spelling, but "luminal" is probably correct

Introduction:

- 3) Line 51: why not develop CDK5 abbreviation, too
- 4) Line 57: "bind" missing?
- 5) Line 61: looks like too many "the"
- 6) Line 69: would suggest a simple "show" instead of "decipher"

Results:

- 7) Line 255: don't need "for"
- 8) Line 356: not explained what CHX does, so the rationale for the chase experiment may be unclear to some readers

Methods:

- 9) Line 484: looks like ", a" should be after "SNAP"
- 10) Line 537: "donor"
- 11) Line 541: "were then"
- 12) Line 565: "processed for electron microscopy as follows:"
- 13) Line 598: delete "it"
- 14) Line 601: "after three washes in PBS"
- 15) Examples in lines 606 and 618: inconsistent naming of Fiji software - unify across the whole manuscript
- 16) Line 609: "expressing"
- 17) Line 620: colon after the heading, not found elsewhere
- 18) Line 647: ". Runs were performed..."
- 19) Line 651: abbreviation "AEC" not found previously developed
- 20) Line 669: "After the washes"?
- 21) Line 686: not explained what the proteinase inhibitor was - anything else on top of PMSF? Then need to specify
- 22) Line 688: "with" missing? What were the proportions?

- 23) Line 688: "After boiling"?
- 24) Lines 695-696: "followed by three washes with PBS"?
- 25) Line 708: concentrations of glucose oxidase and catalase are missing
- 26) Line 807: Would be good to add details about resolution filtering and normalisation, e.g. software
- 27) Line 832: "second branch: -35"?
- 28) Line 841: "200003-EMPIAR dataset"? Would be good to explain what that dataset represents
- 29) Examples in lines 864 and 878: Warp/WARP
- 30) Line 882: suggest adding "name et al." before reference numbers
- 31) Line 927: mention criteria for particle retainment/selection
- 32) Lines 928, 931, 932...: was it 3D auto-refinement? Suggest precision here
- 33) Line 938: call out the table that provides the B factor and mention the software used for filtering

Figures:

- 34) Line 1325, Figure 6 legend: should "(red)" and "(green)" be swapped?
- 35) Extended Data Fig. 4: would be good to number the spokes, at least 13 and 14 in the panel I
- 36) Line 1449: how filtered, in Relion, Chimera?
- 37) Extended Data Fig. 5, panel G: the third "i" is missing in the "unidentified" label
- 38) Line 1524-1525: again, would be good to mention how filtered and normalised
- 39) After Extended Data Fig. 10, figures are indexed as "S11" etc. (tables also: S1...)

Questions:

- 1) Based on the differential patterns of ring decoration with CM1 modules, correlated with the conformational rearrangements in the ring, can the authors comment if the activation model in which CDK5RAP2 induces the closure of the ring is likely to be correct? It seems difficult to reconcile with the lack of nascent MT nucleation events captured.
- 2) Why do the authors think purified centrosomes lose luminal γ -TuRCs? Are the identified tethers predicted to be weak? Intuitively, one would assume they might be somewhat protected from being stripped as opposed to γ -TuRCs exposed in the PC region, which in contrast appear to withstand the isolation procedure.

Reviewer #2

(Remarks to the Author)

The manuscript by Gao et al uses cryo-electron tomography and alphafold predictions to describe the structure and arrangement of the microtubule nucleator gamma-tubulin ring complex (gamma-TuRC) in situ at centrosomes in cells and at purified centrosomes. They focus on two different sub-populations associated with the pericentriolar material (PCM) and the centriole lumen, respectively. The authors define the structure of the gamma-TuRC in the PCM, where they observe two different conformations that are proposed to correlate with different interaction modes with the activator CDK5RAP2, and describe a density at the seam of the gamma-TuRC cone that likely corresponds to the adapter NEDD1. gamma-TuRCs are also present in the lumen of the centrioles, where the authors describe their ultrastructural organization, although the advance here over previous work is limited. They propose a protective function for luminal localization that prevents the degradation of augmin and gamma-TuRC during interphase and upon release during mitosis provides additional nucleation activity to boost spindle assembly.

The main novelty of the manuscript resides in that it is the first study of the structure and arrangement of the vertebrate centrosomal gamma-TuRCs in situ. Similar work was recently published, but for the budding yeast oligomeric gamma-TuSC bound to spindle pole bodies, a system that differs in several ways from vertebrate gamma-TuRC at centrosomes (Dendooven et al., 2024, Nat Struc Mol Biol). The current work also describes the first structure of the binding interface between gamma-TuRC and its adapter NEDD1.

Overall, this is an important study that raises some interesting ideas and potentially controversial concepts. Unfortunately, many claims are overly ambitious and are not supported by the data. It seems that the manuscript tries to answer too many questions. It would have been useful to focus the manuscript on fewer of the various aspects of centrosomal gamma-TuRC structure and function, interpret the data with more caution, and provide more mature analyses.

Regarding the text, the manuscript is well-written, but in both results and figure legends the descriptions are too brief and lack important information to understand what exactly was done. Also, the authors need to do a better job in citing and discussing previous studies.

Major points:

1) In situ cryo-EM of gamma-TuRC: the obtained resolution of 16.4 Å may allow docking of the known subunit structures, but it is not sufficient, using only alphafold predictions and no additional experimental structures, to determine the identity of new densities. Thus, the claim "This model could be seamlessly docked as a rigid body into the unoccupied density segment capping the γ -TuRC cone (Fig. 1D), identifying the structure and binding site of NEDD1 on the γ -TuRC." is an overinterpretation of the data.

2) Related to point 1, the authors also obtained higher resolution maps that include the extra density for soluble TuRC. However, TuRC from extract is used, which is not of a defined composition. Again, the indicated resolution of 5-8 Å for the

new density is not sufficient to “unambiguously” identify part of this density as NEDD1, more caution is needed here. While the known MZT1 modules have a characteristic fold and may be docked, the part claimed to belong to NEDD1 is very small (2 consecutive helices) and could also belong to another protein. There is also the possibility that more than one protein could bind at this site by providing helices that engage with MZT modules. The authors could use a recombinant system, as in their previous work, to reconstitute and directly probe the structure of gamma-TuRC-bound NEDD1.

3) Lines 150-160: Description of residues and interactions. How are the positions of these side chains determined? There does not seem to be sufficient resolution for such detailed analysis.

4) Identity of the MZT1 module in the extra density: apart from N-GCP3/MZT1 in the luminal bridge and N-GCP5/MZT1 in spoke 14, the cross correlation analysis in Table S1 shows minimal differences for MZT1 modules in the different densities of the current study. Also, the legend states that “models with the highest cross correlation for each density are highlighted in bold”, but there is no consistent pattern underlying the bold highlights.

In the density map of the current study 7QJD and AF2 correlate similarly well with the spoke 14 module (0.55-0.56; for some reason not highlighted in bold) as with the grapnel modules (0.53-0.57).

Have the authors tested cross-correlation with N-GCP2/MZT2 modules? They have a similar structure and should be included in the analysis.

5) Inwards vs outwards conformation: as presented in the figures, it is not clear what the main difference between the conformations is. It seems that “inwards” refers mainly to the last gamma-TuSC at pos. 13/14. Positions 11/12 of the same conformation actually appear more “outwards” (more elliptical; seen in Fig. 3A, B, C). Related to this, there is no information in the results or figure legend, how these differences were quantified in Fig. 3. How are alignments done, what residues are used for measurements, etc. It would also be useful to plot and compare the measurements for recently published MT end bound gamma-TuRC, which is in a more circular, closed conformation.

6) Extra densities assigned to CM1 around the gamma-TuRC: the authors claim that these match “the position and shape of CM1/N-GCP2/MZT2 modules”. I understand that this interpretation makes sense based on the recently posted bioRxiv study by the Wicczorek group that is cited in the introduction, but the data presented in the current study should stand for themselves. The low resolution extra densities in panel 3D do not support this claim. Apart from this, it is also unclear how the subtraction was done, since the differential densities do not seem to correspond to the extra densities seen in the individual maps. Perhaps displaying superposed densities would help.

Also, apart from panel 3D, which shows only two examples, I cannot find data analysing all positions and which positions are occupied simultaneously to support this statement: “In the ‘outwards’ conformation, density for CM1 modules could be detected exclusively on spoke 13, as described before. In contrast, in the ‘inwards’ conformation, the CM1 module binding site on GCP2 at spoke 13 was vacant and the γ -TuRC harbored CM1 modules on the remaining GCP2 subunits at spokes 1, 3, 5 and 7.”

7) While analysis of centrosomes from CDK5RAP2 KO cells is a good experiment to identify CDK5RAP2-dependent densities, this does not identify these as CM1 densities. Again, it makes sense in the light of results from studies by others, but the data provided in the current study do not prove this.

Also, similar to panel 3D, the Extended Data Fig. 7 shows differential densities that are not seen in the individual maps. Is this due to differential thresholding? If so, this needs to be adjusted. If I understand the minimal description well, the red densities should correspond to the extra densities seen in the green maps, which does not seem to be the case.

8) Spatial organization of gamma-TuRC relative to centrioles: how many centrioles were analyzed for this and how many gamma-TuRCs per centriole? It would be important for the reader to know, whether the authors present some fraction of all gamma-TuRCs present in a centrosome or a comprehensive mapping of all gamma-TuRCs (within the limits of the method).

9) POC5-augmin interaction: I cannot find evidence in the presented experiments for the claimed direct interaction between POC5 and augmin. This should be phrased and interpreted with more caution.

10) Analysis of POC5 and POC5mut cells:

a) regular wide field imaging is not suited to quantify luminal pools of POC5, augmin or g-tub. This should be done by super resolution imaging.

b) it is not explained how exogenous POC5 and POC5mut are expressed in POC5 KO cells, since they seem to be controlled by an inducible promoter. When is expression induced and what are the expression levels relative to endogenous?

11) The current quality of the UExM is not very convincing. In panel S13C, exogenous POC5 and in particular the POC5mut clearly do not localize robustly along the length of the central inner wall as seen for endogenous POC5. Signals appear restricted to a smaller region and to more distal parts. The same is true for HAUS4. In full length centrioles the central scaffold lines the central region but does not extend to the distal end (see work by Guichard/Hamel group).

12) What is the relevance of the centriole length defect observed for POC5 KO cells? Previous work using POC5 RNAi has observed not only length defects, but also structural defects including broken centrioles and should be referenced in this context. (Azimzadeh et al., 2009, JCB; Steib et al., 2020, eLife). Do the authors also see this here and are all defects rescued by both WT and mutant POC5? See also comment on mitotic defects below.

13) Stability of luminal augmin-gamma-TuRC:

a) The FRAP experiment was done only for about 2 min, which is not a relevant time scale related to cell cycle dependent turnover during interphase. Much longer times should be tested to assess turnover rates (CHX chase in 6F was done for 10 hours)

b) The destabilization in the absence of POC5 does not necessarily indicate stabilization in the lumen, since there is likely a cytosolic fraction of POC5. Do the authors have evidence that POC5 is associated with augmin-gamma-TuRC only in the centriole lumen?

14) The sub-centrosomal localization of HAUS1-Strep is unclear. The quantification in Fig. 6 needs to be done with super resolution imaging to reveal the specific luminal signal.

15) The idea of simultaneous decoration of gamma-TuRC with NEDD1 and CM1 of CDK5RAP2 is interesting, but contrasts with previous studies. NEDD1 has not been observed in CM1/CDK5RAP2-bound gamma-TuRC and vice versa (Choi et al, 2010, JCB; Muroyama et al, 2016, JCB) and both adapters were shown to form distinct complexes not only in keratinocytes (Muroyama et al, 2016). Considering that the simultaneous binding cannot be unambiguously identified at centrosomes in cells (see above), the authors should add biochemical proof of this simultaneous interaction. Also, the current discussion of the previous work regarding this point is too brief.

16) For all CRISPR-edited cells lines, homo/heterozygosity testing by PCR and sequencing needs to be provided. Deletion should also be confirmed by western blotting. For all tagged proteins (endogenous or exogenous) western blotting needs to be provided to confirm the size and expression level. In addition, correct subcellular localization needs to be confirmed by immunofluorescence microscopy. Super resolution is needed to confirm luminal vs PCM localization. Regarding tagging of augmin subunits, which are vital for mitosis, it should be confirmed that these cells cycle and divide normally (normal mitotic index, mitotic figures and proliferation).

17) In Fig. 6A the HAUS1-Strep spindle staining is very weak. One would expect robust localization to spindle MTs (Lawo et al, 2009) including some augmin signal coming from the midzone area at later stages (Uehara et al, 2009). Do these cells undergo normal mitosis and do other augmin antibodies show proper spindle staining in these cells?

18) In Fig. 6E– the authors show a decrease in the total levels of augmin and gamma-tubulin in the absence of POC5 and attribute this to destabilization of the luminal pool. However, previous studies have shown that only a very small fraction of the total cellular gamma-TuRC is present at the centrosome (including both luminal and PCM pools), and that the large majority is in the cytosolic fraction, ranging from 80% (Moudjou et al, 1996) to 99% (Bauer et al, 2016). If only the luminal pool is stabilized, one would not expect a strong decrease in the total levels. The authors should test what the proportion of luminal augmin and gamma-TuRC relative to the cytosolic pool is in their case, to see if this is compatible with their model.

19) The POC5 KO background produces mitotic defects in most of the cells according to Fig. 6I. However, the nature of the mitotic defects is unclear. Since POC5 RNAi was shown to produce structurally aberrant centrioles (Steib et al., 2020, Elife), which may interfere with mitosis, the authors should quantify centriole number and configuration using a centriole marker such as centrin and test for structural abnormalities by UExM. The same should be performed for the rescue lines expressing wild type POC5 and mutant POC5. If centrioles are abnormal in POC5 mutant cells (due to a defective inner scaffold), this would readily explain the mitotic defects.

20) In Fig. 6F, the use of POC5 KO cells with induced POC5 rescue construct would make more sense as a control. This would confirm that destabilization happens because of the deletion of POC5 and not as a result of a secondary effect of clonal selection.

Minor points:

-Line 59: "Biochemical studies have revealed that the key adapter protein NEDD1 co-purifies with the γ -TuRC." References are needed here.

-Line 47: Several references of the multiple previous cryo-EM structures are missing here.

-Line 54: Several additional studies should be cited here: Zhu et al., 2008, Curr Biol; Gomez-Ferrera et al., 2007, Curr Biol; Luders et al., 2006, NCB; O'Rourke et al., 2014, Plos One; Chinen et al., 2021, JCB; Gavilan et al., 2018, EMBO Rep,...

-The colors of the immunofluorescence images that are red and green could be changed to color-blind friendly ones.

-Although this is the first article to describe structural aspects of the luminal gamma-TuRCs, luminal gamma-TuRC has already been studied previously not only in Schweizer et al, but also a more recent paper (Laporte et al, 2024). Both papers should be cited and discussed, since their findings differ in some aspects from the current manuscript. Also, numerous previous studies showing luminal localization of gamma-TuRC have not been cited at all (just to name a few: Fuller et al. 1995, Curr Biol; Moudjou et al. 1996, JCS; Sonnen et al., 2012, Biol Open; Lawo et al., 2012, NCB).

-In line 56 references 11 and 12 are cited for the luminal organization of POC5-Augmin-gamma-tubulin, but this is not discussed in these articles.

-In line 57, a word is missing

-In the Extended Data Fig. 7C the numbers in the graph are missing, and the number of cells quantified are also missing from the figure legend.

-Extended Data Fig. 7D, the density that is suggested to be NEDD1 should be labeled to understand the figure more easily.

-In Extended Data Fig. 13B, HA intensity, changing the left axis of the graph would facilitate the visualization of the data. Is it normally distributed? With this format it's difficult to see the distribution, and everything looks close to 0.

-There are inconsistencies with the naming of the supplementary figures, some of them are named "Extended Data", and some of them just "Figure S...".

Reviewer #3

(Remarks to the Author)

y-TuRC is a vital part of the cell's machinery, essential for templating microtubule formation including at centrosomes. In this manuscript, Gao and colleagues use various advanced microscopy techniques to analyze y-TuRC distribution and structure in human cells and purified centrosomes.

They begin by using cryo-ET to visualize y-TuRC in the pericentriolar material (PCM) of centrosomes purified from human KE-37 cells. By applying subtomogram averaging to the y-TuRC particles, they identify additional density resembling a grapnel bound near the base of the y-TuRC cone, which they tentatively assign to a tetramer of the C-terminus of NEDD1. In support of this assignment, they purified y-TuRC from *X. laevis* egg extract, where it is known to copurify with NEDD1, and performed single-particle analysis cryo-EM. The structure has improved density for the grapnel (~4-8 Å resolution), allowing more confident (but not conclusive) docking of an AlphaFold2 prediction of a NEDD1C/N-GCP3/MZT1 tetramer.

Next, they performed classification of the PCM y-TuRCs and identified two different conformations – outwards and inwards – each associated with additional density likely corresponding to CM1 motifs. A hypothesis that these CM1 motifs came from CDK5RAP2 was supported by cryo-ET of centrosomal y-TuRC purified from wild-type and CDK5RAP2^{-/-} RPE1 cells. Assuming their assignments are correct, this would suggest that, contrary to a prior report, CDK5RAP2 and NEDD1 can bind y-TuRC simultaneously.

Purification of centrosomes could influence PCM organization, so the authors next turned to in-situ cryo-ET of centrosomes in FIB-milled human HCT116 cells. They observed two pools of y-TuRC – one in the PCM and one in the centriolar lumen, as expected from prior expansion microscopy studies. They demonstrate that y-TuRCs within the centriolar lumen are preferentially orientated with spokes 9-12 towards the centriolar wall.

Next, the authors speculated that y-TuRC could be anchored via the NEDD1C/N-GCP3/MZT1 module to the centriole by Augmin. In support of this hypothesis, they used AlphaFold predictions, and MINIFLUX to show that Augmin subunit HAUS4 has a similar distribution to luminal y-TuRCs. They made a mutant of POC5, designed to disrupt its interface with Augmin, and showed using immunofluorescence microscopy that the localization of y-TuRC and Augmin to the centriole lumen was reduced but not abolished.

They next examined the cell-cycle-dependent localization of the different proteins using various types of fluorescence microscopy. They find that HAUS4 (Augmin subunit), GCP4 (y-TuRC) and NEDD1 are lost at the onset of M phase and that the signal decrease of Augmin is partially blocked by addition of PLK1 inhibitor, suggesting release from the centriole is mediated by PLK1 activity.

In summary, this paper uses cutting-edge methods to explore y-TuRC distribution and structure at centrosomes making a number of novel observations that should be of interest within the field. The findings are well-presented, and the methods section and extended data are comprehensive. However, as mentioned below, further information is required to fully support some of the conclusions of the paper prior to publication.

Major comments

1. The authors have not provided sufficient evidence to conclusively identify the additional density bound to human and frog y-TuRC as a tetramer of NEDD1 bound to N-GCP3:MZT1. Currently, the assignment is based on prior work showing that NEDD1 associates with y-TuRC purified from *X. laevis* and in-vitro studies showing NEDD1 C-termini form tetramers. While the AlphaFold prediction fits the density well, the local resolution of the cryo-EM structure (4-8 Å) is too low to make a conclusive assignment and the authors have had to hide/exclude additional regions of the proteins (including the N-terminal beta-propeller domain of NEDD1, which is never mentioned in the text) which do not fit the density. Further raising doubt on the assignment is an inconsistency with prior work showing that the C-terminus of NEDD1 (residues 572-660) directly binds y-tubulin (PMID: 20224777). The authors need to provide additional evidence that NEDD1 is responsible for the grapnel-like

density, as this is a major conclusion of the paper.

2. The model that Augmin connects γ -TuRC to the MTTs of the centriole via POC5 is interesting and supported by the preliminary studies. However, additional studies, for example in vitro reconstitution or mutation of the NEDD1-Augmin interface – are needed to provide support for the model.

3. All the structural predictions use AlphaFold2 with subcomplexes. The authors should demonstrate that these predictions are replicated with AlphaFold3 and larger subcomplexes.

Minor comments

1. The description of the γ -TuRC:NEDD1 structure could be improved to reduce reliance on prior knowledge of the γ -TuRC structure.
2. SDS-PAGE and mass spectrometry analysis should be provided of the purified *X. laevis* γ -TuRC to indicate whether other potential interaction partners are present. A pure sample would help support NEDD1 as being responsible for the grapnel-like density.
3. Data from Extended Data Figure 7D and Figure S13 could be moved to a main figure, as they provide key support for some of the main conclusions of the paper
4. Label the C-terminus of NEDD1 in Figures 1 and 2
5. Label individual tubules of the MTT in Figure 4
6. Better labeling is required in Figure 5H to prevent it being misinterpreted as an experimentally resolved reconstruction.
7. The grapnel should be clearly labeled in the local resolution figure of Extended Data Fig. 4
8. In the Research Animals section, “the” is repeated twice
9. Is Augmin long enough to bridge the larger gaps between γ -TuRC and the MTT?

Version 1:

Reviewer comments:

Reviewer #1

(Remarks to the Author)

This reviewer is completely satisfied.

(Remarks on code availability)

Reviewer #2

(Remarks to the Author)

The authors have done a great effort to address the reviewers' concerns and the manuscript has been substantially improved. In particular the structural analyses with improvements in resolution are now more convincing.

The following remaining issues should be addressed:

Our previous comment:

19) The POC5 KO background produces mitotic defects in most of the cells according to Fig. 6I. However, the nature of the mitotic defects is unclear. Since POC5 RNAi was shown to produce structurally aberrant centrioles (Steib et al., 2020, Elife), which may interfere with mitosis, the authors should quantify centriole number and configuration using a centriole marker such as centrin and test for structural abnormalities by UExM. The same should be performed for the rescue lines expressing wild type POC5 and mutant POC5. If centrioles are abnormal in POC5 mutant cells (due to a defective inner scaffold), this would readily explain the mitotic defects.

Author reply:

We now show in Supplementary Fig. 17g that centrioles are not fragmented in POC5mut cells in contrast to POC5^{-/-} cells. Similarly, expression of POC5 and POC5mut rescues centriole number compared to POC5^{-/-} cells (Supplementary Fig. 17d). This indicates that the inner scaffold is intact and functional in POC5mut centrioles.

Our new comment:

Supp Fig 17D: The description in the results refers to centrioles, but g-tub is quantified here, which does not allow counting of centrioles. In the plot, what are “defined” g-tub foci? Are there additional “undefined” foci?
A centriole marker should be used (ideally centrin), as asked in the first review round, to confirm that there are no abnormalities in centriole numbers/configurations in POC5 vs POC5 mut.
Most importantly, centrin staining should be done for the mitotic figures in Fig 6I that we refer to in our original comment and where the extra g-tub foci are observed.

The authors should explain how they selected the centrioles for U-ExM and quantifications in Supp Fig 17 (since only

isolated centrioles are shown). This is important because the results will depend on the cell cycle stage, whether daughter or mother centrioles are analysed, etc.

The authors should provide example images for the centrioles quantified in Supp Fig 17G (centriole length, % broken centrioles). Also, the graph axis label says broken "centrosomes", this should be "centrioles" (I assume).

Author reply:

What we see during mitosis in POC5mut cells is an increase in mitotic γ -tubulin centers and an increase in metaphase cells with misaligned chromosomes (Fig. 6h,i). POC5mut cells in comparison to POC5 wild-type cells show γ -tubulin and augmin release from interphase centrioles and overall less cellular γ -tubulin and augmin. Taken these observations together, it is sensible to propose the model that the γ -tubulin and augmin localization inside interphase centrioles and their timely release in mitosis are important to prevent mitotic defects.

Our new comment:

Assuming there are no centriole defects (see above) and since the total levels of augmin and g γ ub are reduced, an alternative explanation would be that the authors see an augmin depletion phenotype for POC5mut cells, which also causes spindle pole fragmentation (see Lawo et al 2009 Curr Biol). If this cannot be excluded, it should be discussed.

(Remarks on code availability)

Reviewer #3

(Remarks to the Author)

In my opinion, the authors have done a tremendous job of addressing the concerns of the reviewers. The additional experiments strongly support the assignment of NEDD1, and I have no further comments except to congratulate the authors on an impressive body of work.

(Remarks on code availability)

Point-by-Point Reply to Reviewers

We thank the reviewers for their thoughtful and supportive comments. Below we address all the specific points raised by the reviewers and elaborate on the corresponding changes in the manuscript.

Importantly, we have included an extensive set of new experiments, which overall significantly strengthen our manuscript. This entails experiments that validate the assignment of NEDD1 in our cryo-EM reconstruction and strongly support the model that γ -TuRCs are coordinated in the centriole lumen by direct POC5-Augmin- γ -TuRC interactions. Moreover, we performed an extensive characterization of centriole structure and centrosomal protein distribution in *POC5^{mut}* and *HAUS1-SNAP-S* cell lines using ultrastructure expansion microscopy and complementary methods. Our findings confirm the critical role of POC5 in retaining Augmin- γ -TuRC within the centriole lumen during interphase for protection from degradation and validate the specific release of Augmin- γ -TuRC during mitosis, ensuring proper chromosome alignment.

Reviewer #1 (Remarks to the Author):

Centrosomal recruitment, distribution around the centrosome, and key interacting partners of the microtubule nucleator gamma-tubulin ring complex (γ -TuRC) that facilitate the former have been enigmatic and difficult to study. This impressive work by Qi Gao and colleagues from Schiebel and Pfeffer labs is a breakthrough, uncovering numerous structural determinants of the specific spatial arrangement of γ -TuRCs around centrosomes, in their native cellular context, in various stages of the cell cycle. It is an important, elaborate study of high quality and impact. The manuscript is well written, with a good, logical storyline, easy to follow despite the volume of work presented. The interpretation and discussion of the results are convincing and interesting.

We thank the reviewer for the detailed review and positive evaluation of our manuscript.

Minor corrections and suggestions

Abstract:

1) Line 25: not clear what the “ring” is in this context, as the nature of the γ -TuRC is not explained. Perhaps start the abstract with: “The γ -tubulin ring complex (γ TuRC)...”

We have adapted the abstract as suggested.

2) Lines 28 and 31: inconsistent spelling, but “luminal” is probably correct

We now use ‘lumenal’ throughout the text.

Introduction:

3) Line 51: why not develop CDK5 abbreviation, too

We have adapted the text as suggested.

4) Line 57: “bind” missing?

We have corrected the text as suggested.

5) Line 61: looks like too many “the”

We have corrected the text.

6) Line 69: would suggest a simple “show” instead of “decipher”

We have rephrased the text to: “We characterize both the structure and ...”.

Results:

7) Line 255: don’t need “for”

We have corrected the text as suggested.

8) Line 356: not explained what CHX does, so the rationale for the chase experiment may be unclear to some readers

We have provided the rationale for the CHX experiment in the text now: “This indicates that Augmin and γ -TuRC might be stabilized by localization inside centrioles, which is disturbed in *POC5*^{-/-} and *POC5*^{mut} cells. To test this notion further, we performed cycloheximide (CHX) chase experiments. In these experiments, CHX inhibits *de novo* protein synthesis on ribosomes, allowing us to monitor the degradation of pre-existing proteins.”

Methods:

9) Line 484: looks like “, a” should be after “SNAP”

We have corrected the text as suggested.

10) Line 537: “donor”

We have corrected the text.

11) Line 541: “were then”

We have corrected the text as suggested.

12) Line 565: “processed for electron microscopy as follows:”

We have corrected the text as suggested.

13) Line 598: delete “it”

We have corrected the text.

14) Line 601: “after three washes in PBS”

We have corrected the text.

15) Examples in lines 606 and 618: inconsistent naming of Fiji software - unify across the whole manuscript

We have corrected the text and unified the naming of Fiji software across the entire manuscript.

16) Line 609: “expressing”

We have corrected the text as suggested.

17) Line 620: colon after the heading, not found elsewhere

We have corrected the text.

18) Line 647: “. Runs were performed...”

The corresponding sentences has been rephrased and now reads: “. AEC was performed ...”.

19) Line 651: abbreviation “AEC” not found previously developed
We have expanded “AEC” at first use in the text.

20) Line 669: “After the washes”?
We have corrected the text.

21) Line 686: not explained what the proteinase inhibitor was - anything else on top of PMSF? Then need to specify
We now provide the exact composition of the proteinase inhibitor in the text (1 mM PMSF and 1 tablet of cOmplete Protease Inhibitor Cocktail tablet (Roche)).

22) Line 688: “with” missing? What were the proportions?
We have corrected the text as suggested and given the relative proportions.

23) Line 688: “After boiling”?
We have corrected the text: “After boiling and centrifugation, ...”.

24) Lines 695-696: “followed by three washes with PBS”?
We have clarified the protocol in the text: “Formaldehyde was quenched for 5 min in 100 mM NH₄Cl in PBS followed by three washes with PBS.”

25) Line 708: concentrations of glucose oxidase and catalase are missing
We have added the concentrations to the text.

26) Line 807: Would be good to add details about resolution filtering and normalisation, e.g. software
We now provide the missing information: “For identification of CM1 modules, cryo-EM reconstructions of γ -TuRCs from RPE1 wild-type and *CDK5RAP2*^{-/-} cells were low-pass-filtered to 30 Å using `reliion_image_handler`, histogram-scaled in ChimeraX¹⁻³ and subsequently superposed and subtracted from each other.”

27) Line 832: “second branch: -35”?
The second branch was started at a stage tilt of 38°. We have corrected the text.

28) Line 841: “200003-EMPIAR dataset”? Would be good to explain what that dataset represents
We modified the text as suggested: “Tilt images were scaled to an object pixel-size of 4.3 Å during import into Warp to match the publicly available dataset 200003-EMPIARC, which contains tilt-series of native HeLa cell centrosomes obtained by cellular cryo-ET.”

29) Examples in lines 864 and 878: Warp/WARP
We have corrected all instances in the text.

30) Line 882: suggest adding “name et al.” before reference numbers
We have corrected the text as suggested: “... using MATLAB script from Brandt et al. .”

31) Line 927: mention criteria for particle retainment/selection

We have extended the text: “Particle selection in both classification runs was based on the presence of density for the GRIP2 domains and γ -tubulins at spokes 5 and 6 in the respective classes.”

Density for the GRIP2 domains and γ -tubulins at spokes 5 and 6 were removed in the reference for auto-picking as well as 3D classification.

32) Lines 928, 931, 932...: was it 3D auto-refinement? Suggest precision here
Yes, particles were refined using 3D auto-refinement. We have corrected all instances in the method section.

33) Line 938: call out the table that provides the B factor and mention the software used for filtering
We have added a callout to Supplementary Table 2 and provided the missing information.

Figures:

34) Line 1325, Figure 6 legend: should “(red)” and “(green)” be swapped?
We have corrected the figure legend, taking the updated color scheme into account.

35) Extended Data Fig. 4: would be good to number the spokes, at least 13 and 14 in the panel I.
We have added full spoke numbering to the figure panel.

36) Line 1449: how filtered, in Relion, Chimera?
Reconstructions were local resolution filtered in RELION. We have added this to the figure legend.

37) Extended Data Fig. 5, panel G: the third “i” is missing in the “unidentified” label
We have corrected the figure labeling.

38) Line 1524-1525: again, would be good to mention how filtered and normalized
We have provided the requested information in the figure legend.

39) After Extended Data Fig. 10, figures are indexed as “S11” etc. (tables also: S1...)
We have unified Supplementary Figure titles.

Questions:

1) Based on the differential patterns of ring decoration with CM1 modules, correlated with the conformational rearrangements in the ring, can the authors comment if the activation model in which CDK5RAP2 induces the closure of the ring is likely to be correct? It seems difficult to reconcile with the lack of nascent MT nucleation events captured.

We thank the reviewer for this insightful question. Functional data clearly established a link between CDK5RAP2 CM1 motif binding and the activation of γ -TuRC microtubule nucleation activity^{4,5}. From a structural perspective, our study indicates that CM1 motif binding in the native context of centrosomes transitions γ -TuRC spokes 1-8 into a MT-compatible conformation, which should promote the function of the γ -TuRC as a structural template during the initial stages of MT nucleation. A similar conformational change was also observed in two recently published studies presenting cryo-EM structures of isolated γ -TuRCs that were decorated with recombinant CM1 motif^{6,7}. Thus, available functional and structural

data overall indicate that the CDK5RAP2 CM1 motif at least contributes to γ -TuRC activation by inducing partial ring closure.

However, we cannot exclude that additional factors are important for regulating the MT nucleation activity of γ -TuRCs in the cellular context. We therefore extended the respective discussion section: “This suggests that other interaction partners or post-translational modifications present only in the context of the centrosome participate in and finetune conformational regulation of the γ -TuRC by CDK5RAP2. This is also consistent with the absence of γ -TuRC-associated MTs in the PCM of cellular centrosomes, which suggests that additional regulatory layers beyond CDK5RAP2 CM1 motif binding may impact on the MT nucleation activity of γ -TuRCs in the cellular context”.

2) Why do the authors think purified centrosomes lose luminal γ -TuRCs? Are the identified tethers predicted to be weak? Intuitively, one would assume they might be somewhat protected from being stripped as opposed to γ -TuRCs exposed in the PC region, which in contrast appear to withstand the isolation procedure.

We apologize for this misunderstanding - we do not think that centrosomes lose luminal γ -TuRCs during purification. Luminal γ -TuRCs are just not well visible and only very rarely detected by template matching due to their location in the centriole lumen, where sample thickness is maximal and signal-to-noise ratio minimal. To prevent misunderstandings, we now explicitly explain this in the results section and show examples of luminal γ -TuRCs in Supplementary Figure 1b.

Reviewer #2 (Remarks to the Author)

The manuscript by Gao et al uses cryo-electron tomography and alphafold predictions to describe the structure and arrangement of the microtubule nucleator gamma-tubulin ring complex (gamma-TuRC) in situ at centrosomes in cells and at purified centrosomes. They focus on two different sub-populations associated with the pericentriolar material (PCM) and the centriole lumen, respectively. The authors define the structure of the gamma-TuRC in the PCM, where they observe two different conformations that are proposed to correlate with different interaction modes with the activator CDK5RAP2, and describe a density at the seam of the gamma-TuRC cone that likely corresponds to the adapter NEDD1. gamma-TuRCs are also present in the lumen of the centrioles, where the authors describe their ultrastructural organization, although the advance here over previous work is limited. They propose a protective function for luminal localization that prevents the degradation of augmin and gamma-TuRC during interphase and upon release during mitosis provides additional nucleation activity to boost spindle assembly.

The main novelty of the manuscript resides in that it is the first study of the structure and arrangement of the vertebrate centrosomal gamma-TuRCs in situ. Similar work was recently published, but for the budding yeast oligomeric gamma-TuSC bound to spindle pole bodies, a system that differs in several ways from vertebrate gamma-TuRC at centrosomes (Dendooven et al., 2024, Nat Struc Mol Biol).

It is important to highlight that the MT nucleation system in budding yeast is strongly simplified in terms of molecular complexity and regulatory networks as compared to the vertebrate system. Our study thus is unique in providing important insights into fundamental aspects of centrosomal recruitment, activation and organization of the γ -TuRC in humans. We

also would like to point out that the study mentioned by the reviewer fully focuses on structural analysis of the activated yeast γ -TuRC associated with a MT, while our manuscript goes well beyond a structure-focused view and addresses several aspects of vertebrate-specific recruitment and cell cycle-dependent molecular organization of the γ -TuRC.

The current work also describes the first structure of the binding interface between gamma-TuRC and its adapter NEDD1.

Overall, this is an important study that raises some interesting ideas and potentially controversial concepts. Unfortunately, many claims are overly ambitious and are not supported by the data. It seems that the manuscript tries to answer too many questions. It would have been useful to focus the manuscript on fewer of the various aspects of centrosomal gamma-TuRC structure and function...

In our manuscript, we provide a comprehensive structural view on γ -TuRC recruitment, activation and spatial organization at human centrosomes. All of these aspects are tightly interconnected and cannot be separated.

... interpret the data with more caution, and provide more mature analyses.

In our revised version of the manuscript, we have refined all aspects of our analyses suggested by Reviewer #2 and included extensive new experiments to strengthen our conclusions.

Regarding the text, the manuscript is well-written, but in both results and figure legends the descriptions are too brief and lack important information to understand what exactly was done. Also, the authors need to do a better job in citing and discussing previous studies.

While most methodological details requested by the Reviewer were already present in the methods section of the original manuscript version, we have now expanded the description of experiments in the figure legends and main text and included additional citations as suggested by the reviewer.

Major points:

1) In situ cryo-EM of gamma-TuRC: the obtained resolution of 16.4 Å may allow docking of the known subunit structures, but it is not sufficient, using only alphafold predictions and no additional experimental structures, to determine the identity of new densities. Thus, the claim “This model could be seamlessly docked as a rigid body into the unoccupied density segment capping the γ -TuRC cone (Fig. 1D), identifying the structure and binding site of NEDD1 on the γ -TuRC.” is an overinterpretation of the data.

We agree with the reviewer that rigid-body docking of the predicted NEDD1 grapple model into our subtomogram average may not unambiguously identify NEDD1 in the structure. We therefore removed this claim and reorganized the section to include extensive new experimental data that significantly strengthen our assignment of NEDD1 (see comment 2 below).

2) Related to point 1, the authors also obtained higher resolution maps that include the extra density for soluble γ TuRC. However, γ TuRC from extract is used, which is not of a defined composition. Again, the indicated resolution of 5-8 Å for the new density is not sufficient to

“unambiguously” identify part of this density as NEDD1, more caution is needed here. While the known MZT1 modules have a characteristic fold and may be docked, the part claimed to belong to NEDD1 is very small (2 consecutive helices) and could also belong to another protein. There is also the possibility that more than one protein could bind at this site by providing helices that engage with MZT modules. The authors could use a recombinant system, as in their previous work, to reconstitute and directly probe the structure of gamma-TuRC-bound NEDD1.

To address this comment, we have assembled a vast body of new experimental data that strongly support NEDD1 identification in our cryo-EM reconstructions:

- 1) We subjected the purified *X. laevis* γ -TuRC to mass spectrometry analysis (Supplementary Table 1, Supplementary Dataset 1). As expected, γ -TuRC components were the most abundant proteins in the sample (apart from trypsin). Notably, NEDD1 was detected at similar total intensities as γ -TuRC proteins, suggesting that it was co-purified with γ -TuRC at high stoichiometry. The only known γ -TuRC interactor with unclear binding site present among the top identified proteins in addition to NEDD1 was NME7 (Nucleoside diphosphate kinase 7). However, NME7 could be excluded to contribute to the grapple density based on its structure, which does not contain any coiled-coil segments (PDB: 7UNG⁸, PDB: 8J07⁹). Overall, this significantly strengthens our assignment of NEDD1 in the cryo-EM reconstruction. Please also see our response to Reviewer #3, minor comment 2.
- 2) By optimizing our cryo-EM image processing workflow, we could improve resolution of the NEDD1-containing grapple density segment to 4.3 Å global resolution, with local resolution for the NEDD1-derived coiled coil segment in the central region of the grapple ranging from 4-5 Å (Supplementary Fig. 5). This enabled us to identify a number of bulky aromatic amino acid side chains resolved in the cryo-EM reconstruction that are fully consistent with the sequence of the respective NEDD1 segments predicted to be part of the grapple by AlphaFold2, strongly supporting our identification of NEDD1 in the cryo-EM density (Fig. 2b). Moreover, the improved resolution further highlights the excellent match between the length of α -helices resolved in the density and predicted for the C-terminus of NEDD1 by AlphaFold2.
- 3) Aiming to characterize the overall structure and properties of the NEDD1-N-GCP3/MZT1 tetramer, we created a FLAG-tagged NEDD1-N-GCP3 fusion construct and co-expressed it in insect cells with MZT1, which was readily co-purified, suggesting successful complex formation (Supplementary Fig. 4).
 - a. Consistent with previous data for the isolated NEDD1 C-terminus^{10,11}, mass photometry indicated that NEDD1-N-GCP3/MZT1 formed stable tetramers, clearly supporting the oligomeric state of NEDD1 predicted by AlphaFold2.
 - b. Negative stain electron microscopy of NEDD1-N-GCP3/MZT1 complexes in conjunction with fully data-driven and reference-free particle averaging produced a 3D density highly similar to the γ -TuRC-associated grapple density, strongly supporting our identification of NEDD1 in the cryo-EM reconstruction.
- 4) To further support our model for molecular architecture of the NEDD1 grapple, we designed structure-guided point mutations in NEDD1 expected to disrupt either NEDD1 tetramerization (*NEDD1^{LLM}*) or binding of N-GCP3/MZT1 modules to NEDD1^C (*NEDD1^{VVI}*) (Fig. 2d). HA-tagged wild-type NEDD1 (*NEDD1^{WT}*) and HA-

tagged NEDD1 mutant variants were overexpressed in HEK293 cells and the interaction with γ -TuRC was probed by HA-pulldown. The γ -TuRC co-eluted with HA-NEDD1^{WT} but interaction was mostly lost for both NEDD1 mutant variants (Fig. 2d,e). This further strengthens our model for the NEDD1 grapnel architecture and supports our assignment of NEDD1 in the cryo-EM reconstructions.

Cumulatively, the additional data now included in the revised version of the manuscript sufficiently validate the assignment of NEDD1 in the cryo-EM reconstruction.

Importantly, in a recently published preprint¹², a highly similar γ -TuRC-associated grapnel density was observed only after co-expression of NEDD1 with the recombinant human γ -TuRC. Although at significantly lower resolution, their data fully support our identification of NEDD1 in our cryo-EM reconstruction.

3) Lines 150-160: Description of residues and interactions. How are the positions of these side chains determined? There does not seem to be sufficient resolution for such detailed analysis.

The side chain positions underlying the described residue interactions are partially based on our cryo-EM reconstruction and partially based on AlphaFold predictions of the interfaces. Importantly, our cryo-EM density validates these AlphaFold predictions on protein back-bone and partially side chain level, indicating high reliability of the predicted per-residue interactions. We now clarify this in the main text.

4) Identity of the MZT1 module in the extra density: apart from N-GCP3/MZT1 in the luminal bridge and N-GCP5/MZT1 in spoke 14, the cross correlation analysis in Table S1 shows minimal differences for MZT1 modules in the different densities of the current study.

Generally, differences in cross-correlation values are expected to be small, because the fold of N-GCP3/MZT1 and N-GCP5/MZT1 are very similar. Consistently, the differences observed when comparing MZT1 and MZT2 modules, which are structurally less similar, are larger (Supplementary Table 4).

Further, the magnitude of cross correlations and hence their differences is also dependent on local resolution of the density segments analyzed. Resolution for the luminal bridge and the N-GCP5/MZT1 module on spoke 14 in EMD-10491¹³ is higher as compared to the NEDD1 grapnel modules in our cryo-EM density, explaining larger differences in correlation values for the former modules.

While the differences for the NEDD1 grapnel modules are small, they still highly consistently distinguish between N-GCP3/MZT1 and N-GCP5/MZT1 for three out of four NEDD1 grapnel modules when using different atomic models.

Also, the legend states that “models with the highest cross correlation for each density are highlighted in bold”, but there is no consistent pattern underlying the bold highlights. In the density map of the current study 7QJD and AF2 correlate similarly well with the spoke 14 module (0.55-0.56; for some reason not highlighted in bold) as with the grapnel modules (0.53-0.57).

Because local resolution for many of the density segments varies, cross correlation values can only be compared column wise, i.e. different models correlated against the same cryo-EM density segment.

To improve accessibility of the table, we now color code cross correlation values for each column from minimum to maximum correlation values within each column and improve the separation of table entries between different cryo-EM density segments.

Have the authors tested cross-correlation with N-GCP2/MZT2 modules? They have a similar structure and should be included in the analysis.

We thank the reviewer for this suggestion. In the revised version of the manuscript, we have included a comparison with N-GCP2/MZT2 modules (Supplementary Table 4), in which we demonstrate that N-GCP2/MZT2 modules consistently show lower correlation with the density segments in our cryo-EM reconstruction as compared to N-GCP/MZT1 modules.

5) Inwards vs outwards conformation: as presented in the figures, it is not clear what the main difference between the conformations is. It seems that “inwards” refers mainly to the last gamma-TuSC at pos. 13/14. Positions 11/12 of the same conformation actually appear more “outwards” (more elliptical; seen in Fig. 3A, B, C).

This is correct and was suggested in the main text of the original manuscript version already. However, we now edited the text to emphasize this even better:

“... and spokes 13 and 14 are positioned outwards from the helical axis (‘outwards conformation’ of spokes 13 and 14)”.

“... and spokes 13 and 14 move slightly inward towards the helical axis of the γ -TuRC (‘inwards conformation’ of spokes 13 and 14).”

Related to this, there is no information in the results or figure legend, how these differences were quantified in Fig. 3. How are alignments done, what residues are used for measurements, etc.

We have included the information requested by the Reviewer into the figure legend: “Deviation of γ -TuRC geometry from the fully closed MT-capping γ -TuRC (PDB: 8VRK). The inwards and outwards conformations of PCM-located human γ -TuRCs, as well as purified recombinant (PDB: 7AS4) and native human γ -TuRCs (PDB: 6V6S) were analyzed. All models were superposed according to spokes 2-8 and deviations were calculated based on the γ -tubulin centers determined in ChimeraX using the “measure center” command.”

It would also be useful to plot and compare the measurements for recently published MT end bound gamma-TuRC, which is in a more circular, closed conformation.

We thank Reviewer #2 for this suggestion. We now compare all γ -TuRC conformations to the recently published structure of the fully closed MT-capping γ -TuRC from Aher et al ¹⁴, which highlights the similarity of γ -TuRC spokes 1-8 in the ‘inwards conformation’ with the MT-nucleating conformation of the γ -TuRC.

6) Extra densities assigned to CM1 around the gamma-TuRC: the authors claim that these

match “the position and shape of CM1/N-GCP2/MZT2 modules”. I understand that this interpretation makes sense based on the recently posted bioRxiv study by the Wieczorek group that is cited in the introduction, but the data presented in the current study should stand for themselves.

Here, we have to respectfully disagree with Reviewer #2. A fundamental principle in science is to achieve incremental progress by building upon and further developing the work of others.

The pre-print mentioned by the reviewer has now been published back-to-back with a second study ^{6,7}. Both studies characterize the structure of recombinant CDK5RAP2 CM1 motif bound to all five GCP2 subunits of the γ -TuRC and the associated conformational changes by cryo-EM ^{6,7}. These two independent studies are highly consistent with our data and thus significantly strengthen the assignment of CM1 modules in our subtomogram averages.

To further support the assignment, we followed the Reviewer’s suggestions as detailed below.

The low resolution extra densities in panel 3D do not support this claim.

Aiming to improve resolution of the GCP2-associated extra density, we took advantage of the presence of multiple GCP2 subunits per γ -TuRC to increase the number of asymmetric units for subtomogram averaging and classification. This allowed us to improve resolution of GCP2 and the associated extra density from 23 Å to now 18 Å. Furthermore, we could achieve higher stoichiometry of the extra density by subtomogram classification.

In the improved subtomogram average, the position and shape of the extra density is fully explained by the structure of CM1 modules on GCP2 recently solved by cryo-EM ^{6,7,15,16} (Figure 3d, Supplementary Fig. 9a). Moreover, cryo-EM density projecting downwards from the CM1 module towards the cone of the γ -TuRC is highly consistent with the position of extended dimeric coiled-coil segments of CDK5RAP2 C-terminal to the CM1 motif as predicted by AlphaFold3 (Supplementary Fig. 9a).

In combination with our observation that the GCP2-associated extra densities are strictly dependent on the presence of CDK5RAP2, this sufficiently validates the assignment of CM1 modules in the subtomogram averages.

Apart from this, it is also unclear how the subtraction was done, since the differential densities do not seem to correspond to the extra densities seen in the individual maps. Perhaps displaying superposed densities would help.

In response to this comment, we have strongly improved visualization for the comparison of γ -TuRC inwards and outwards conformations. We now show both cryo-EM densities side-by-side, in superposition and with difference densities superposed for all five GCP2 subunits of the γ -TuRC individually (Supplementary Fig. 11a). Moreover, we superpose the atomic model of GCP2-associated CM1 modules in all comparisons to highlight the fit for each of the spokes. This confirms a conformation-dependent pattern of CM1 module binding to the five different GCP2 subunits of the γ -TuRC.

Also, apart from panel 3D, which shows only two examples, I cannot find data analysing all positions and which positions are occupied simultaneously to support this statement: “In the ‘outwards’ conformation, density for CM1 modules could be detected exclusively on spoke

13, as described before. In contrast, in the ‘inwards’ conformation, the CM1 module binding site on GCP2 at spoke 13 was vacant and the γ -TuRC harbored CM1 modules on the remaining GCP2 subunits at spokes 1, 3, 5 and 7.”

We now present a comparison for all five individual GCP2 subunits in Supplementary Fig. 11a.

7) While analysis of centrosomes from CDK5RAP2 KO cells is a good experiment to identify CDK5RAP2-dependent densities, this does not identify these as CM1 densities. Again, it makes sense in the light of results from studies by others, but the data provided in the current study do not prove this.

We fully agree with the Reviewer that this experiment identifies the extra densities as CDK5RAP2-dependent. Taking into account that these CDK5RAP2-dependent densities are highly consistent with the position and structure of CDK5RAP2 CM1 modules as observed by high-resolution cryo-EM^{6,7,15,16} (see Reviewer #2, comment 6), it is highly likely that they correspond to the CDK5RAP2-derived CM1 modules.

Also, similar to panel 3D, the Extended Data Fig. 7 shows differential densities that are not seen in the individual maps. Is this due to differential thresholding? If so, this needs to be adjusted. If I understand the minimal description well, the red densities should correspond to the extra densities seen in the green maps, which does not seem to be the case.

To clarify this comment, we have strongly improved visualization for the comparison of γ -TuRCs from wildtype and *CDK5RAP2* KO RPE1 cells. We now show both cryo-EM densities side-by-side, in superposition and with difference densities superposed for all five GCP2 subunits of the γ -TuRC individually (Supplementary Fig. 9e). Moreover, we superpose the atomic model of GCP2-associated CM1 modules in all comparisons to highlight the fit for each of the spokes. This confirms that the GCP2-associated extra densities are strictly dependent on the presence of CDK5RAP2.

8) Spatial organization of gamma-TuRC relative to centrioles: how many centrioles were analyzed for this and how many gamma-TuRCs per centriole? It would be important for the reader to know, whether the authors present some fraction of all gamma-TuRCs present in a centrosome or a comprehensive mapping of all gamma-TuRCs (within the limits of the method).

As described in the methods section of our manuscript, we used all γ -TuRCs detected in the tomograms for spatial organization analysis, unless the orientation of its paired MTT segment was incorrectly determined during 3D refinement. This was judged based on visual inspection of MTT particle orientations and their consistency with the interpolated MTT axis on a per tomogram level in UCSF Chimera. For the final analysis shown in Supplementary Fig. 14, we retained 232 γ -TuRCs and their nearest MTT-segments from 15 individual centrioles distributed over 12 tomograms. We now added this to the figure legend.

9) POC5-augmin interaction: I cannot find evidence in the presented experiments for the claimed direct interaction between POC5 and augmin. This should be phrased and interpreted with more caution.

To address this comment, we aimed to show direct interaction between recombinantly expressed and purified POC5 (insect cells) and the Augmin TII N-clamp (*E. coli*). POC5 was

co-expressed with centrin 2, which improves POC5 stability and solubility¹⁷. The purified proteins were incubated *in vitro* and their interaction was tested by co-IP experiments, in which we could observe that Augmin TII N-clamp can pulldown POC5 (Supplementary Fig. 16). This experiment indicates a direct interaction between the two components.

10) Analysis of POC5 and POC5mut cells:

a) regular wide field imaging is not suited to quantify luminal pools of POC5, augmin or g-tub. This should be done by super resolution imaging.

In response to this comment, we analyzed *POC5* WT and *POC5^{mut}* cells by ultrastructure expansion microscopy (U-ExM), which represent a special type of super resolution microscopy and is a widely accepted method for analyzing the distribution of centriole proteins¹⁸. We now show that in *POC5^{mut}* cells, γ -tubulin and HAUS4 are strongly reduced in the centriole lumen, but still present in the PCM (Supplementary Fig. 17h,i). In contrast, POC5 and POC5^{mut} show similar localization inside centrioles for both cell types (Supplementary Fig. 17f).

b) it is not explained how exogenous POC5 and POC5mut are expressed in POC5 KO cells, since they seem to be controlled by an inducible promoter. When is expression induced and what are the expression levels relative to endogenous?

We have included information on the expression of *POC5* and *POC5^{mut}* into the methods section and now show by Western blotting that expression levels are comparable to endogenous POC5 (Supplementary Fig. 17a).

11) The current quality of the UExM is not very convincing. In panel S13C, exogenous POC5 and in particular the POC5mut clearly do not localize robustly along the length of the central inner wall as seen for endogenous POC5. Signals appear restricted to a smaller region and to more distal parts. The same is true for HAUS4. In full length centrioles the central scaffold lines the central region but does not extend to the distal end (see work by Guichard/Hamel group).

In the revised manuscript version, we have provided better U-ExM images, in which localization of exogenous POC5 and POC5^{mut} along the length of the centriole is comparable to endogenous POC5 (Supplementary Fig. 17f). Similarly, we now provide U-ExM images, in which the centriole localization of HAUS4 is comparable between cells with exogenous and endogenous POC5.

12) What is the relevance of the centriole length defect observed for POC5 KO cells? Previous work using POC5 RNAi has observed not only length defects, but also structural defects including broken centrioles and should be referenced in this context. (Azimzadeh et al., 2009, JCB; Steib et al., 2020, eLife). Do the authors also see this here and are all defects rescued by both WT and mutant POC5? See also comment on mitotic defects below.

In response to this comment, we have quantified defects in centriole length and integrity in control and *POC5^{-/-}* cells, as well as in *POC5^{-/-}* cells after rescue with POC5 WT and *POC5^{mut}* using U-ExM (Supplementary Fig. 17g). In *POC5^{-/-}* cells, both defects were enriched as expected. After *POC5* WT or *POC5^{mut}* rescue, phenotypes were comparable to control cells, suggesting that the stabilizing inner scaffold remained functional in *POC5^{mut}* centrioles.

13) Stability of luminal augmin-gamma-TuRC:

a) The FRAP experiment was done only for about 2 min, which is not a relevant time scale related to cell cycle dependent turnover during interphase. Much longer times should be tested to assess turnover rates (CHX chase in 6F was done for 10 hours)

We respectfully disagree with Reviewer #2 in this respect. We are not comparing FRAP experiments to our CHX chase experiments. FRAP experiments are typically done for short periods of time (1 -1000 seconds) to assess the short-term dynamics of protein localization¹⁹ and for instance allow to distinguish between freely diffusible, phase-separated and stably bound protein pools. CHX chase experiments typically report on the cellular stability of proteins over longer time windows (min - hours).

To address the Reviewer's comment, we nevertheless extended our FRAP experiment to 30 min (Supplementary Fig. 19m), which was very challenging, because both the cellular component of interest and the whole cell itself are moving during live-cell imaging²⁰, but could be achieved by imaging larger volumes (z-stacks). Even over this very extended period of time, no recovery of signal could be detected, suggesting there is no constant exchange of centriole luminal Augmin with cytosolic pools in interphase.

b) The destabilization in the absence of POC5 does not necessarily indicate stabilization in the lumen, since there is likely a cytosolic fraction of POC5. Do the authors have evidence that POC5 is associated with augmin-gamma-TuRC only in the centriole lumen?

In the experiments mentioned by the Reviewer, we analyzed interphase cells. During interphase, cytosolic Augmin is inactivated by binding to importin and becomes active only with nuclear envelope breakdown²¹. Whether or not inactive Augmin can still interact with cytosolic fractions of POC5 is unknown and we do not have any evidence for whether or not a putative cytosolic Augmin-POC5 complex may have a function during interphase.

We have included a sentence to the discussion section to address this comment: "Augmin and the γ -TuRC are likely shielded from degradation in the centriole lumen during interphase, as indicated by CHX chase experiments. However, we cannot entirely exclude the possibility that POC5 may also contribute to stabilizing γ -TuRC and Augmin outside centrioles."

14) The sub-centrosomal localization of HAUS1-Strep is unclear. The quantification in Fig. 6 needs to be done with super resolution imaging to reveal the specific luminal signal.

To address this comment, we have included U-ExM data demonstrating that HAUS1-SNAP-S is localized to the centriole lumen and absent from the PCM in interphase cells (Supplementary Fig. 19l). Consistently, MINFLUX microscopy also indicates that centrosomal HAUS1-SNAP-S is exclusively localized to the centriole lumen in interphase (Fig. 5f). Cumulatively, this suggests that centrosomal HAUS1-SNAP-S signal detected in the now improved fluorescence image data in Fig. 6a predominantly originates from centriole-luminal HAUS1.

For further quantification of Augmin, γ -TuRC and the inner scaffold component POC5 inside centrioles, interphase and STLC arrested prometaphase cells were analyzed by u-ExM (Supplementary Fig. 21). These data indicate strongly diminished inner centriole localization of Augmin and γ -TuRC in mitosis, while POC5 was not affected. These super resolution data are fully consistent with Fig. 6a and its quantification in Fig. 6b.

15) The idea of simultaneous decoration of gamma-TuRC with NEDD1 and CM1 of CDK5RAP2 is interesting, but contrasts with previous studies. NEDD1 has not been observed in CM1/CDK5RAP2-bound gamma-TuRC and vice versa (Choi et al, 2010, JCB; Muroyama et al, 2016, JCB) and both adapters were shown to form distinct complexes not only in keratinocytes (Muroyama et al, 2016). Considering that the simultaneous binding cannot be unambiguously identified at centrosomes in cells (see above), the authors should add biochemical proof of this simultaneous interaction. Also, the current discussion of the previous work regarding this point is too brief.

Both studies mentioned by Reviewer #2 mostly rely on purification of complexes either via CDK5RAP2 or NEDD1. CDK5RAP2 may largely dissociate during isolation of γ -TuRCs via NEDD1, and vice versa, which could be interpreted as indication for the presence of two compositionally distinct populations of γ -TuRCs. In particular, CDK5RAP2 is a protein stably integrated into the PCM and therefore unlikely to co-purify at high stoichiometry with γ -TuRCs unless used as a target for co-IP.

In contrast, we here present the first study to analyze binding of NEDD1 and CDK5RAP2 to γ -TuRCs without purification and in the native context of centrosomes. In the revised version of the manuscript, we have further strengthened the identification of NEDD1 and CDK5RAP2-derived CM1 modules in the subtomogram averages. While CM1 modules are present substoichiometrically, NEDD1 is a stoichiometric component of the γ -TuRC, implying that both γ -TuRC interacting proteins can bind at the same time.

To address this comment and to explicitly demonstrate simultaneous binding of CDK5RAP2 and NEDD1 to γ -TuRCs, we selected the set of γ -TuSC units observed to contain CM1 modules using subtomogram classification (Supplementary Fig. 2e), identified the corresponding γ -TuRC particles and averaged them to obtain a cryo-EM density of γ -TuRCs associated with at least one CM1 module (Supplementary Fig. 11b). As expected, NEDD1 density was present at high significance level in the resulting reconstruction, confirming that CDK5RAP2 and NEDD1 simultaneously bind γ -TuRCs at centrosomes.

Fully consistent with our data, a recently published preprint by the Wieczorek lab¹² demonstrated that recombinant NEDD1-bound γ -TuRCs can be decorated with CDK5RAP2-derived CM1 motif (residues 44-93) using cryo-EM structural analysis. The authors of the preprint also highlight that the binding sites for NEDD1 and CDK5RAP2 CM1 motifs are well separated on the γ -TuRC and that there are no indications for mutual steric exclusion.

As suggested by the Reviewer, we extended the discussion section:

“Our structural data on γ -TuRCs embedded into the native PCM indicate that CDK5RAP2 and NEDD1 can simultaneously bind to γ -TuRCs at centrosomes and do not define distinct pools of γ -TuRCs, as was previously suggested in studies using purified γ -TuRCs^{4,22}. This discrepancy may stem from the limitations of biochemical purifications, in which CDK5RAP2 may have dissociated during γ -TuRC purification via NEDD1, and vice versa.”

16) For all CRISPR-edited cells lines, homo/heterozygosity testing by PCR and sequencing needs to be provided. Deletion should also be confirmed by western blotting. For all tagged proteins (endogenous or exogenous) western blotting needs to be provided to confirm the size and expression level. In addition, correct subcellular localization needs to be confirmed by immunofluorescence microscopy. Super resolution is needed to confirm luminal vs PCM

localization. Regarding tagging of augmin subunits, which are vital for mitosis, it should be confirmed that these cells cycle and divide normally (normal mitotic index, mitotic figures and proliferation).

We have performed PCR testing (Supplementary Fig. 19b) and DNA sequencing (Supplementary Fig. 19c) for the *HAUS1-SNAP-S* RPE1 cell line. In addition, as suggested by Reviewer #2, we have analyzed *HAUS1-SNAP-S* cells in greater detail and compared them with the corresponding *HAUS1* RPE1 cells. In Supplementary Fig. 19d, we show equal expression of HAUS1-SNAP-S and endogenous HAUS1 by immunoblotting. We further show that the mitotic cell number for *HAUS1-SNAP-S* is not increased compared to wild type (Supplementary Fig. 19f). HAUS1-SNAP-S also does not have an impact on the levels of mitotic regulators/signals Aurora A kinase, CDK1 and pH3 phosphorylation (Supplementary Fig. 19g-j). Furthermore, the established proliferation marker Ki67 is not affected by HAUS1-SNAP-S (Supplementary Fig. 19k). In mitosis, HAUS1-SNAP-S localizes to spindle poles and the spindle microtubules as published for GFP-tagged HAUS subunits or using HAUS antibodies (Supplementary Fig. 19e)²³. U-ExM analysis further confirmed centriole-luminal localization in interphase (Supplementary Fig. 19l). Cumulatively, this strongly indicates that SNAP-S tagging of HAUS1 does not impact the expression and function of the HAUS1 protein.

The *POC5*^{-/-} RPE1 cell line was recently published by Sala et al.¹⁷. The publication contains the analyses requested by Reviewer #2. The *POC5*^{mut} cell line is derived from the *POC5*^{-/-} cell line without CRISPR/Cas9 editing.

17) In Fig. 6A the HAUS1-Strep spindle staining is very weak. One would expect robust localization to spindle MTs (Lawo et al, 2009) including some augmin signal coming from the midzone area at later stages (Uehara et al, 2009). Do these cells undergo normal mitosis and do other augmin antibodies show proper spindle staining in these cells?

The localization of Augmin along spindle microtubules has been frequently studied with tagged and overexpressed HAUS subunits (see Uehara et al, 2009²³, Fig. 5d as an example), because detection of endogenous HAUS subunits with antibodies is challenging. This is why we were following a similar tagging strategy and used HAUS1-SNAP-S in some of the experiments. In contrast to these published Augmin localization studies, *HAUS1-SNAP-S* is endogenously tagged and is not overexpressed.

As suggested by Reviewer #2, we now show by STED microscopy that HAUS1-SNAP-S indeed localizes along spindle microtubules and to the spindle poles (Supplementary Fig. 19e). Moreover, we strongly improved the fluorescence image data on HAUS1-SNAP-S in Fig. 6a. The spindle staining is clearly visible now. Taken together, we have strongly improved the microscopic analysis of Augmin localization on spindle microtubules, spindle poles and centrioles as requested by Reviewer #2.

In our response to Reviewer #2, comment 16, we also demonstrate that cells with HAUS1-SNAP-S expression undergo normal mitosis.

18) In Fig. 6E→ the authors show a decrease in the total levels of augmin and gamma-tubulin in the absence of POC5 and attribute this to destabilization of the luminal pool. However, previous studies have shown that only a very small fraction of the total cellular gamma-TuRC is present at the centrosome (including both luminal and PCM pools), and that the large

majority is in the cytosolic fraction, ranging from 80% (Moudjou et al, 1996) to 99% (Bauer et al, 2016).

We thank Reviewer #2 for bringing up this interesting point. Two manuscripts estimated the cytoplasmic-to-centrosome γ -tubulin ratio in cells.

1) Based on GFP quantification analysis, Bauer et al.²⁴ come to the conclusion that 99% of γ -tubulin-GFP resides in the cytoplasm. However, Bauer et al. use a heterozygous γ -tubulin-GFP/ γ -tubulin RPE1 cell line for quantification (this was before CRISPR/Cas9 was widely used and the authors used rAAV-mediated gene targeting), without testing whether γ -tubulin-GFP is functional. While studying *CDC31-GFP/CDC31* yeast cells, we have experienced that Cdc31-GFP does not localize to nuclear pore complexes as long as wild type Cdc31 is expressed²⁵. Only when wild type *CDC31* was removed, Cdc31-GFP localized to nuclear pore complexes, indicating that the affinity of Cdc31 for nuclear pore complexes is much higher than that of Cdc31-GFP. In conclusion, 1% of GFP-tagged γ -tubulin may be present at centrosomes, but this number may not reflect the localization of untagged wild type γ -tubulin.

2) Moudjou et al.²⁶ fractionate KE37 cells grown in suspension. They conclude that 80% of γ -tubulin is cytoplasmic based on immunoblotting data, but state that this is “a **crude estimate**” only. The quantification is based on one experiment without statistics and therefore should indeed be considered at best as a rough estimation. Still, even a 20-40% centriole localization is probably sufficient to explain the stabilizing effect that we observe in *POC5* wild type cells. Furthermore, KE37 cells are quite different from the RPE1 cells that we use in our study and centriole localization in RPE1 cells may be even higher.

If only the luminal pool is stabilized, one would not expect a strong decrease in the total levels. The authors should test what the proportion of luminal augmin and gamma-TuRC relative to the cytosolic pool is in their case, to see if this is compatible with their model.

As outlined above, determining the exact ratio of cytoplasmic versus centrosomal γ -tubulin is challenging and beyond the scope of this manuscript. However, the experimental data are clear: γ -tubulin and HAUS4 are more stable in control and *POC5* cells, while they are less stable in *POC5*^{mut} cells (Fig. 6g), which is accompanied by strongly reduced localization to the centriole lumen. This difference in stability also has an impact on the steady state levels. The most straight forward explanation is that localization of Augmin and γ -TuRC inside centrioles increases the stability of these proteins because they are shielded from the cytoplasmic degradation machinery.

However, we agree with Reviewer #2 that we cannot exclude an additional stabilizing function of cytoplasmic *POC5*. We therefore now mention this possibility in the discussion: “However, we cannot entirely exclude the possibility that *POC5* may also contribute to stabilizing γ -TuRC and Augmin outside centrioles.”

19) The *POC5* KO background produces mitotic defects in most of the cells according to Fig. 6I. However, the nature of the mitotic defects is unclear. Since *POC5* RNAi was shown to produce structurally aberrant centrioles (Steib et al., 2020, Elife), which may interfere with mitosis, the authors should quantify centriole number and configuration using a centriole marker such as centrin and test for structural abnormalities by UExM. The same should be performed for the rescue lines expressing wild type *POC5* and mutant *POC5*. If centrioles are abnormal in *POC5* mutant cells (due to a defective inner scaffold), this would readily explain

the mitotic defects.

We now show in Supplementary Fig. 17g that centrioles are not fragmented in *POC5^{mut}* cells in contrast to *POC5^{-/-}* cells. Similarly, expression of *POC5* and *POC5^{mut}* rescues centriole number compared to *POC5^{-/-}* cells (Supplementary Fig. 17d). This indicates that the inner scaffold is intact and functional in *POC5^{mut}* centrioles.

What we see during mitosis in *POC5^{mut}* cells is an increase in mitotic γ -tubulin centers and an increase in metaphase cells with misaligned chromosomes (Fig. 6h,i). *POC5^{mut}* cells in comparison to *POC5* wild-type cells show γ -tubulin and augmin release from interphase centrioles and overall less cellular γ -tubulin and augmin. Taken these observations together, it is sensible to propose the model that the γ -tubulin and augmin localization inside interphase centrioles and their timely release in mitosis are important to prevent mitotic defects.

20) In Fig. 6F, the use of *POC5* KO cells with induced *POC5* rescue construct would make more sense as a control. This would confirm that destabilization happens because of the deletion of *POC5* and not as a result of a secondary effect of clonal selection.

We agree with Reviewer #2 and have repeated the experiment with cells expressing *POC5* and *POC5^{mut}* (Fig. 6f). We observed the same destabilization of HAUS4 and γ -TuRC as with *POC5^{-/-}* cells, indicating a specific effect of perturbed Augmin- γ -TuRC interaction.

Minor points:

-Line 59: “Biochemical studies have revealed that the key adapter protein NEDD1 co-purifies with the γ -TuRC.” References are needed here.

We have included references as suggested.

-Line 47: Several references of the multiple previous cryo-EM structures are missing here.

We have included references as suggested.

-Line 54: Several additional studies should be cited here: Zhu et al., 2008, Curr Biol; Gomez-Ferrera et al., 2007, Curr Biol; Luders et al., 2006, NCB; O’Rourke et al., 2014, Plos One; Chinen et al., 2021, JCB; Gavilan et al., 2018, EMBO Rep,...

We have included references as suggested.

-The colors of the immunofluorescence images that are red and green could be changed to color-blind friendly ones.

We have changed the immunofluorescence images to a color-blind friendly combination of colors.

-Although this is the first article to describe structural aspects of the luminal gamma-TuRCs, luminal gamma-TuRC has already been studied previously not only in Schweizer et al, but also a more recent paper (Laporte et al, 2024). Both papers should be cited and discussed, since their findings differ in some aspects from the current manuscript. Also, numerous previous studies showing luminal localization of gamma-TuRC have not been cited at all (just to name a few: Fuller et al. 1995, Curr Biol; Moudjou et al. 1996, JCS; Sonnen et al., 2012, Biol Open; Lawo et al., 2012, NCB).

We have included the suggested references and discuss the studies of Schweizer et al and Laporte et al. in the discussion section.

-In line 56 references 11 and 12 are cited for the luminal organization of POC5-Augmin-gamma-tubulin, but this is not discussed in these articles.

We now cite only Schweizer et al in this context.

-In line 57, a word is missing

We have corrected the text.

-In the Extended Data Fig. 7C the numbers in the graph are missing, and the number of cells quantified are also missing from the figure legend.

We now provide the p-value in the figure and the number of cells analyzed in the figure legend.

-Extended Data Fig. 7D, the density that is suggested to be NEDD1 should be labeled to understand the figure more easily.

We have labeled NEDD1 in the figure panel as suggested.

-In Extended Data Fig. 13B, HA intensity, changing the left axis of the graph would facilitate the visualization of the data. Is it normally distributed? With this format it's difficult to see the distribution, and everything looks close to 0.

We have adjusted the Y-axis to better visualize the distribution of data points.

-There are inconsistencies with the naming of the supplementary figures, some of them are named "Extended Data", and some of them just "Figure S...".

We now name all supplementary figures consistently.

Our previous comment:

19) The POC5 KO background produces mitotic defects in most of the cells according to Fig. 6I. However, the nature of the mitotic defects is unclear. Since POC5 RNAi was shown to produce structurally aberrant centrioles (Steib et al., 2020, Elife), which may interfere with mitosis, the authors should quantify centriole number and configuration using a centriole marker such as centrin and test for structural abnormalities by UExM. The same should be performed for the rescue lines expressing wild type POC5 and mutant POC5. If centrioles are abnormal in POC5 mutant cells (due to a defective inner scaffold), this would readily explain the mitotic defects.

Author reply:

We now show in Supplementary Fig. 17g that centrioles are not fragmented in POC5mut cells in contrast to POC5^{-/-} cells. Similarly, expression of POC5 and POC5mut rescues centriole number compared to POC5^{-/-} cells (Supplementary Fig. 17d). This indicates that the inner scaffold is intact and functional in POC5mut centrioles.

Our new comment:

Supp Fig 17D: The description in the results refers to centrioles, but g-tub is quantified here, which does not allow counting of centrioles. In the plot, what are "defined" gtub foci? Are there additional "undefined" foci?

A centriole marker should be used (ideally centrin), as asked in the first review round, to confirm that there are no abnormalities in centriole numbers/configurations in POC5 vs POC5 mut.

Most importantly, centrin staining should be done for the mitotic figures in Fig 6I that we refer to in our original comment and where the extra g-tub foci are observed.

We agree that in specific cases γ -tubulin signal may not be suited to quantify the number of centrioles/centrosomes, in particular when multiple γ -tubulin foci may reflect PCM fragmentation. However, in our case, we see exactly two γ -tubulin foci in most POC5 and POC5mut cells (Supplementary Fig. 17d), thus there is no sign of PCM fragmentation in these cells.

In addition, we observe almost quantitative co-localization between γ -tubulin and POC5-HA/POC5mut-HA (new addition to Supplementary Fig. 17d), which localize to centrioles like wild-type POC5 in U-ExM (Supplementary Fig. 17f). POC5 is a bona-fide centriole component that further binds to centrin and therefore the POC5-HA/POC5mut-HA signal, which overlaps with the γ -tubulin signal (new Supplementary Fig. 17d), reflects to a large extent the localization of centrin and can be considered a centriole marker itself.

Thus, based on the combination of γ -tubulin and POC5 as markers, we can conclude that centrioles are not amplified or fragmented in POC5mut cells.

To clarify this point, we have added additional examples of POC5mut cells to Supplementary Fig. 17b and quantified co-localization of the γ -tubulin and POC5-HA/POC5mut-HA signals in Supplementary Fig. 17d.

We further removed 'defined' from the figure legend accordingly.

The authors should explain how they selected the centrioles for U-ExM and quantifications in Supp Fig 17 (since only isolated centrioles are shown). This is important because the results will depend on the cell cycle stage, whether daughter or mother centrioles are analysed, etc.

In Supplementary Fig. 17f-i, we analyzed mother centrioles of interphase cells (daughter centrioles were not quantified). Most centrioles were in G1 phase (XX%) and S/G2 phase (XX%; attached daughter centriole). Importantly, POC5 and POC5mut G1 centrioles did not show signs of fragmentation (Supplementary Fig. 17g), which strongly suggests that centrioles were intact during mitosis.

To further clarify this point, we will include a figure in the rebuttal letter, in which we compare centriole length and fragmentation in G1 cells (only one centriole) and G2/S cells (mother centrioles with attached daughter), as in Supplementary Fig. 17g. We observed no difference between the two cell populations.

The authors should provide example images for the centrioles quantified in Supp Fig 17G (centriole length, % broken centrioles). Also, the graph axis label says broken "centrosomes", this should be "centrioles" (I assume).

We have included example images for the centrioles quantified in Supplementary Fig. 17g and corrected the axis labeling as suggested.

Author reply:

What we see during mitosis in POC5mut cells is an increase in mitotic γ -tubulin centers and an increase in metaphase cells with misaligned chromosomes (Fig. 6h,i). POC5mut cells in comparison to POC5 wild-type cells show γ -tubulin and augmin release from interphase centrioles and overall less cellular γ -tubulin and augmin. Taken these observations together, it is sensible to propose the model that the γ -tubulin and augmin localization inside interphase centrioles and their timely release in mitosis are important to prevent mitotic defects.

Our new comment:

Assuming there are no centriole defects (see above) and since the total levels of augmin and γ -tubulin are reduced, an alternative explanation would be that the authors see an augmin depletion phenotype for POC5mut cells, which also causes spindle pole fragmentation (see Lawo et al 2009 Curr Biol). If this cannot be excluded, it should be discussed.

In response to this comment, we have included a short discussion section:

POC5mut cells exhibit defects in metaphase chromosome alignment. As centriole fragmentation was not observed in POC5mut cells with an intact inner POC5 scaffold, this defect is likely attributable to reduced levels of γ -tubulin and augmin. However, we cannot entirely rule out the possibility that subtle centriole aberrations contribute to this phenotype.

Reviewer #3 (Remarks to the Author)

γ -TuRC is a vital part of the cell's machinery, essential for templating microtubule formation including at centrosomes. In this manuscript, Gao and colleagues use various advanced microscopy techniques to analyze γ -TuRC distribution and structure in human cells and purified centrosomes.

They begin by using cryo-ET to visualize γ -TuRC in the pericentriolar material (PCM) of centrosomes purified from human KE-37 cells. By applying subtomogram averaging to the γ -TuRC particles, they identify additional density resembling a grapnel bound near the base of the γ -TuRC cone, which they tentatively assign to a tetramer of the C-terminus of NEDD1. In support of this assignment, they purified γ -TuRC from *X. laevis* egg extract, where it is known to copurify with NEDD1, and performed single-particle analysis cryo-EM. The structure has improved density for the grapnel (~4-8 Å resolution), allowing more confident (but not conclusive) docking of an AlphaFold2 prediction of a NEDD1C/N-GCP3/MZT1 tetramer.

Next, they performed classification of the PCM γ -TuRCs and identified two different conformations – outwards and inwards – each associated with additional density likely corresponding to CM1 motifs. A hypothesis that these CM1 motifs came from CDK5RAP2 was supported by cryo-ET of centrosomal γ -TuRC purified from wild-type and CDK5RAP2^{-/-} RPE1 cells. Assuming their assignments are correct, this would suggest that, contrary to a prior report, CDK5RAP2 and NEDD1 can bind γ -TuRC simultaneously.

Purification of centrosomes could influence PCM organization, so the authors next turned to in-situ cryo-ET of centrosomes in FIB-milled human HCT116 cells. They observed two pools of γ -TuRC – one in the PCM and one in the centriolar lumen, as expected from prior

expansion microscopy studies. They demonstrate that γ -TuRCs within the centriolar lumen are preferentially orientated with spokes 9-12 towards the centriolar wall.

Next, the authors speculated that γ -TuRC could be anchored via the NEDD1C/N-GCP3/MZT1 module to the centriole by Augmin. In support of this hypothesis, they used AlphaFold predictions, and MINIFLUX to show that Augmin subunit HAUS1 has a similar distribution to luminal γ -TuRCs. They made a mutant of POC5, designed to disrupt its interface with Augmin, and showed using immunofluorescence microscopy that the localization of γ -TuRC and Augmin to the centriole lumen was reduced but not abolished.

They next examined the cell-cycle-dependent localization of the different proteins using various types of fluorescence microscopy. They find that HAUS4 (Augmin subunit), GCP4 (γ -TuRC) and NEDD1 are lost at the onset of M phase and that the signal decrease of Augmin is partially blocked by addition of PLK1 inhibitor, suggesting release from the centriole is mediated by PLK1 activity.

In summary, this paper uses cutting-edge methods to explore γ -TuRC distribution and structure at centrosomes making a number of novel observations that should be of interest within the field. The findings are well-presented, and the methods section and extended data are comprehensive. However, as mentioned below, further information is required to fully support some of the conclusions of the paper prior to publication.

Major comments

1. The authors have not provided sufficient evidence to conclusively identify the additional density bound to human and frog γ -TuRC as a tetramer of NEDD1 bound to N-GCP3:MZT1. Currently, the assignment is based on prior work showing that NEDD1 associates with γ -TuRC purified from *X. laevis* and in-vitro studies showing NEDD1 C-termini form tetramers.

In the revised version of our manuscript, we validate tetramerization of NEDD1-N-GCP3/MZT1 (Supplementary Fig. 4d). See Reviewer #2, comment 2 for more details.

While the AlphaFold prediction fits the density well, the local resolution of the cryo-EM structure (4-8 Å) is too low to make a conclusive assignment and the authors have had to hide/exclude additional regions of the proteins (including the N-terminal beta-propeller domain of NEDD1, which is never mentioned in the text) which do not fit the density.

The beta-propeller domains of NEDD1 are not expected to be visible, because they are separated by a very long and highly flexible linker from NEDD1^C (Fig. 2d). Cryo-EM is an averaging based approach, in which such flexibly linked domains are averaged out and become invisible. In the revised manuscript version, we now mention and explain the absence of the NEDD1 beta-propeller domains in our cryo-EM reconstruction in the results section:

“Notably, the N-terminal beta-propeller domain of NEDD1 was not resolved, most likely because they are separated by long and flexible linkers from the stably docked NEDD1^C helices and therefore averaged out.”

We have now significantly improved local resolution for the NEDD1-containing density segment by optimizing the cryo-EM image analysis workflow. The improved density significantly strengthens our assignment of NEDD1 (Fig. 2b). Please see Reviewer #2, comment 2 for more details.

Further raising doubt on the assignment is an inconsistency with prior work showing that the C-terminus of NEDD1 (residues 572-660) directly binds γ -tubulin (PMID: 20224777).

The study mentioned by Reviewer #3 identifies NEDD1^C as the main interaction site between NEDD1 and complexes containing γ -tubulin, which is fully consistent with our cryo-EM reconstruction. However, for establishing direct interaction between NEDD1 and γ -tubulin, the study relies on interaction analysis of isolated NEDD1 and γ -tubulin recombinantly expressed in *E. coli*. γ -tubulin requires eukaryotic TRiC/CCT for correct folding²⁷ so γ -tubulin analyzed in this context may well be mostly unfolded and partially aggregated. If folded correctly, NEDD1 and γ -tubulin have exposed unoccupied binding interfaces in *E. coli* (normally associated with MZT1 modules and GCP2-6, respectively) and the interactions detected in this experiment may therefore well be unphysiological.

The authors need to provide additional evidence that NEDD1 is responsible for the grapnel-like density, as this is a major conclusion of the paper.

Please see our response to Reviewer #2, major point 2, in which we provide extensive new experimental data significantly strengthening our assignment of NEDD1.

2. The model that Augmin connects γ -TuRC to the MTTs of the centriole via POC5 is interesting and supported by the preliminary studies. However, additional studies, for example *in vitro* reconstitution or mutation of the NEDD1-Augmin interface – are needed to provide support for the model.

For the POC5-Augmin interaction, please see our response to Reviewer #2, major point 9, in which we show direct interaction between recombinantly expressed and purified POC5 and Augmin TII N-clamp *in vitro* (Supplementary Fig. 16).

To address the Augmin-NEDD1 interface, we reconstituted interaction between recombinantly expressed and purified Augmin TIII subcomplex from insect cells and purified NEDD1-N-GCP3/MZT1 complex *in vitro*. We observed clear pull-down of NEDD1-N-GCP3/MZT1 dependent on Augmin TIII, which could be strongly increased by incubation of Augmin TII with CDK1 and PLK1 kinases (Supplementary Fig. 18). This indicates direct interaction between NEDD1 and Augmin TIII.

3. All the structural predictions use AlphaFold2 with subcomplexes. The authors should demonstrate that these predictions are replicated with AlphaFold3 and larger subcomplexes.

As requested, we have repeated structural predictions using AlphaFold3. Most of the predictions from AlphaFold2 could be replicated by AlphaFold3 using the same subcomplexes and fragments (Figure R1).

In case of inconsistent predictions, the AlphaFold3 models were clearly not supported by our experimental structural data, while AlphaFold2 models fitted well. Using larger subcomplexes as suggested by the reviewer in most cases resulted in predictions that were not supported by our experimental structural data. Overall, this suggests that AlphaFold3 had a lower success rate in predicting complex structures in our case.

Due to the limited success in predicting models in AlphaFold3, we refrained from including the results into the manuscript.

Figure R 1. Superposition of models successfully predicted by AlphaFold2 and AlphaFold3. Superposition of AlphaFold2 predictions (colored models) and their respective counterparts as predicted in AlphaFold3 (dark grey).

Minor comments

1. The description of the γ -TuRC:NEDD1 structure could be improved to reduce reliance on prior knowledge of the γ -TuRC structure.

To address this comment, we have expanded the description of the γ -TuRC molecular architecture in the introduction section.

2. SDS-PAGE and mass spectrometry analysis should be provided of the purified *X. laevis* γ -TuRC to indicate whether other potential interaction partners are present. A pure sample would help support NEDD1 as being responsible for the grapnel-like density.

To address this comment, we have included SDS-PAGE analysis of the purified *X. laevis* γ -TuRC (Fig. 2a), showing a clear band for NEDD1.

Additionally, we subjected the purified *X. laevis* γ -TuRC to mass spectrometry analysis (Supplementary Table 1, Supplementary Dataset 1). As expected, γ -TuRC components were the most abundant non-contaminant proteins in the sample. Notably, NEDD1 was the most abundant γ -TuRC interactor, detected at similar total intensities as γ -TuRC proteins. This suggests that it was co-purified with γ -TuRC at high stoichiometry. The only known γ -TuRC interactor present among the top identified proteins in addition to NEDD1 was NME7 (Nucleoside diphosphate kinase 7). However, NME7 could be excluded to contribute to the grapnel density based on its structure, which does not contain any coiled-coil segments. Zona pellucida sperm-binding proteins, which were also detected at high abundance, could be excluded as a contaminant from the *X. laevis* egg extract, because they are integral plasma

membrane proteins lacking any soluble alpha-helices. Overall, this significantly strengthens our assignment of NEDD1 in the cryo-EM reconstruction.

We modified the main text to include this new experiment: “To further support our assignment of NEDD1 and to gain high-resolution insights into the interaction of NEDD1 with the γ -TuRC, we used *Xenopus laevis* egg extract to purify γ -TuRCs for cryo-EM SPA analysis¹³. Mass spectrometry analysis indicated that γ -TuRC components and NEDD1 were among the most abundant proteins in the sample (Supplementary Table 1, Supplementary Dataset 1), suggesting that NEDD1 was co-purified with γ -TuRC at high stoichiometry. NME7, another known γ -TuRC interactor²⁸ abundant in the sample, could be excluded to contribute to the grapnel density based on its structure, which does not contain any coiled-coil segments (PDB 7UNG⁸, PDB 7RRO²⁹).”

3. Data from Extended Data Figure 7D and Figure S13 could be moved to a main figure, as they provide key support for some of the main conclusions of the paper

We thank the Reviewer for the suggestion, but we prefer to keep the figure panels in the Supplementary Information of the revised manuscript version.

In case of Supplementary Fig. 7D (now Supplementary Fig. 9e), it is important to show the cryo-EM densities from different perspectives and superposed in different combinations at sufficient size to clearly demonstrate CDK5RAP2-dependency of the GCP2-associated additional densities. We believe that a Supplementary Figure is better suited for this purpose than a main figure panel.

Supplementary Fig. 13 (now Supplementary Fig. 17) mostly contains control experiments and data consistent with published work, which in our eyes does not warrant a complete main figure.

4. Label the C-terminus of NEDD1 in Figures 1 and 2

We have adapted the labeling as suggested.

5. Label individual tubules of the MTT in Figure 4

We have adapted the labeling as suggested.

6. Better labeling is required in Figure 5H to prevent it being misinterpreted as an experimentally resolved reconstruction.

We have specifically pointed out in the figure labeling that it is a predicted model supported by several lines of evidence.

7. The grapnel should be clearly labeled in the local resolution figure of Extended Data Fig. 4

We have adapted the labeling as suggested.

8. In the Research Animals section, “the” is repeated twice

We have corrected the text as suggested.

9. Is Augmin long enough to bridge the larger gaps between γ -TuRC and the MTT?

The distances shown in Fig. 4B have been measured between the γ -TuRC center-of-mass and the center of the MTT B-tubule. Thus, the effective distance between the proposed augmin binding site on the NEDD1/ γ -TuRC complex and the MTT surface ranges from ~35-55 nm. Taking into account some conformational flexibility of augmin and the MTT inner scaffold

protein POC5, this correlates well with the 45 nm separated binding sites for NEDD1 and POC5 on augmin.

References

- 1 Goddard, T. D. *et al.* UCSF ChimeraX: Meeting modern challenges in visualization and analysis. *Protein Sci* **27**, 14-25 (2018). <https://doi.org/10.1002/pro.3235>
- 2 Meng, E. C. *et al.* UCSF ChimeraX: Tools for structure building and analysis. *Protein Science* **32**, e4792 (2023). <https://doi.org/https://doi.org/10.1002/pro.4792>
- 3 Pettersen, E. F. *et al.* UCSF ChimeraX: Structure visualization for researchers, educators, and developers. *Protein Sci* **30**, 70-82 (2021). <https://doi.org/10.1002/pro.3943>
- 4 Choi, Y.-K., Liu, P., Sze, S. K., Dai, C. & Qi, R. Z. CDK5RAP2 stimulates microtubule nucleation by the γ -tubulin ring complex. *Journal of Cell Biology* **191**, 1089-1095 (2010). <https://doi.org/10.1083/jcb.201007030>
- 5 Rale, M. J., Romer, B., Mahon, B. P., Travis, S. M. & Petry, S. The conserved centrosomin motif, γ TuNA, forms a dimer that directly activates microtubule nucleation by the γ -tubulin ring complex (γ TuRC). *eLife* **11**, e80053 (2022). <https://doi.org/10.7554/eLife.80053>
- 6 Serna, M. *et al.* CDK5RAP2 activates microtubule nucleator γ TuRC by facilitating template formation and actin release. *Developmental Cell* (2024). <https://doi.org/https://doi.org/10.1016/j.devcel.2024.09.001>
- 7 Xu, Y. *et al.* Partial closure of the γ -tubulin ring complex by CDK5RAP2 activates microtubule nucleation. *Developmental Cell*, 2023.2012.2014.571518 (2024). <https://doi.org/10.1016/j.devcel.2024.09.002>
- 8 Gui, M. *et al.* SPACA9 is a luminal protein of human ciliary singlet and doublet microtubules. *Proceedings of the National Academy of Sciences* **119**, e2207605119 (2022). <https://doi.org/10.1073/pnas.2207605119>
- 9 Walton, T. *et al.* Axonemal structures reveal mechanoregulatory and disease mechanisms. *Nature* **618**, 625-633 (2023). <https://doi.org/10.1038/s41586-023-06140-2>
- 10 Gomez-Ferreria, M. A. *et al.* Novel NEDD1 phosphorylation sites regulate γ -tubulin binding and mitotic spindle assembly. *Journal of Cell Science* **125**, 3745-3751 (2012). <https://doi.org/10.1242/jcs.105130>
- 11 Manning, J. A., Shalini, S., Risk, J. M., Day, C. L. & Kumar, S. A Direct Interaction with NEDD1 Regulates γ -Tubulin Recruitment to the Centrosome. *PLOS ONE* **5**, e9618 (2010). <https://doi.org/10.1371/journal.pone.0009618>
- 12 Muñoz-Hernández, H. *et al.* Structure of the microtubule anchoring factor NEDD1 bound to the γ -tubulin ring complex. *bioRxiv*, 2024.2011.2005.622067 (2024). <https://doi.org/10.1101/2024.11.05.622067>
- 13 Liu, P. *et al.* Insights into the assembly and activation of the microtubule nucleator γ -TuRC. *Nature* **578**, 467-471 (2020). <https://doi.org/10.1038/s41586-019-1896-6>
- 14 Aher, A., Urnavicius, L., Xue, A., Neselu, K. & Kapoor, T. M. Structure of the γ -tubulin ring complex-capped microtubule. *Nature Structural & Molecular Biology* (2024). <https://doi.org/10.1038/s41594-024-01264-z>
- 15 Wieczorek, M., Huang, T.-L., Urnavicius, L., Hsia, K.-C. & Kapoor, T. M. MZT Proteins Form Multi-Faceted Structural Modules in the γ -Tubulin Ring Complex. *Cell*

- Reports* **31**, 107791 (2020).
<https://doi.org/https://doi.org/10.1016/j.celrep.2020.107791>
- 16 Wiczonek, M. *et al.* Asymmetric Molecular Architecture of the Human γ -Tubulin Ring Complex. *Cell* **180**, 165-175.e116 (2020).
<https://doi.org/10.1016/j.cell.2019.12.007>
- 17 Sala, C. *et al.* An interaction network of inner centriole proteins organised by POC1A-POC1B heterodimer crosslinks ensures centriolar integrity. *Nature Communications* **15**, 9857 (2024). <https://doi.org/10.1038/s41467-024-54247-5>
- 18 Laporte, M. H. *et al.* Time-series reconstruction of the molecular architecture of human centriole assembly. *Cell* **187**, 2158-2174.e2119 (2024).
<https://doi.org/10.1016/j.cell.2024.03.025>
- 19 Sprague, B. L., Pego, R. L., Stavreva, D. A. & McNally, J. G. Analysis of binding reactions by fluorescence recovery after photobleaching. *Biophys J* **86**, 3473-3495 (2004). <https://doi.org/10.1529/biophysj.103.026765>
- 20 Saito, T., Matsunaga, D. & Deguchi, S. Long-Term Fluorescence Recovery After Photobleaching (FRAP). *Methods Mol Biol* **2600**, 311-322 (2023).
https://doi.org/10.1007/978-1-0716-2851-5_21
- 21 Ustinova, K., Ruhnow, F., Gili, M. & Surrey, T. Microtubule binding of the human augmin complex is directly controlled by importins and Ran-GTP. *Journal of Cell Science* **136** (2023). <https://doi.org/10.1242/jcs.261096>
- 22 Muroyama, A., Seldin, L. & Lechler, T. Divergent regulation of functionally distinct γ -tubulin complexes during differentiation. *Journal of Cell Biology* **213**, 679-692 (2016). <https://doi.org/10.1083/jcb.201601099>
- 23 Uehara, R. *et al.* The augmin complex plays a critical role in spindle microtubule generation for mitotic progression and cytokinesis in human cells. *Proc Natl Acad Sci U S A* **106**, 6998-7003 (2009). <https://doi.org/10.1073/pnas.0901587106>
- 24 Bauer, M., Cubizolles, F., Schmidt, A. & Nigg, E. A. Quantitative analysis of human centrosome architecture by targeted proteomics and fluorescence imaging. *Embo j* **35**, 2152-2166 (2016). <https://doi.org/10.15252/embj.201694462>
- 25 Fischer, T. *et al.* Yeast centrin Cdc31 is linked to the nuclear mRNA export machinery. *Nat Cell Biol* **6**, 840-848 (2004). <https://doi.org/10.1038/ncb1163>
- 26 Moudjou, M., Bordes, N., Paintrand, M. & Bornens, M. gamma-Tubulin in mammalian cells: the centrosomal and the cytosolic forms. *J Cell Sci* **109** (Pt 4), 875-887 (1996). <https://doi.org/10.1242/jcs.109.4.875>
- 27 Melki, R., Vainberg, I., Chow, R. & Cowan, N. Chaperonin-mediated folding of vertebrate actin-related protein and gamma-tubulin. *Journal of Cell Biology* **122**, 1301-1310 (1993). <https://doi.org/10.1083/jcb.122.6.1301>
- 28 Liu, P., Choi, Y.-K. & Qi, R. Z. NME7 is a functional component of the γ -tubulin ring complex. *Molecular Biology of the Cell* **25**, 2017-2025 (2014).
<https://doi.org/10.1091/mbc.e13-06-0339>
- 29 Gui, M. *et al.* *De novo* identification of mammalian ciliary motility proteins using cryo-EM. *Cell* **184**, 5791-5806.e5719 (2021).
<https://doi.org/10.1016/j.cell.2021.10.007>

Point-by-Point Reply to Reviewers

We thank the reviewers for their helpful comments on our revised manuscript. Below we address the remaining points and elaborate on the corresponding changes in the manuscript.

Reviewer #1 (Remarks to the Author)

This reviewer is completely satisfied.

Reviewer #2 (Remarks to the Author)

The authors have done a great effort to address the reviewers' concerns and the manuscript has been substantially improved. In particular the structural analyses with improvements in resolution are now more convincing.

We thank the Reviewer for the positive evaluation of our revised manuscript.

The following remaining issues should be addressed:

Our previous comment:

19) The POC5 KO background produces mitotic defects in most of the cells according to Fig. 6I. However, the nature of the mitotic defects is unclear. Since POC5 RNAi was shown to produce structurally aberrant centrioles (Steib et al., 2020, Elife), which may interfere with mitosis, the authors should quantify centriole number and configuration using a centriole marker such as centrin and test for structural abnormalities by UExM. The same should be performed for the rescue lines expressing wild type POC5 and mutant POC5. If centrioles are abnormal in POC5 mutant cells (due to a defective inner scaffold), this would readily explain the mitotic defects.

Author reply:

We now show in Supplementary Fig. 17g that centrioles are not fragmented in POC5mut cells in contrast to POC5^{-/-} cells. Similarly, expression of POC5 and POC5mut rescues centriole number compared to POC5^{-/-} cells (Supplementary Fig. 17d). This indicates that the inner scaffold is intact and functional in POC5mut centrioles.

Our new comment:

Supp Fig 17D: The description in the results refers to centrioles, but g-tub is quantified here, which does not allow counting of centrioles. In the plot, what are "defined" g-tub foci? Are there additional "undefined" foci?

A centriole marker should be used (ideally centrin), as asked in the first review round, to confirm that there are no abnormalities in centriole numbers/configurations in POC5 vs POC5 mut.

Most importantly, centrin staining should be done for the mitotic figures in Fig 6I that we refer to in our original comment and where the extra g-tub foci are observed.

We agree that in specific cases γ -tubulin signal may not be suited to quantify the number of centrioles/centrosomes (e.g. in case of multiple γ -tubulin dots, which may reflect PCM fragmentation). However, in this case, we see exactly two γ -tubulin signals in most POC5 and POC5^{mut} cells (no sign of PCM fragmentation, Supplementary Fig. 18e) and in addition, almost quantitative co-

localization between foci of γ -tubulin and POC5-HA/POC5^{mut}-HA (Supplementary Fig. 18d). POC5 is a *bona-fide* centriole component that binds to centrin and therefore at least in part reflects the localization of centrin. POC5^{mut} has the same centriole localization as POC5 (Supplementary Fig. 18g) and can therefore also be used as centriole marker. To make this clearer, we have added additional POC5^{mut} cell examples to Supplementary Fig. 18b and have quantified co-localization of the γ -tubulin and POC5-HA signals in Supplementary Fig. 18d. Based on the combination of γ -tubulin and POC5 as markers, we can conclude that centrosomes are not amplified or fragmented in POC5^{mut} cells.

We agree that using the word 'defined' is confusing and removed it from the figure legend.

The authors should explain how they selected the centrioles for U-ExM and quantifications in Supp Fig 17 (since only isolated centrioles are shown). This is important because the results will depend on the cell cycle stage, whether daughter or mother centrioles are analysed, etc.

In Supplementary Fig. 18g-k, we analyzed mother centrioles of interphase cells (daughter centrioles were not quantified). Most centrioles were in G1 phase (83%) or S/G2 phase (17%; attached daughter centriole). Importantly, POC5 or POC5^{mut} G1 centrioles did not show signs of fragmentation (Supplementary Fig. 18i), which strongly suggests that centrioles were intact during mitosis.

To clarify this point, we have attached a figure below in which we compare centriole length and fragmentation between G1 cells (only one centriole) and S/G2 cells (mother centrioles with attached daughter). The tendency between both cell populations is the same: POC5^{-/-} cells have more broken or shorter centrioles, while centriole length and the percentage of broken centrioles is unaffected in POC5 and POC5^{mut} cells compared to control cells.

Figure R 1. The percentage of broken centrioles and centriole length are comparable between the centrioles from G1 cells and G2/S cells across four different cell lines.

a) The percentage of broken centrioles in G1 and S/G2 from wild type control, POC5^{-/-}, POC5 and POC5^{mut} cells based on U-ExM images. N=2 independent experimental repeats were performed, and a total of at least n=31 in G1 and n=5 in S/G2 centrioles were analyzed. Two-way ANOVA was applied for statistical analysis. **b)** Measurement of centriole length in G1 and S/G2 from wild type control, POC5^{-/-}, POC5-HA and POC5^{mut}-HA cells based on U-ExM images. N=2 independent experimental repeats were performed, and a total of at least n=20 samples in G1 and n=4 in S/G2 were analyzed. Two-way ANOVA was applied for statistical analysis.

The authors should provide example images for the centrioles quantified in Supp Fig 17G (centriole length, % broken centrioles). Also, the graph axis label says broken "centrosomes", this should be "centrioles" (I assume).

We have included example images for the centrioles quantified in Supplementary Fig. 18h and corrected the axis labeling as suggested.

Author reply:

What we see during mitosis in *POC5*^{mut} cells is an increase in mitotic γ -tubulin centers and an increase in metaphase cells with misaligned chromosomes (Fig. 6h,i). *POC5*^{mut} cells in comparison to *POC5* wild-type cells show γ -tubulin and augmin release from interphase centrioles and overall less cellular γ -tubulin and augmin. Taken these observations together, it is sensible to propose the model that the γ -tubulin and augmin localization inside interphase centrioles and their timely release in mitosis are important to prevent mitotic defects.

Our new comment:

Assuming there are no centriole defects (see above) and since the total levels of augmin and *gtub* are reduced, an alternative explanation would be that the authors see an augmin depletion phenotype for *POC5*^{mut} cells, which also causes spindle pole fragmentation (see Lawo et al 2009 Curr Biol). If this cannot be excluded, it should be discussed.

In response to this comment, we have included a short discussion section:

„This notion is consistent with the metaphase chromosome misalignment in *POC5*^{mut} cells. As centriole fragmentation was not observed in *POC5*^{mut} cells with an intact inner *POC5* scaffold, this defect is likely attributable to reduced levels of γ -tubulin and Augmin and the strongly reduced amounts of both proteins to be released from the centriole lumen at the beginning of mitosis. However, we cannot entirely rule out the possibility that subtle centriole aberrations also contribute to this phenotype.“

Reviewer #3 (Remarks to the Author)

In my opinion, the authors have done a tremendous job of addressing the concerns of the reviewers. The additional experiments strongly support the assignment of NEDD1, and I have no further comments except to congratulate the authors on an impressive body of work.